# Linear Regression with Unknown Truncation Beyond Gaussian Features

**Alexandros Kouridakis** [1]  **Anay Mehrotra** [2]  **Alkis Kalavasis** [3]  **Constantine Caramanis** [1]

## Abstract

In truncated linear regression, samples $(x, y)$ are shown only when the outcome $y$ falls inside a certain survival set $S^\star$ and the goal is to estimate the unknown $d$-dimensional regressor $w^\star$. This problem has a long history of study in Statistics and Machine Learning going back to the works of (Galton, 1897; Tobin, 1958) and more recently in, *e.g.*, (Daskalakis et al., 2019; 2021; Lee et al., 2023; 2024). Despite this long history, however, most prior works are limited to the special case where $S^\star$ is precisely known. The more practically relevant case, where $S^\star$ is unknown and must be learned from data, remains open: indeed, here the only available algorithms require strong assumptions on the distribution of the feature vectors (*e.g.*, Gaussianity) and, even then, have a $d^{\mathrm{poly}(1/\varepsilon)}$ run time for achieving $\varepsilon$ accuracy. In this work, we give the first algorithm for truncated linear regression with unknown survival set that runs in $\mathrm{poly}(d/\varepsilon)$ time, by only requiring that the feature vectors are sub-Gaussian. Our algorithm relies on a novel subroutine for efficiently learning unions of a bounded number of intervals using access to positive examples (without any negative examples) under a certain smoothness condition. This learning guarantee adds to the line of works on positive-only PAC learning and may be of independent interest.

## 1. Introduction

Selection biases, that is, systematic omissions in data, arise in almost all domains, from the physical sciences to econometrics and social network analysis. Indeed, as Lisa Gitelman argues, *all* data exhibits selection biases and the term "raw data" is an oxymoron (Gitelman, 2013). Among different types of selection biases, *truncation* is one of the most ubiquitous and challenging: truncation arises when samples are recorded only when they fall inside a "survival" region $S^\star$, and all other samples are deleted, or *truncated*, from the data, *e.g.*, (Tobin, 1958). Truncation arises naturally across diverse domains: from econometrics (Hausman & Wise, 1976) and astronomy (Woodroofe, 1985) to medical and biological sciences (Klein & Moeschberger, 2003), engineering (Jiang et al., 2020; Meeker et al., 2021), and the social sciences (Breen, 1996). The difficulty of truncation lies in the fact that no samples are observed outside the survival region. Thus, any algorithm must extrapolate from data observed within $S^\star$ to the region outside it. Standard estimation procedures such as ordinary least squares fail to handle this and produce biased estimates, *e.g.*, (Maddala, 1983; Cohen, 1991; Tobin, 1958).

In this work, we revisit the classical problem of *truncated linear regression*, which has a rich history of study in statistics (Cohen, 1991), machine learning (*e.g.*, (Daskalakis et al., 2018; 2019; Plevrakis, 2021; Lee et al., 2023; 2024)), and related fields (Maddala, 1983). In this problem, we observe samples of the form $(x, y)$ where $x$ is a $d$-dimensional feature vector and $y$ is a scalar outcome. These samples are generated as follows: first, the feature vector $x$ is drawn from a distribution $\mathcal{D}$; then, the output $y = x^\top w^\star + \xi$ is computed, where $w^\star$ is an unknown regression vector and $\xi \sim \mathcal{N}(0, 1)$ is standard Gaussian noise. Unlike in standard linear regression, we do not observe every sample $(x, y)$. Instead, there exists a set $S^\star \subseteq \mathbb{R}$, called the *survival set*, such that we observe $(x, y)$ if and only if $y \in S^\star$; otherwise, the sample is hidden from us. The observed dataset thus consists of i.i.d. draws from the distribution of $(x, y)$ conditioned on $y \in S^\star$, and the goal is to estimate $w^\star$ from these truncated samples (see Definition 1).

To gain some intuition, consider the following example from labor economics: Hausman & Wise (1976) studied the effect of education and (different measures of) intelligence (which are modeled as a feature vector $x$) on the earnings $y$ in low-income populations, but discovered that the dataset only included an observation $(x, y)$ when the annual income $y$ lay below a prescribed eligibility threshold $\tau$, leading to the truncation set $S^\star = (-\infty, \tau]$. More broadly, the sur-

[1] University of Texas at Austin, U.S.A. [2] Stanford University, U.S.A. [3] Yale University, U.S.A.. Correspondence to: Alexandros Kouridakis <alexkouridakis@utexas.edu>, Anay Mehrotra <anaymehrotra1@gmail.com>, Alkis Kalavasis <alkis.kalavasis@yale.edu>, Constantine Caramanis <constantine@utexas.edu>.

*Proceedings of the 43rd International Conference on Machine Learning*, Seoul, South Korea. PMLR 306, 2026. Copyright 2026 by the author(s).

vival sets arising in real-world datasets can be far more complicated than a single threshold on the response. A representative example comes from modern astronomical surveys, where whether a star of a certain luminosity $y$ is observed is governed by a multi-stage pipeline that can truncate different non-contiguous intervals of luminosities. In practice, the resulting truncation set reflects a composition of instrument detection limits, survey design choices, and upstream catalog filters, among other factors, together making $S^\star$ complex and hard to know explicitly (Teerikorpi, 1997; Woodroofe, 1985; SDSS-IV Collaboration, 2022a;b) (See Appendix A for a concrete list of various sources of truncation in large-scale astronomical surveys.) Beyond direct applications in problems where truncation arises, algorithms for truncated linear regression also have applications in other domains, including treatment effect estimation in observational studies (see, *e.g.*, (Cai et al., 2025) and references therein).

**Prior Work.** Classical works in statistics laid the groundwork for truncated linear regression (Tobin, 1958; Hausman & Wise, 1976; Cohen, 1991). However, these works were restricted to simple survival sets and did not come with computational efficiency guarantees in high-dimensional feature-regimes that regularly arise in modern applications. Indeed, no computationally efficient algorithm was known in high dimensions until the work of Daskalakis et al. (2019), who studied the special case where the truncation set $S^\star$ is precisely known via a membership oracle.[1] Under certain assumptions (which we discuss later in this section), they gave a $\mathrm{poly}(d/\varepsilon)$-time algorithm to obtain an $\varepsilon$-approximation of $w^\star$. Several subsequent works have continued the study of truncated linear regression in this known-survival-set setting, *e.g.*, (Daskalakis et al., 2020; 2021; Lee et al., 2023).

However, as the earlier examples illustrate, in many practical settings the survival set is not known to the analyst (Zampetakis, 2022). When $S^\star$ is unknown, no efficient algorithms are known in high dimensions. In fact, even getting finite sample complexity bounds is non-trivial: if we put *no* assumptions on $S^\star$, the parameter estimation problem is information-theoretically impossible (Daskalakis et al., 2018). To ensure statistical tractability, one must assume that $S^\star$ has bounded "complexity", for instance, that $S^\star$ is a union of at most $k$ intervals (note that as $S^\star$ is a subset of $\mathbb{R}$, it is always a union of intervals.) This leads us to the following problem, which is the main focus of this paper.

---

**Main Question.** Is there an algorithm that learns $w^\star \in \mathbb{R}^d$ to $\varepsilon$ accuracy in $\mathrm{poly}(dk/\varepsilon)$ time in truncated linear regression when $S^\star$ is unknown and a union of (at most) $k$ intervals?

---

Under this assumption, Kontonis et al. (2019) established a sample complexity bound of $\mathrm{poly}(dk/\varepsilon)$ but had an exponential-in-$d$ running time and, hence, were not qualitatively faster than brute-force algorithms also running in exponential-in-$d$ time (see Section 1.1). The recent work of Lee et al. (2024) made progress by designing the first faster-than-brute-force algorithm for this problem, but only applied to the stylized case where the feature distribution $\mathcal{D}$ is Gaussian and its running time remains exponential, namely $d^{\mathrm{poly}(k/\varepsilon)}$.[2] This assumption of Gaussianity on the features is crucial for their algorithm to work, but it constitutes a very strong requirement on the feature vectors, which does not hold in real-world regression data. [3]

**Our Contributions.** Our main contribution (Theorem 1.1) is a $\mathrm{poly}(dk/\varepsilon)$-time algorithm that, given access to samples from the truncated regression model, outputs an $\varepsilon$-accurate estimate of $w^\star$ without knowledge of $S^\star$. Our algorithm is computationally efficient and importantly operates under a set of assumptions that are the same as, or much weaker than, those in prior works. Notably, we do not require the feature-distribution $\mathcal{D}$ to have a parametric form, *e.g.*, Gaussian. Instead, we only need $\mathcal{D}$ to satisfy the significantly weaker requirement of sub-Gaussianity. Concretely, we require the following assumptions.

**Assumption 1** (Survival Probability)**.** *There is a known constant $\alpha \in (0, 1]$ such that $\mathbb{E}_{x \sim \mathcal{D}}[\Pr[y \in S^\star \mid x]] \geq \alpha$.*

**Assumption 2** (Sub-Gaussianity and Boundedness)**.** *There are known $\sigma, R \geq 1$, s.t., the feature distribution $\mathcal{D}$ is $\sigma$-sub-Gaussian and regression parameter satisfies $\|w^\star\| \leq R$.*

**Assumption 3** (Identifiability)**.** *Let $\mathcal{D}_{\mathrm{obs}}$ be the distribution of $x$ in the observed data (see Equation (1)). There exists a constant $\rho \in (0, 1)$ such that $\mathbb{E}_{x \sim \mathcal{D}_{\mathrm{obs}}}[xx^\top] \succeq \rho^2 \cdot I$.*

A few remarks about the assumptions are in order:

1. The first assumption ensures that the amount of truncation is not too aggressive; at least a constant fraction of the points survive. This assumption is information theoretically necessary: indeed, as $\Pr[y \in S^\star]$ approaches $0$, then an unbounded number of samples are required to estimate $w^\star$ (as proved by (Daskalakis et al., 2018)).

2. The second assumption is a mild and standard tail condition on the features. In fact, Assumption 2 is even weaker than the assumptions required by prior works studying

---

[1]A membership oracle for $S^\star$ is a primitive that, given a point $x$, outputs Yes if $x \in S^\star$ and No otherwise.

[2]When $d \gg k, 1/\varepsilon$, it holds that $d^{\mathrm{poly}(k/\varepsilon)} \ll (d/\varepsilon)^d$.

[3]While Lee et al. (2024) require Gaussianity and have an exponential-in-$d$ running time, their approach is technically interesting as they are able to handle high-dimensional truncation sets, over both $x$ and $y$, while all prior work on truncated regression, including ours, focus on handling truncation on $y$; see Remark B.2.

the easier version of truncated regression with *known* survival sets: they require $\mathcal{D}$ to have a bounded support (Daskalakis et al., 2019; 2021).

3. The third assumption is required to ensure that $w^\star$ can be identified from data in the survival region; it is also made by prior works, *e.g.*, (Daskalakis et al., 2021). We prove that it is *necessary* in Appendix G. We also mention that since we have sample access to $\mathcal{D}_{\text{obs}}$, we can estimate the parameter $\rho$ from samples.

In short, Assumptions 1 and 3 are *necessary* for the problem to be solvable by a sample-efficient algorithm, while Assumption 2 is weaker compared to corresponding assumptions in prior work. We are now ready to state our main result in more detail.

**Theorem 1.1** (Informal; see Theorem 3.1)**.** *Assume Assumptions 1 to 3 hold with $\alpha, \rho = \Omega(1)$, $\sigma = O(1)$, $R = \text{poly}(d)$, and $S^\star$ is a union of at most $k$ intervals (for known $k$). Then, there is an algorithm that given $n = \text{poly}(dk/\varepsilon)$ i.i.d. samples from the truncated linear regression model, with probability $0.99$, outputs an estimate $\widehat{w}$ satisfying $\|\widehat{w} - w^\star\|_2 \leq \varepsilon$. The algorithm runs in time $\text{poly}(n)$.*

Importantly, both the sample complexity and running time of our algorithms scale polynomially in the relevant parameters $d, k, 1/\varepsilon$. Moreover, the success probability can be made $1 - \delta$ for any $\delta \in (0, 1)$. Since the algorithm only requires Assumptions 2 and 3 on distribution $\mathcal{D}$, it extends to the cases where $\mathcal{D}$ is non-Gaussian. Further, in the special case where $\mathcal{D}$ is Gaussian, the above result improves the best running time (namely, $d^{\text{poly}(k/\varepsilon)}$ by (Lee et al., 2024)) to polynomial in all parameters of interest (see Theorem 3.1).

**Technical Overview.** Next, we briefly overview the main challenges in proving Theorem 1.1 and how we overcome them, while comparing with the work of (Lee et al., 2024) which requires Gaussian features. A more detailed overview appears in Section 3.

**Approach with known $S^\star$.** Without any truncation, the standard method for regression is ordinary least squares (OLS). A principled way to derive OLS is to view it as a special case of minimizing the negative log-likelihood (NLL). This viewpoint extends naturally to truncated linear regression: existing algorithms for estimating $w^\star$ with known survival set $S^\star$ minimize the (population version of the) NLL function $\mathcal{L}_{S^\star}(\cdot)$ using, *e.g.*, stochastic gradient descent (SGD) (Lee et al., 2023; Daskalakis et al., 2019; 2021). This approach works because the NLL satisfies two key properties:

P1. $\mathcal{L}_{S^\star}(\cdot)$ is convex and $w^\star$ is its unique minimizer.

P2. $\mathcal{L}_{S^\star}(\cdot)$ is strongly convex near $w^\star$.

Convexity ensures that one can find $\widehat{w}$ such that $\mathcal{L}_{S^\star}(\widehat{w}) \approx \mathcal{L}_{S^\star}(w^\star)$, and strong convexity then implies $\widehat{w} \approx w^\star$. How-

ever, the NLL depends explicitly on $S^\star$, and hence new techniques are required when $S^\star$ is unknown.

**Existing approach with unknown $S^\star$.** A natural idea is to first learn an approximation $S$ of $S^\star$ and then minimize $\mathcal{L}_S(\cdot)$. However, $\mathcal{L}_S(\cdot)$ is not even well-defined when $S \neq S^\star$, so this direct approach fails. Lee et al. (2024) bypass this issue by designing a generalized objective $\widetilde{\mathcal{L}}_S(\cdot)$ that remains well-defined for $S \neq S^\star$ and approximately retains the good properties of the NLL when $\mathcal{D}$ is Gaussian. Their algorithm works in two phases: first, (i) it learns an approximation $S$ of $S^\star$ in $d^{\text{poly}(k/\varepsilon)}$ time; then, (ii) it minimizes $\widetilde{\mathcal{L}}_S(\cdot)$ using SGD. Both phases (i) and (ii) require the Gaussianity assumption on the covariates, and the first set learning part runs in super-polynomial time. We examine these issues and discuss how to overcome them next.

**Issue I: Properties and Optimization of $\widetilde{\mathcal{L}}_S(\cdot)$ without Gaussianity.** We begin by examining phase (ii), and we will discuss phase (i) afterwards. Hence, let us assume that the learning algorithm is given an approximation to the true set $S \approx S^\star$. Now, we have to understand how to show the desired properties of NLL (phase (ii)) without Gaussianity. In particular, using the fact that $x$ is Gaussian, Lee et al. (2024) prove that $\mathbb{E}_x[\boldsymbol{\nabla}\widetilde{\mathcal{L}}_S(w; x)] \approx 0$[4] when $w \approx w^\star$ (which implies that $w^\star$ is an approximate minimizer of $\widetilde{\mathcal{L}}_S(\cdot)$). The key technical ingredient for their proof is to show that the gradient $\boldsymbol{\nabla}\widetilde{\mathcal{L}}_S(w; x)$ evaluated at $x$ is sub-exponential, which crucially uses the Gaussianity of $x$.

To go beyond Gaussian features, we follow a different route. We prove two important properties: for any $w \approx w^\star$, we show that (i) $\boldsymbol{\nabla}\widetilde{\mathcal{L}}_S(w; x) \approx 0$ with probability $1 - o(1)$ over the draw of $x$ in the observed data, and (ii) while $\boldsymbol{\nabla}\widetilde{\mathcal{L}}_S(w; x)$ can be large (and unbounded) with probability $o(1)$, we show that by suitably post-processing the set $S$, we can ensure $\mathbb{E}[\|\boldsymbol{\nabla}\widetilde{\mathcal{L}}_S(w; x)\|] \leq \text{poly}(d)$; this is one place where we utilize the sub-Gaussianity of $x$ to ensure that this post-processing can be performed while retaining the fact that $S$ is a good approximation of $S^\star$. These two properties together suffice to show that $\boldsymbol{\nabla}\widetilde{\mathcal{L}}_S(w) \approx 0$ when $w \approx w^\star$ (Lemma E.10), implying that $w^\star$ is close to the minimizer of $\widetilde{\mathcal{L}}_S$ (Lemma E.11).

Finally, we must still minimize $\widetilde{\mathcal{L}}_S(\cdot)$ efficiently. Here, prior works perform SGD analysis on NLL but require a stronger assumption on the survival probability. To this end, since our assumption on survival probability (Assumption 1) is simpler than that in (Daskalakis et al., 2019), we have to perform a new, more complicated analysis for SGD (see

---

[4]Here $\boldsymbol{\nabla}\widetilde{\mathcal{L}}_S(w; x)$ corresponds to the gradient of the NLL evaluated at the single point $x$ with guess vector $w$. Since we study the population version of NLL, we take an expectation over $x$ drawn from $\mathcal{D}_{\text{obs}}$. For a formal derivation of the gradient, we refer to Appendix E.2.1.

Section 3 for details).

**Issue II: Efficiently learning $S^\star$ from positive samples only.** For the above phase (ii) to be useful, we need to get an estimation for the true set $S^\star$ (and in particular in time $\text{poly}(dk/\varepsilon)$). Hence, our goal is to efficiently find $S \subseteq \mathbb{R}$ such that $\Pr[\mathbb{1}\{y \in S^\star\} \neq \mathbb{1}\{y \in S\}] \approx 0$; the challenge is that we only observe samples $y$ inside $S^\star$ (i.e., only positive samples) and no samples outside $S^\star$ (i.e., no negative samples). In particular, if we denote by $\mathcal{D}_y$ the marginal on $y$, then the learner observes points $y$ from the distribution with density proportional to $\mathcal{D}_y(y)\mathbb{1}\{y \in S^\star\}$. Most results on PAC learning from positive examples are negative: without assumptions on the underlying distribution $\mathcal{D}_y$, even simple classes such as unions of $k > 1$ intervals are information-theoretically unlearnable (Natarajan, 1987; Shvaytser, 1990). A standard way to circumvent this impossibility is to obtain auxiliary information about the marginal distribution $\mathcal{D}_y$ of $y$, e.g., by finding (and using samples from) a distribution $\widetilde{\mathcal{D}}_y$ that is "close" to it (Denis, 1998; Lee et al., 2026); see Definition 2.

In the truncated linear regression setting, Lee et al. (2024) show how to construct $\widetilde{\mathcal{D}}_y$ when $x$ is Gaussian; this relies heavily on the parametric form of the Gaussian and even fails for other parametric distributions. Our main contribution here is to give a new way to construct this auxiliary distribution $\widetilde{\mathcal{D}}_y$ (Lemma 3.5) by only using the sub-Gaussianity of $x$ (without having any specific parametric form) and appropriate upper and lower bounds on the tail of the noise-distribution.

Finally, we mention that even with access to $\widetilde{\mathcal{D}}_y$, we need to design an algorithm for learning the set that runs in time $\text{poly}(d, k, 1/\varepsilon)$. Existing algorithms require constrained variants of empirical risk minimization (Lee et al., 2026), which are often NP-hard in high dimensions. We show that the constrained empirical risk minimization algorithm of Lee et al. (2026) can be implemented in $\text{poly}(dk/\varepsilon)$ time for one-dimensional sets (Theorem 3.4), using a carefully designed greedy procedure (see Algorithm 1).

### 1.1. Related Works

Truncated statistics has a long history of study in statistics and econometrics (Galton, 1897; Pearson, 1902; Pearson & Lee, 1908; Lee, 1914; Fisher, 1931; Hannon & Dahiya, 1999; Raschke, 2012) and has attracted considerable recent interest in machine learning (Daskalakis et al., 2018; 2020; Kontonis et al., 2019; Nagarajan & Panageas, 2020; Plevrakis, 2021; Fotakis et al., 2020; Nagarajan et al., 2023; Tai & Aragam, 2023; Lee et al., 2023; De et al., 2023; 2024; Diakonikolas et al., 2024; Galanis et al., 2024; Kalavasis et al., 2024; Zampetakis & Zhou, 2025; Chauhan & Panageas, 2026). We refer the reader to Maddala (1983) for foundational works. Below, we focus on works most

relevant to ours.

**Truncated Linear Regression.** Daskalakis et al. (2019) give the first polynomial-time algorithms for truncated linear regression with bounded features and a *known* survival set under variants of Assumption 1; Daskalakis et al. (2021) later extended this to unknown noise variance. In contrast, we study the significantly harder case where the survival set is *unknown*. Even for known $S^\star$, our results slightly generalize (Daskalakis et al., 2019) by only requiring the feature distribution to be sub-Gaussian rather than bounded.[5]

For unknown $S^\star$, Kontonis et al. (2019) gave an algorithm for Gaussian features with $\text{poly}(dk/\varepsilon)$ sample complexity but $k \cdot (d/\varepsilon)^d$ running time.[6] Lee et al. (2024) improved the running time to $d^{\text{poly}(k/\varepsilon)}$, still for Gaussian features. Theorem 3.1 improves upon both works: we only assume sub-Gaussian features and achieve $\text{poly}(dk/\varepsilon)$ running time.

**Learning from Positive Examples.** A key subroutine in our work is learning the survival set from positive examples only. This problem dates back to Valiant (Valiant, 1984) and was formalized by Natarajan (1987), who characterized classes that can be properly PAC learned from positive samples alone. Shvaytser (1990); Kivinen (1995) completed the characterization for improper learners and bounded the sample complexity. These results are largely negative: even unions of $k > 1$ intervals in one dimension are information-theoretically unlearnable without additional assumptions. Subsequent works have designed algorithms when some information about the data distribution $\mathcal{D}$ is available: Denis (1998) reduced the problem to agnostic PAC learning when $\mathcal{D}$ is known; Lee et al. (2024) relaxed this to knowing a distribution $\widetilde{\mathcal{D}}$ close to $\mathcal{D}$; and Lee et al. (2026) further generalized this to mild smoothness assumptions on $\mathcal{D}$. However, most general algorithms in these works run in time $d^{\text{poly}(1/\varepsilon)}$ or slower. Our work contributes to this line by giving an efficient algorithm to learn unions of $k$ intervals in one dimension when the data distribution satisfies a smoothness condition (Definition 2).

## 2. Preliminaries and Model

In this section, we introduce key preliminaries and then formally introduce the model studied in this work. We begin with some basic notation.

**Notation.** Let $\mathcal{N}(\mu, \sigma^2)$ be the 1-dimensional normal distribution with mean $\mu$ and variance $\sigma^2$. We use $\mathcal{N}(x; \mu, 1)$

---

[5]Daskalakis et al. (2019) also work in the fixed-design setting. Our algorithms extend to fixed design with known $S^\star$, but we focus on the i.i.d. setting as it is more natural with unknown $S^\star$.

[6]Kontonis et al. (2019) also had a $d^{\text{poly}(k/\varepsilon)}$-time method, but this did not apply to truncated linear regression; it was for the simpler task of truncated gaussian mean-estimation with covariance $\Sigma \approx I$.

to denote the density of $\mathcal{N}(\mu, 1)$ at $x$ and $\mathcal{N}(S; \mu, 1) :=$ $\int_S \mathcal{N}(x; \mu, 1) \, dx$ to denote the mass that $\mathcal{N}(\mu, 1)$ assigns to a measurable set $S$. Further, we denote by $\mathcal{N}(\mu, 1, S)$ the distribution $\mathcal{N}(\mu, 1)$ truncated on the set $S$, and the density of this distribution at $y$ by $\mathcal{N}(y; \mu, 1, S)$. For an event $\mathcal{E}$, we write $\mathbb{1}\{\mathcal{E}\} \in \{0, 1\}$ for the indicator of $\mathcal{E}$. For vectors $u, v \in \mathbb{R}^d$, we denote $\langle u, v \rangle := u^\top v$ and $\|u\|_2^2 := \sum_i u_i^2$. For $A \in \mathbb{R}^{d \times d}$, we write $\|A\|_{\mathrm{op}} := \max_{\|v\|_2 = 1} \|Av\|_2$. For symmetric $A, B \in \mathbb{R}^{d \times d}$, $A \succeq B$ denotes that $A - B$ is positive semi-definite. For $k \geq 1$, let $\mathcal{S}_k$ be the family of unions of at most $k$ (possibly unbounded) intervals in $\mathbb{R}$, and $\mathcal{H}_k := \{\mathbb{1}_S : S \in \mathcal{S}_k\}$. We use standard asymptotic notation (*e.g.*, $O(\cdot), \widetilde{O}(\cdot), \lesssim$) to suppress constants.

**Sub-Gaussianity.** We will use standard definitions for sub-Gaussian random variables and random vectors (see (Vershynin, 2018)). For a random variable $X \in \mathbb{R}$, its *sub-Gaussian* norm is defined as (see also Definition 4)

$$\|X\|_{\psi_2} := \inf \left\{ \sigma > 0 \colon \mathbb{E}\, e^{X^2/\sigma^2} \leq 2 \right\} \ .$$

We will say that $X$ is $\sigma$-sub-Gaussian if $\|X - \mathbb{E}\, X\|_{\psi_2} \leq \sigma$. For a random vector $X \in \mathbb{R}^d$, we will say that $X$ is a $\sigma$-sub-Gaussian random vector if, for any unit vector $v \in \mathbb{R}^d$, the random variable $\langle X, v \rangle$ is $\sigma$-sub-Gaussian.

**Definition 1** (Truncated Linear Regression Model)**.** *The truncated linear regression model is parameterized by a distribution $\mathcal{D}$ on $\mathbb{R}^d$, a* survival set $S^\star$ *that is a measurable subset of the real line $\mathbb{R}$ (w.r.t. the Lebesgue measure), and an* unknown *parameter $w^\star \in \mathbb{R}^d$. A sample $(x, y) \in \mathbb{R}^d \times \mathbb{R}$ from the model is generated as follows:*

1. *The feature $x$ is sampled from $\mathcal{D}$ (independent of all other random variables)*
2. *The label $y$ is computed as $y = \langle w^\star, x \rangle + \xi$ where $\xi$ is independent Gaussian noise, i.e., $\xi \sim \mathcal{N}(0, 1)$.*
3. *If $y \in S^\star$, the sample is returned and, otherwise, Step 1 is repeated.*

Equivalently, a sample $(x, y)$ of the truncated linear regression model is generated by sampling $x \sim \mathcal{D}$ and $y = \langle w^\star, x \rangle + \xi$ conditioned on $y \in S^\star$; thus the observed data are i.i.d. from $(x, y) \mid (y \in S^\star)$. Note that both $\mathcal{D}$ and $S^\star$ are unknown to the learner. Furthermore, we denote by $\mathcal{D}_{\mathrm{obs}}$ the marginal distribution of $x$ in the above model; Definition 1 implies that the density of $\mathcal{D}_{\mathrm{obs}}$ is:

$$\mathcal{D}_{\mathrm{obs}}(x) := \frac{\mathcal{N}(S^\star; \langle w^\star, x \rangle, 1) \cdot \mathcal{D}(x)}{\mathbb{E}_{x \sim \mathcal{D}}\left[\mathcal{N}(S^\star; \langle w^\star, x \rangle, 1)\right]} \ . \quad (1)$$

Additionally, the marginal distribution of $y$ given $x$ is $\mathcal{N}(\langle w^\star, x \rangle, 1, S^\star)$, and its density is given by

$$\mathcal{N}\left(y; \langle w^\star, x \rangle, 1, S^\star\right) := \frac{\mathcal{N}\left(y; \langle w^\star, x \rangle, 1\right) \cdot \mathbb{1}\left\{y \in S^\star\right\}}{\mathcal{N}\left(S^\star; \langle w^\star, x \rangle, 1\right)} \ .$$

## 3. Our Results

In this section, we present our main result: an end-to-end algorithm (Algorithm 2) for efficiently estimating the parameter vector $w^\star$ in the truncated linear regression model, which comes with the following guarantees.

**Theorem 3.1.** *Suppose Assumptions 1 to 3 hold with parameters $(\alpha, R, \sigma, \rho)$ and that $S^\star$ is a union of at most $k$ intervals for some known integer $k$. Further assume that $\|\mathbb{E}_{\mathcal{D}}\, x\|_2 \leq \beta$ with $\beta \geq 0$. Let $\varepsilon, \delta \in (0, 1/2)$. There exists an algorithm that, given parameters $(\alpha, \beta, \varepsilon, \delta, R, k)$ and $n$ i.i.d. samples from the truncated linear regression model of Definition 1 for*

$$n = \mathrm{poly}\left(d, R, k, 1/\varepsilon, \log 1/\delta\right)^{\mathrm{poly}(\sigma, \beta, 1/\rho, 1/\alpha)}$$

*runs in time $\mathrm{poly}(n)$ and outputs an estimate $\overline{w}$ satisfying $\|\overline{w} - w^\star\|_2 \leq \varepsilon$ with probability at least $1 - \delta$.*

The proof of Theorem 3.1 is given in Appendix E. Crucially, both the sample complexity and the runtime of our algorithm scale polynomially with the dimension $d$ and the inverse accuracy $1/\varepsilon$. Hence, our algorithm indeed achieves polynomial dependence on the key problem parameters, without relying on an explicit parametric form for the covariate distribution $\mathcal{D}$, but instead requiring only sub-Gaussianity (Assumption 2), along with the very mild Assumption 1 and the information-theoretically necessary Assumption 3. The dependence on the parameters in the exponent is expected when $S^\star$ is unknown, and is similar to that appearing in (Lee et al., 2024), who consider the problem of learning exponential families under unknown truncation. On the other hand, if the truncation set $S^\star$ is known, this exponential dependence can be avoided (see Appendix B.2). Further, the assumption $\|\mathbb{E}_{\mathcal{D}}\, x\| \leq \beta$ is redundant under a non-centered notion of sub-Gaussianity (which controls $\|x\|$ rather than $\|x - \mathbb{E}_{\mathcal{D}}\, x\|$); we state this requirement separately to avoid confusion. Finally, note that in the natural regime where $\beta$ is constant but $R$ (*i.e.*, $\|w^\star\|_2$) is allowed to scale as $O(\sqrt{d})$ (which is the setting also considered in (Daskalakis et al., 2019)), the sample complexity and runtime still remain polynomial in $d$ and $1/\varepsilon$.

**Remark 3.2** (Explicit Exponential Dependencies)**.** The explicit sample complexity in Theorem 3.1 is

$$n = k \cdot (d/\varepsilon)^{\widetilde{O}\left((\sigma + \beta)^6 / (\rho^4 \alpha^2)\right)}$$

As we discussed, in the simpler case where $S^\star$ is known, the sample complexity required to learn $w^\star$ is $\Theta(d/\varepsilon^2)$ (Daskalakis et al., 2019). Our increased sample complexity is due to the fact that we must learn $S^\star$. Here, our algorithm has a dependence on $d$ because, to estimate $w^\star \in \mathbb{R}^d$ to distance $\varepsilon$, our algorithm needs to learn $S^\star$ to accuracy $\mathrm{poly}(\varepsilon/d)$, for which we need $\mathrm{poly}(\varepsilon/d)$ samples. For more details, we refer the reader to the full proof in Appendix E, and specifically Theorems E.1 and E.2.

**Outline of this section.** Our Algorithm 2 operates in two phases: Phase I approximately learns $S^\star$, and Phase II uses this approximation to estimate $w^\star$. Both phases are technically involved and require various tools to analyze, as we mentioned in Section 1, and present in more detail in this section: In Section 3.1, we present an efficient algorithm for PAC learning unions of intervals from positive-only samples. Section 3.2 shows how we use this algorithm to learn $S^\star$. Finally, Section 3.3 describes our approach for learning $w^\star$ given $S^\star$: we define a *generalized* negative log-likelihood objective (which is different from the usual negative log-likelihood), establish its required properties, and show how to optimize it to obtain an estimate for $w^\star$.

### 3.1. Positive PAC Learning Unions of Intervals

Next, we introduce PAC learning from positive samples only (henceforth, positive-only learning) and give an efficient algorithm for learning unions of intervals in this model.

**Positive-Only Learning.** In positive-only learning (Natarajan, 1987), as in classical PAC learning, there is a distribution $\mathcal{D}^\star$ over a feature space $\mathcal{X}$ and some concept class $\mathcal{H}$ containing the target concept $h^\star$. In contrast to standard PAC learning, here we are only given sample access to the conditional distribution $\mathcal{D}^\star_+ := \mathcal{D}^\star|_{h^\star(x)=1}$ of positive samples (a distribution over $\mathcal{X}$) and not given any information about the negative samples. Our goal is the same as PAC learning: to output a hypothesis $\widehat{h}$ with small misclassification error under $\mathcal{D}^\star$, *i.e.*, $\Pr_{x \sim \mathcal{D}^\star}(\widehat{h}(x) \neq h^\star(x)) \leq \varepsilon$. Crucially, we aim for small misclassification error on both positive and negative examples, even though we only have sample access to positives.

**Smoothed Positive-Only Learning.** (Lee et al., 2026) develop a *smoothed* version of positive-only learning, where in addition to positive samples, the learner also has access to a distribution $\mathcal{D}$ over the feature space which is "smooth" with respect to $\mathcal{D}^\star$. Intuitively, samples from $\mathcal{D}$ help the learner identify high-probability regions of the unconditional $\mathcal{D}^\star$. In particular, (Lee et al., 2026) introduce the following notion of generalized smoothness, which extends notions of smoothness studied by works in smoothed online learning, *e.g.*, (Haghtalab et al., 2022; 2020; 2024; Block et al., 2024; Blanchard, 2025).

**Definition 2** (Smoothness; (Lee et al., 2026)). *For any $s \in (0, 1]$ and $q \geq 1$, and any two distributions $\mathcal{D}, \mathcal{D}^\star$ over the same space, we say that $\mathcal{D}$ is $(s, q)$-smooth w.r.t. $\mathcal{D}^\star$ if, for any measurable set $S$:*

$$\mathcal{D}^\star(S) \leq (1/s) \cdot \mathcal{D}(S)^{1/q} \,.$$

To gain some intuition about this definition, we refer the reader to Figure 1 which illustrates this setting. Under this

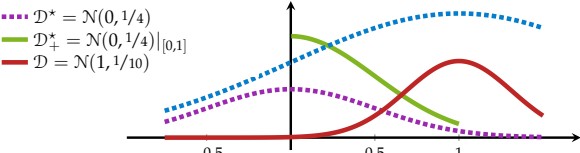

*Figure 1.* Example of positive-only learning under smoothness. Here, we wish to PAC learn the interval $[0, 1]$ under base distribution $\mathcal{D}^\star$. We are given sample access only to $\mathcal{D}^\star_+$, which is the restriction of $\mathcal{D}^\star$ to $[0, 1]$, and to $\mathcal{D}$, which is smooth w.r.t. $\mathcal{D}^\star$. In particular, the dotted blue line shows the function $2\mathcal{D}(x)^{1/10}$, which lies above the density $\mathcal{D}^\star(x)$, hence $\mathcal{D}$ is $(1/2, 10)$-smooth w.r.t. $\mathcal{D}^\star$.

smoothness assumption on $\mathcal{D}$, (Lee et al., 2026) establish the following learning guarantee.

**Theorem 3.3** (Theorem 1.1 of (Lee et al., 2026)). *Suppose that $\mathcal{D}$ is $(s, q)$-smooth w.r.t. $\mathcal{D}^\star$, and fix any $\varepsilon, \delta \in (0, 1/2)$. There exists an algorithm that, given $\varepsilon, \delta$ and $n = \widetilde{O}\left((\varepsilon s)^{-2q} \cdot (\mathrm{VC}(\mathcal{H}) + \log 1/\delta)\right)$ independent samples from $\mathcal{D}^\star_+$ and $\mathcal{D}$ (with $h^\star \in \mathcal{H}$), outputs a hypothesis $\widehat{h}$, such that, with probability at least $1 - \delta$*

$$\Pr_{x \sim \mathcal{D}^\star}(\widehat{h}(x) \neq h^\star(x)) \leq \varepsilon \,.$$

We stress that the above algorithm is only sample-efficient and does not come with any computational-efficiency guarantee. That said, this result is surprising because it ensures high-accuracy w.r.t. the base distribution $\mathcal{D}^\star$, even though we do *not* have sample access to $\mathcal{D}^\star$ itself at training time. This framework thus provides a way to *extrapolate* information about $\mathcal{D}^\star$ from only $\mathcal{D}^\star_+$ and a smooth reference distribution $\mathcal{D}$. The connection to learning $S^\star$ in truncated regression is now apparent: in that setting, we only observe samples in $S^\star$ (*i.e.*, positive samples). We formalize this connection in Section 3.2.

Although Theorem 3.3 is sample-efficient, it is not computationally efficient in general, since it solves an empirical-risk-minimization problem which is, typically, computationally hard. However, when $\mathcal{H}$ consists of unions of at most $k$ intervals in $\mathbb{R}$, we prove that the above becomes computationally tractable. We thus obtain a polynomial-time algorithm for learning unions of $k$ intervals from positive examples provided the distribution $\mathcal{D}^\star$ satisfies the smoothness requirement (Definition 2).

Formally, let $\mathcal{S}_k$ be the collection of all unions of at most $k$ intervals in $\mathbb{R}$, and define the hypothesis class $\mathcal{H}_k := \{\mathbb{1}_S : S \in \mathcal{S}_k\}$. Given sample access to $\mathcal{D}$ and $\mathcal{D}^\star_+$ as defined above, we can learn $h^\star$ in $\mathrm{poly}_{s,q}(k/\varepsilon)$ time:

**Theorem 3.4** (Positive PAC Learning unions of intervals under Smoothness). *Assume that the true concept $h^\star$ belongs to the hypothesis class $\mathcal{H}_k$ and suppose that $\mathcal{D}$ is $(s, q)$-smooth w.r.t. $\mathcal{D}^\star$. There exists an algorithm that, given $\varepsilon, \delta \in (0, 1/2)$ and $n = \widetilde{O}\left((\varepsilon s)^{-2q} \cdot (k + \log 1/\delta)\right)$ inde-*

*pendent samples from $\mathcal{D}_+^\star$ and $\mathcal{D}$, outputs a hypothesis $\widehat{h}$, such that, with probability at least $1 - \delta$*

$$\Pr_{x \sim \mathcal{D}^\star}(\widehat{h}(x) \neq h^\star(x)) \leq \varepsilon.$$

*Moreover, the algorithm runs in time $O(n \log n)$, and its output $\widehat{h}$ is a union of at most $\frac{k-1}{(\varepsilon s)^q} + 1$ intervals.*

Thus, Theorem 3.4 shows that with sample access to $\mathcal{D}_+^\star$ and to a distribution $\mathcal{D}$ which witnesses the smoothness of $\mathcal{D}^\star$, we can efficiently PAC learn any union of $k$ intervals under $\mathcal{D}^\star$. The proof appears in Appendix E.1. This result holds independent of our other results on truncated linear regression and can be of broader interest.

**Overview of Set-Learning Algorithm.** The algorithm of Theorem 3.4 (Algorithm 1) is simple. After drawing $n$ samples from $\mathcal{D}_+^\star$ and $\mathcal{D}$, we consider the $n - 1$ intervals on the real line defined by consecutive samples from $\mathcal{D}_+^\star$. We count how many samples from $\mathcal{D}$ fall into each of these intervals, and discard the $\frac{k-1}{(\varepsilon s)^q}$ intervals containing the most samples. Finally, we return the union of the remaining intervals. Figure 2 illustrates this procedure on one example.

The intuition behind discarding the intervals with the most samples from $\mathcal{D}$ is as follows: if there are many samples from $\mathcal{D}$ between two consecutive samples $y_1 < y_2$ from $\mathcal{P}$, then w.h.p. either $(y_1, y_2)$ lies outside the support of $\mathcal{D}_+^\star$ (*i.e.*, outside of the set we wish to learn) and should be discarded, or $\mathcal{D}_+^\star$ places little mass in that interval (otherwise we would have seen another sample from $\mathcal{D}_+^\star$ inside it), so discarding it incurs little learning error. To make this intuition rigorous, we need a careful analysis of the above greedy process; see Appendix E.1.

### 3.2. Learning the Truncation Set $S^\star$

Having presented our result on positive-only learning of unions of intervals under smoothness, we now return to the truncated regression setting and show how to efficiently learn the truncation set $S^\star$. To learn $S^\star$ (which is a union of $k$ intervals), we use Algorithm 1 from Theorem 3.4. The challenge in this setting is that we do not have access to an auxiliary distribution $\mathcal{D}$ a priori (in order to apply the smoothness framework from the previous section). To overcome this, we show how to construct this distribution in the truncated regression problem.

Concretely, we must define the base distribution $\mathcal{D}^\star$ under which we wish to learn $S^\star$, and show how to obtain sample access to $\mathcal{D}_+^\star = \mathcal{D}^\star|_{S^\star}$ and to some distribution $\mathcal{D}$ smooth w.r.t. $\mathcal{D}^\star$. The natural base distribution $\mathcal{D}^\star$ is the *untruncated* distribution over $\mathbb{R}$ of the response $y$. Formally, for any $w \in \mathbb{R}^d$, consider the distribution over $(x, y) \in \mathbb{R}^d \times \mathbb{R}$ where the marginal of $x$ is $\mathcal{D}_{\mathrm{obs}}$ (see Equation (1)) and the conditional distribution of $y \mid x$ is the untruncated Gaussian $\mathcal{N}(\langle w, x \rangle, 1)$. Denote by $\mathcal{D}_y(w)$ the (unconditional)

marginal distribution of $y$. Our goal is to approximately learn $S^\star$ under $\mathcal{D}_y(w^\star)$, *i.e.* to find a set $S \subseteq \mathbb{R}$ such that

$$\mathcal{D}_y(w; S^\star \triangle S) := \Pr_{y \sim \mathcal{D}_y(w^\star)}(y \in S^\star \triangle S) \leq \varepsilon. \quad (2)$$

Here, $A \triangle B := (A \backslash B) \cup (B \backslash A)$ denotes the set symmetric difference. To learn such a set $S$, we apply Theorem 3.4. This requires access to positive samples from $\mathcal{D}_y(w^\star)$, as well as samples from a distribution smooth w.r.t. $\mathcal{D}_y(w^\star)$. We can obtain positive samples directly from our truncated regression model, however it is not clear how to construct the smooth distribution. If we knew $w^\star$, then we could obtain a sample $x \sim \mathcal{D}_{\mathrm{obs}}$ from the truncated regression model and then generate (the untruncated) $y \sim \mathcal{N}(\langle w^\star, x \rangle, 1)$ to sample exactly from $\mathcal{D}_y(w^\star)$, but $w^\star$ is of course unknown.

**Key Lemma: Sub-Gaussianity Implies Smoothness.** To obtain the smooth distribution, we rely on the observation that, under our sub-Gaussianity assumption for $x$, for any parameter vector $w$ within constant distance of $w^\star$, the distribution $\mathcal{D}_y(w)$ is smooth w.r.t. $\mathcal{D}_y(w^\star)$.

**Lemma 3.5** (Smoothness). *Suppose that Assumptions 1 and 2 hold. Let $w \in \mathbb{R}^d$ be such that $\|w - w^\star\|_2 \leq D$, and assume that $\|\mathbb{E}_\mathcal{D} x\|_2 \leq \beta$ for $\beta > 0$. For any measurable $T \subseteq \mathbb{R}$ such that $\mathbb{E}_{x \sim \mathcal{D}_{\mathrm{obs}}}[\mathcal{N}(T; \langle w^\star, x \rangle, 1] > 0$:*

$$\mathcal{D}_y(w^\star; T) \leq (1/s) \cdot [\mathcal{D}_y(w; T)]^{1/q}$$

*for constants $s = \Theta(1)$ and $q = \Theta\left(D^2\left(\sigma^2 \log 1/\alpha + \beta^2\right)\right)$.*

The proof of Lemma 3.5 appears in Appendix E.1.2; as we discuss there, the proof crucially relies on (i) upper and lower bounds on the tails of the (Gaussian) noise, and (ii) sub-Gaussianity of the covariate distribution $\mathcal{D}$. Observe that the smoothness parameter $q$ scales with $D$, the distance between $w$ and $w^\star$, affecting the sample and time complexity of set learning. However, as we show in Section 3.3.1, we can efficiently find an initial parameter vector $\widehat{w}$ with $\|\widehat{w} - w^\star\|_2 \leq D = O(1)$. Thus, we can use $\mathcal{D}_y(\widehat{w})$ (from which we can sample efficiently by drawing $x \sim \mathcal{D}_{\mathrm{obs}}$ and (the untruncated) $y \sim \mathcal{N}(\langle \widehat{w}, x \rangle, 1)$ conditionally on $x$ as the smooth distribution $\mathcal{D}$ in Algorithm 1, yielding an efficient procedure for approximately learning $S^\star$.

### 3.3. Learning $w^\star$ via PSGD

We now turn to Phase II of our algorithm: Estimating the parameter vector $w^\star$ via PSGD on an appropriate objective function, given a sufficiently accurate approximation $S \approx S^\star$ (from previous sections). Let $\gamma_{x,w,S}(y)$ be the density of $\mathcal{N}(\langle w, x \rangle, 1, S)$ at $y$. If we knew the survival set $S^\star$, we could use the following (pseudo) negative log-likelihood (NLL) as our objective:

$$\mathscr{L}(w) := -\mathbb{E}_{x \sim \mathcal{D}_{\mathrm{obs}}}\left[\mathbb{E}_{y \sim \mathcal{N}(\langle w^\star, x \rangle, 1, S^\star)}\left[\log \gamma_{x,w,S^\star}(y) \mid x\right]\right]$$

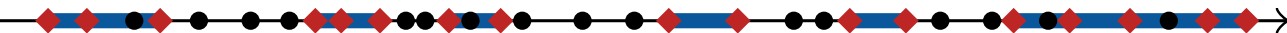

*Figure 2.* Example of Algorithm 1 in action, for learning a union of $k = 2$ intervals with error $\varepsilon = 0.2$. The points above represent samples on the real line. The red diamonds are positive samples drawn from the distribution $\mathcal{D}_+^\star$, while the black dots are samples from the distribution $\mathcal{D}$ which is smooth w.r.t. the target $\mathcal{D}^\star$. We assume that the smoothness parameters are $s = q = 1$ for simplicity. The algorithm considers the intervals defined by consecutive red points, and discards the $\frac{k-1}{(\varepsilon s)^q} = 5$ intervals with the most black points. The final output is the union of the intervals denoted in blue.

Here, the outer expectation is over the observed distribution $\mathcal{D}_{\mathrm{obs}}$ (see Equation (1)) of the features (not the base distribution $\mathcal{D}$), while the inner expectation is over the true $w^\star$, but the loss function uses the "guess" $w$, which we are optimizing. Since we do not have membership access to $S^\star$ but only to the set $S$, we instead consider the following "perturbed" NLL, following Definition 3 of Lee et al. (2024): given a set $S$, define the perturbed pseudo-NLL $\mathscr{L}_S$ as

$$\mathscr{L}_S(w) := -\mathop{\mathbb{E}}_{x \sim \mathcal{D}_{\mathrm{obs}}} \left[ \mathop{\mathbb{E}}_{y \sim \mathcal{N}(\langle w^\star, x \rangle, 1, S \cap S^\star)} \left[ \log \gamma_{x,w,S}(y) \mid x \right] \right]$$

When $S = S^\star$, the perturbed NLL is equal to the negative log-likelihood. However, in general, it is *not* and may lack the desirable properties of the negative log-likelihood.

In Appendix E.2, we prove that $\mathscr{L}_S$ has the desired properties. While we defer the details to Appendix E, we highlight some key steps here. An important issue with the perturbed log-likelihood $\mathscr{L}_S$ is that its global minimizer is not $w^\star$, so there is no a priori guarantee that optimizing $\mathscr{L}_S$ will recover $w^\star$. Moreover, even if $w^\star$ were the unique minimizer, unless $\mathscr{L}_S$ is also *strongly* convex around $w^\star$, optimizing $\mathscr{L}_S$ in function value need not yield convergence to $w^\star$. To resolve both issues, we do not optimize $\mathscr{L}_S$ globally, but instead restrict optimization to an appropriately defined *projection set* $\mathcal{P}$ such that

R1. $w^\star \in \mathcal{P}$;

R2. the minimizer of $\mathscr{L}_S$ in $\mathcal{P}$ is close to $w^\star$; and

R3. $\mathscr{L}_S$ is strongly convex over $\mathcal{P}$.

### 3.3.1. FINDING A WARM START AND DEFINING THE PROJECTION SET

To define our projection set, we first show how to efficiently find a *warm start*: a point $\widehat{w}$ within constant distance of $w^\star$ (recall that we also require this to learn $S^\star$ in Section 3.2). We draw $n$ i.i.d. samples $\left\{ \left( x^{(i)}, y^{(i)} \right) \right\}_{i \in [n]}$ from our truncated regression model, and define $\widehat{w}$ as the output of ordinary least-squares (OLS), ignoring truncation:

$$\widehat{w} := \left( \tfrac{1}{n} \sum_i x^{(i)} x^{(i)\top} \right)^{-1} \cdot \left( \tfrac{1}{n} \sum_i y^{(i)} \cdot x^{(i)} \right) .$$

Even though the above OLS solution does not account for truncation, sub-Gaussian concentration of $x$ implies that it is not too far from $w^\star$. In fact, for sufficiently large $n$,

w.h.p. $\|\widehat{w} - w^\star\|_2 \leq O(1)$.[7] This argument is formalized in Lemma E.8, whose proof appears in Appendix E.2.3.

Given $\widehat{w}$, we define our projection set as a constant-radius ball around it, *i.e.*, $\mathcal{P} := B(\widehat{w}, \Theta(1))$. This ensures that $w^\star \in \mathcal{P}$ w.h.p. – satisfying requirement R1. To ensure strong convexity of $\mathscr{L}_S$ in $\mathcal{P}$ (*i.e.*, requirement R3), since $\mathscr{L}_S$ is defined using our approximation $S$ of $S^\star$, we need the set learning error $\varepsilon$ to be sufficiently small. Choosing $\varepsilon$ small enough ensures that $\nabla^2 \mathscr{L}_S(w) \succeq \Omega(1) \cdot I$ for all $w \in \mathcal{P}$, yielding strong convexity. This is established in Corollary E.9, proved in Appendix E.2.3.

Finally, for requirement R2, we show in Appendix E.2.4 that the gradient norm satisfies $\|\nabla \mathscr{L}_S(w^\star)\|_2 \leq \widetilde{O}(d^{1/2}\varepsilon^{1/6})$ for $\varepsilon$ small enough (Lemma E.10). This, combined with strong convexity and $w^\star \in \mathcal{P}$, implies that the minimizer of $\mathscr{L}_S$ over $\mathcal{P}$ is at most a distance $\widetilde{O}(d^{1/2}\varepsilon^{1/6})$ away from $w^\star$ (Lemma E.11). Thus, we can make it arbitrarily close to $w^\star$ by choosing $\varepsilon$.

### 3.3.2. ALGORITHMIC CONSIDERATIONS FOR PSGD

The above results yield a projection set $\mathcal{P}$ such that approximately minimizing $\mathscr{L}_S$ over $\mathcal{P}$ produces an estimate of $w^\star$. To actually run PSGD and obtain this approximate minimizer, we need to satisfy some algorithmic considerations. Namely, we must (C1) find an initial point in $\mathcal{P}$ for the first PSGD iteration, (C2) efficiently project onto $\mathcal{P}$ at each step, and (C3) obtain an unbiased gradient estimate of $\nabla \mathscr{L}_S$ at each iteration. For (C1), we use our warm start $\widehat{w}$ from before. Ensuring (C2) is straightforward since $\mathcal{P}$ is a ball with known center and radius. Satisfying (C3), which requires implementing an unbiased gradient estimator for $\mathscr{L}_S$, is challenging and we discuss it further in Appendix E.2.5.

To circumvent this challenge, we allow our gradient estimate to be *biased*: in particular, given a point $w \in \mathbb{R}^d$, we implement a subroutine that returns $g(w) \in \mathbb{R}^d$ satisfying $\mathbb{E}\left[ g(w) \mid w \right] = \nabla \mathscr{L}_S(w) + b$, where $b$ is a bias vector satisfying $\|b\| \leq \mathrm{poly}(d\varepsilon)$. In Appendix E.2.5, we show how to implement this biased sampler. We bound the runtime in Corollary E.14, while in Lemma E.15 we show that the bias satisfies $\|b\|_2 \leq \widetilde{O}(d^{1/2}\varepsilon^{1/4})$, so we can

---

[7]Throughout this section, we hide any constants from Assumptions 1 to 3 in the big-O notation, to simplify the overview and highlight dependencies on the dimension $d$ and the set learning error $\varepsilon$.

control it by appropriately setting $\varepsilon$. We also prove an upper bound on the second moment of the (biased) stochastic gradients (Lemma E.16), necessary for the convergence of PSGD. Finally, we combine these results with an analysis of PSGD with biased gradients to show that running PSGD with projection set $\mathcal{P}$ and our biased gradient sampler yields an estimate of $w^\star$, as desired.

## 4. Simulations with Synthetic Data

We empirically evaluate the rate-of-convergence of our algorithm on synthetic data.

**Setup.** The feature distribution is a 10-dimensional mixture of five Gaussians, and the truncation set is a union of five intervals.

**Baselines.** Our baseline is Ordinary Least Squares (OLS) (which does not account for any truncation). We compare the performance of three algorithms:

- Projected Stochastic Gradient Descent (PSGD) on the MLE objective with misspecified truncation set $S$ (which is the algorithm by Daskalakis et al. (2019) and accounts for truncation but is provided an incorrect set $S \neq S^\star$);
- PSGD on the MLE objective with the correct truncation set $S^\star$ (which is the same as the previous algorithm except that it is provided knowledge of $S^\star$);
- Our algorithm (Algorithm 5).

We note PSGD with $S^\star$ is an idealized algorithm that knows $S^\star$ a priori; we cannot actually implement it in practice since $S^\star$ is unknown, and we only include it for comparison.

**Results.** The results of our experiments are shown in Figure 3. As expected, we observe that blindly applying OLS without accounting for truncation can give significantly biased estimates, and the same is true if we attempt to run PSGD on the MLE objective with an incorrect truncation set. On the other hand, our algorithm converges to the true $w^\star$, at essentially the same rate as the ideal algorithm which uses PSGD on the true MLE objective and assumes the exact knowledge of the truncation set $S^\star$ (which is unavailable in practice).

## 5. Conclusion

We provide the first polynomial-time algorithm for truncated linear regression with an *unknown* survival set (Theorem 3.1). Our algorithm (Algorithm 2) operates under the mild assumption that the covariate vectors are sub-Gaussian, and has polynomial dependence on all key problem parameters. An important ingredient in establishing this result is an efficient procedure for learning unions of intervals from *positive examples* under an appropriate smoothness condition

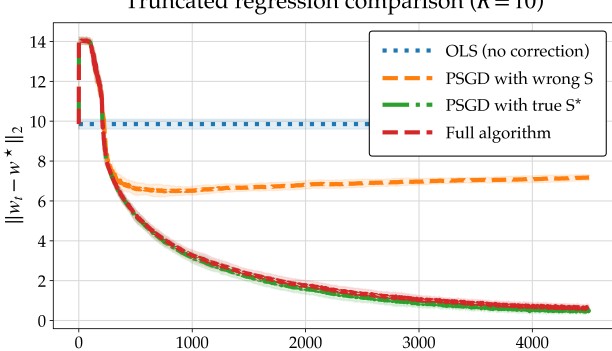

*Figure 3.* **Simulation Results.** The plot shows the Euclidean distance of the PSGD iterates $w_t$ to the true parameter vector $w^\star$ (note that OLS is not an iterative algorithm, so its error is simply plotted as a constant line). The experiment is repeated 10 times, and the shaded regions around each line show standard deviation around the mean Euclidean distance. Our algorithm ("Full Algorithm") converges to $w^\star$ at almost the same rate as the ideal algorithm (PSGD with true $S^\star$) which has exact knowledge of the truncation set $S^\star$ and, hence, is not implementable in practice.

(Theorem 3.4), which may be of independent interest.

Our results suggest several directions for future work. First, truncated regression beyond the linear setting, such as Poisson and logistic regression, have been studied with known survival set (Ilyas et al., 2020) and extending our techniques to *unknown* survival set variants of these is a natural next step. Further, we expect that natural extensions of our interval-learning algorithm can learn richer hypothesis classes in constant dimensions ($d = O(1)$) with just positive examples under smoothness, provided they are well-approximated by unions of axis-aligned hypercubes. Some such classes are studied in Durvasula et al. (2023) and even shown to include classes with infinite VC dimension.

## Impact Statement

This paper presents work whose goal is to advance the field of Machine Learning. There are many potential societal consequences of our work, none which we feel must be specifically highlighted here.

## Acknowledgements

Alexandros Kouridakis was supported by a scholarship from the Onassis Foundation. Alkis Kalavasis was supported by the Institute for Foundations of Data Science at Yale.

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

# Supplementary Material For
# "Linear Regression with Unknown Truncation Beyond Gaussian Features"

## Contents

## A. Complex Survival Sets in Astronomical Surveys

In this section, we illustrate that survival sets arising in real-world data can be highly nontrivial: rather than being simple threshold rules, they are often induced by a layered sequence of instrument constraints, engineering decisions, and scientific priorities, many of which interact in ways that are difficult to formalize.

To make this concrete, we focus on astronomy, where modern *large-scale surveys* have become foundational infrastructure for astrophysics. These surveys, such as the Sloan Digital Sky Survey (SDSS), Gaia, and the Vera Rubin Observatory's Legacy Survey of Space and Time (LSST), systematically measure properties of millions to billions of objects, enabling population-level studies of stars and galaxies, precision tests of cosmological models, and the discovery of rare phenomena (York et al., 2000; Prusti et al., 2016; Ivezić et al., 2019). At the same time, the scale and complexity that make these surveys powerful also make selection effects unavoidable, and careful statistical analysis requires understanding (or approximating) the induced survival set.

**Warm-up.**  We begin with a classical and widely studied selection effect that arises from physical limits of instruments.

> **Example A.1** (Malmquist Bias).  Malmquist bias is a selection effect in astronomical surveys driven by flux limits: any instrument can only observe objects whose *apparent* brightness $y$ exceeds a detection threshold, while dimmer objects remain unobserved (Malmquist, 1922; 1925). This truncation can systematically bias downstream inferences if it is ignored, leading to incorrect conclusions about stellar populations and distances (Teerikorpi, 1997). For instance, among objects at a fixed true distance, the brightness cutoff removes faint members; consequently, the observed sample over-represents intrinsically brighter objects, unless the selection mechanism is explicitly accounted for (as we do in our algorithm).

**Selection effects in large-scale survey pipelines.**  While Malmquist bias is already a nontrivial form of truncation, real survey pipelines typically induce much richer survival sets.

> **Example A.2** (Truncation Biases in Large-Scale Astronomical Surveys).  Large-scale astronomical surveys form the backbone of modern astrophysics: they enable scientists to identify promising regions to study, evaluate foundational theoretical models, and discover rare and new phenomena (*e.g.*, (York et al., 2000; Prusti et al., 2016; Ivezić et al., 2019)). However, their massive scale and operational complexity inevitably introduce truncation biases that must be corrected to draw valid scientific conclusions.
>
> A representative case study is the Apache Point Observatory Galactic Evolution Experiment (APOGEE) (Majewski et al., 2017), one of the most comprehensive stellar spectroscopic surveys conducted to date. APOGEE collected high-resolution near-infrared spectra for over 146,000 red giant stars using custom-built spectrographs. The survey's scale—spanning multiple years, numerous institutions, and extensive operational logistics—illustrates why truncation effects are inevitable: each stage of the pipeline introduces constraints and trade-offs that shape which objects are ultimately observed and how.
>
> Survey pipelines introduce numerous truncation biases, both known and unknown. Some arise from fundamental physical limitations (such as Malmquist bias), while others emerge from practical engineering constraints and resource allocation decisions. The question is therefore not whether truncation biases exist, but how to account for their combined effects on the observed sample. Even within a single survey like APOGEE, truncation effects arise at multiple levels (Majewski et al., 2017). To illustrate their diversity and the resulting complexity of the induced survival set, we group some common mechanisms by their underlying causes (see (SDSS-IV Collaboration, 2022a) for further discussion):
>
> 1. *Sampling-dependent biases* arise from target truncation strategies. Surveys often partition targets into cohorts with different observation priorities, leading to truncation of some cohorts. For example, APOGEE stratifies targets by apparent brightness (H-band magnitude) and preferentially targets certain strata. Importantly, cohort definitions can depend on modeling choices: APOGEE preferentially sampled red giants over dwarfs using a "redness" criterion, but this criterion depends on the chosen de-reddening model used to convert observed color

$C_{\mathrm{obs}}$ into intrinsic color $C_{\mathrm{corr}}$, making truncation model-dependent (SDSS-IV Collaboration, 2022b).

2. *Physical limitations* of instruments impose unavoidable constraints. Beyond flux limits, atmospheric absorption can prevent observation of certain objects, and instrument design can induce truncation over sky position. For instance, APOGEE uses multi-fiber spectrographs that route light from many targets through optical fibers to a separate spectrograph for high-precision analysis (Wilson et al., 2019; O'Connell, 2020). Physical constraints create an "exclusion zone" around each fiber that prevents placing another fiber nearby, which systematically under-samples dense regions such as star clusters and the inner Galaxy (SDSS-IV Collaboration, 2022b), truncating some complex set of outcomes $y$.

3. *Cascading effects* propagate biases from upstream catalogs to downstream surveys. Many surveys begin with a "parent catalog" and inherit its selection criteria. For example, galaxy surveys that start from the Uppsala General Catalogue inherit criteria that omit small angular-size galaxies unless they exceed brightness thresholds (Freudling et al., 1995). These inherited filters, composed with survey-specific constraints, yield selection functions that are effectively complex convolutions of multiple stages.

The consequences of ignoring truncation effects can be severe. A particularly striking example comes from NASA's *Kepler* mission: an incorrectly implemented veto in pipeline version 9.2 created an unexpected period-dependent detection bias, substantially reducing the detectability of planets with orbital periods exceeding 40 days relative to theoretical expectations (Christiansen et al., 2016). The flaw went undetected until extensive validation efforts, and its discovery necessitated reprocessing the catalog and issuing cautionary guidance for analyses performed on earlier releases (Christiansen et al., 2016).

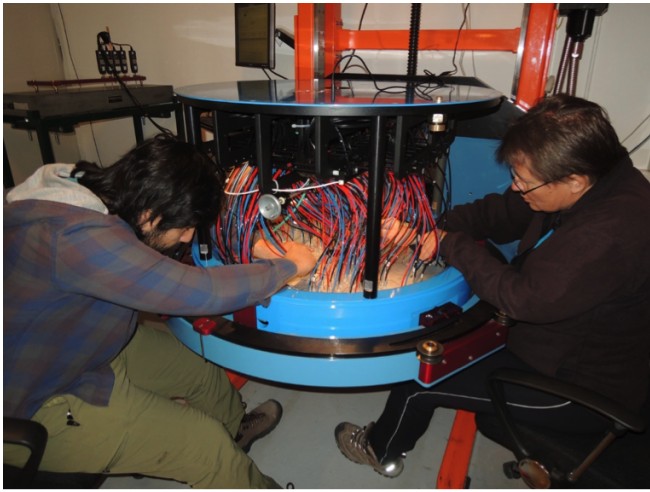

*Figure 4.* Illustrations for Example A.2. A multi-fiber spectroscope from the APOGEE survey, where the requirement of a minimum distance between fibers creates spatially dependent selection effects. Image credit: (SDSS Collaboration, 2022).

## B. Further Consequences of Main Result (Theorem 3.1)

In this section, we present some additional consequences of Theorem 3.1 in special cases where the covariates are Gaussian (Appendix B.1) and when the survival set $S^\star$ is known (Appendix B.2).

### B.1. The Gaussian Case

When the marginal distribution $\mathcal{D}$ of the features is a Gaussian distribution over $\mathbb{R}^d$, Theorem 3.1 takes the following form.

**Corollary B.1.** *Suppose that $\mathcal{D} = \mathcal{N}(\mu, \Sigma)$ for some $\mu \in \mathbb{R}^d$ and $\Sigma \in \mathbb{R}^{d \times d}_{\succeq 0}$, such that $\|\mu\|_2 \leq \beta$ and $\|\Sigma\|_{\mathrm{op}} \leq \lambda$ for constants $\beta, \lambda$. Assume that $\mathcal{D}$ satisfies Assumptions 1 and 3, and that $S^\star$ is a union of at most $k$ intervals for some known integer $k$. Further assume that $\|w^\star\|_2 \leq R$ for constant $R \geq 0$. Let $\varepsilon, \delta \in (0, 1/2)$. There exists an algorithm that, given $n$*

*i.i.d. samples from the truncated regression model for*

$$n = \text{poly}\left(d, R, k, \frac{1}{\varepsilon}, \log\frac{1}{\delta}\right)^{\text{poly}(\lambda, \beta, 1/\rho, 1/\alpha)}$$

*runs in time* $\text{poly}(n)$ *and outputs* $\overline{w} \in \mathbb{R}^d$ *satisfying* $\|\overline{w} - w^\star\|_2 \leq \varepsilon$ *with probability at least* $1 - \delta$.

Ignoring the constants defined in the assumptions, and focusing on the dimension $d$ and the inverse accuracy $1/\varepsilon$, the above corollary gives a $\text{poly}(d, 1/\varepsilon)$ sample complexity for truncated linear regression with Gaussian marginals and unknown truncation. In this Gaussian case, there is an algorithm for truncated linear regression with unknown survival set, by Lee et al. (2024), and we improve their running time from $d^{\text{poly}(k/\varepsilon)}$ to $\text{poly}(dk/\varepsilon)$. We note that Lee et al. (2024) algorithm extends beyond the truncated regression problem we study; see Remark B.2 for details.

*Proof of Corollary B.1.* The result is immediate from Theorem 3.1, upon observing that $\mathcal{D} = \mathcal{N}(\mu, \Sigma)$ is a $\sigma$-sub-Gaussian distribution for $\sigma \lesssim \lambda$, so $\mathcal{D}$ satisfies Assumption 2 as well. $\qquad\square$

**Remark B.2** (Generality of (Lee et al., 2024)'s algorithm). We note that (Lee et al., 2024)'s algorithm extends to a different variant of the truncated linear regression problem, which we do not study in this work: Namely, where samples $(x, y) \in \mathbb{R}^d \times \mathbb{R}$ drawn from the (untruncated) regression model and only shown when $(x, y)$ falls into a high-dimensional survival set $S^\star \subseteq \mathbb{R}^d \times \mathbb{R}$. This variant is somewhat incompatible with the one studied in this work because of two reasons:

1. On the one hand, they consider truncation jointly in $x$ and $y$, which is more general than the model we study, where truncation happens on $y$ independent of $x$;

2. On the other hand, they require the distribution of $x$ to be Gaussian, where as we do not require any such parameteric restriction.

That said, since we allow $\mathcal{D}$ to be non-Gaussian, we can capture an important special case of their model where the (high-dimensional) survival set is of the form $S_X^\star \times S_Y^\star$ for $S_X^\star \subseteq \mathbb{R}^d$ and $S_Y^\star \subseteq \mathbb{R}$. Indeed, in this case, we can simply set $S^\star$ (in our model) to be $S_Y^\star$ and let $\mathcal{D}$ (in our model) be the truncation of $\mathcal{N}(\mu, \Sigma)$ to the set $S_X^\star$. Importantly, the complexity of Lee et al. (2024)'s algorithm is not improved in this special case, as they still require $d^{\text{poly}(k/\varepsilon)}$ samples and runtime to learn the survival set $S_X^\star \times S_Y^\star$. Hence, in the regime where truncation happens on $y$ independent of $x$, our algorithm is the first that provably learns $w^\star$ in $\text{poly}(dk/\varepsilon)$ time, and further does not require parametric assumptions on the feature distribution.

### B.2. Known Survival Set

If the survival set $S^\star$ is actually known in advance, then we can skip the first phase of our learning algorithm (*i.e.* approximately learning $S^\star$) and directly implement the second phase (*i.e.* PSGD on the log-likelihood function) to estimate $w^\star$. Since set estimation is the more computationally demanding part of our algorithm, in this case, we obtain significantly improved sample complexity and running time:

**Corollary B.3.** *Suppose that Assumptions 1 to 3 hold, and also that the truncation set* $S^\star$ *is known.*[8] *Further, assume that* $\|\mathbb{E}_{\mathcal{D}} x\|_2 \leq \beta$ *and* $\|w^\star\|_2 \leq R$ *for some constants* $\beta, R \geq 1$. *Let* $\varepsilon, \delta \in (0, 1/2)$. *There exists an algorithm that, given* $n$ *samples from the truncated regression model for*

$$n = e^{\text{poly}(\sigma, \beta, 1/\rho, 1/\alpha)} \cdot \widetilde{O}\left(R^2 \cdot \frac{d}{\varepsilon^2} \cdot \log\frac{1}{\delta}\right)$$

*runs in time* $\text{poly}(n)$ *and outputs a* $\overline{w} \in \mathbb{R}^d$ *satisfying* $\|\overline{w} - w^\star\|_2 \leq \varepsilon$ *with probability at least* $1 - \delta$.

Importantly, when the truncation set is known, the sample complexity scales as $\widetilde{O}(d/\varepsilon^2)$, which matches the best possible sample complexity even without truncation and qualitatively recovers the result of (Daskalakis et al., 2019).

---

[8]Specifically, if $S^\star$ is a union of $k$ intervals, we require knowledge of all $2k$ endpoints defining it.

## C. Complete Descriptions of Algorithms

In this section, we give the full descriptions of the algorithms presented in the main body. Algorithm 1, which was discussed in Section 3.1, learns a union of $k$ intervals $S^\star \subseteq \mathbb{R}$ under a base distribution $\mathcal{D}^\star$, given sample access to the conditional distribution $\mathcal{D}_+^\star := \mathcal{D}|_{S^\star}$ and to a distribution $\mathcal{D}$ which is smooth w.r.t. $\mathcal{D}^\star$. Algorithm 2 is our end-to-end algorithm for learning the parameter vector $w^\star$ in our truncated linear regression model, which, as we discussed in Section 3, first finds an approximation $S$ for the truncation set $S^\star$ by calling Algorithm 1, and then runs PSGD to estimate $w^\star$.

---

**Algorithm 1** Learning Unions of Intervals under Smoothness

---

**Require:** Smoothness parameters $s, q$, number of intervals $k$, accuracy $\varepsilon$, confidence $\delta$
**Ensure:** Sample access to $\mathcal{D}_+^\star$ and $\mathcal{D}$

1: Set $n := \widetilde{O}\left((\varepsilon s)^{-2q} \cdot (k + \log 1/\delta)\right)$ and $r := \frac{k-1}{(\varepsilon s)^q}$
2: Draw $n$ samples $y^{(1)} \leq \ldots \leq y^{(n)}$ from $\mathcal{D}_+^\star$ and sort them
3: Draw $n$ samples from $\mathcal{D}$ and sort them; let $U$ be the set of those samples
4: For $i \in [n]$, compute $u_i := \left|U \cap (y^{(i)}, y^{(i+1)})\right|$ with a single pass through the data
5: Find the $r$ largest values $u_i$ by sorting; let them be $u_{j_1}, \ldots, u_{j_r}$

6: **Return** $\left[y^{(1)}, y^{(n)}\right] \setminus \left(\bigcup_{\ell=1}^r \left(y^{(j_\ell)}, y^{(j_\ell+1)}\right)\right)$

---

**Algorithm 2** Learning $w^\star$ with unknown truncation

---

**Require:** Constants $\alpha, \rho, \sigma, k, \beta, R$ (see Theorem 3.1), accuracy $\zeta$, confidence $\delta$
**Ensure:** Sample access to the truncated regression model with parameter $w^\star$

▷ *Efficiently find warm-start (Corollary E.9)*
1: Set $n := \widetilde{O}\left(\frac{(\sigma+\beta)^8}{\rho^8 \alpha^2} \cdot R^2 \cdot d \cdot \log \frac{1}{\delta}\right)$
2: Draw $n$ samples $\left(x^{(1)}, y^{(1)}\right), \ldots, \left(x^{(n)}, y^{(n)}\right)$ from the truncated regression model
3: Compute $\widehat{w} := \left(\frac{1}{n}\sum_i x^{(i)} x^{(i)\top}\right)^{-1} \cdot \left(\frac{1}{n}\sum_i y^{(i)} \cdot x^{(i)}\right)$
4: Define the projection set $\mathcal{P} := B\left(\widehat{w}, O\left(\frac{\sigma+\beta}{\rho^2 \alpha}\right)\right)$

▷ *Phase I: Approximately learn truncation set (Theorem E.1)*
5: Set $s := O(1)$, $q := \widetilde{O}\left(\frac{(\sigma+\beta)^4}{\rho^4 \alpha^2}\right)$ and $\varepsilon := \exp\left(-\widetilde{O}\left(\frac{(\sigma+\beta)^8}{\rho^6 \alpha^3}\right)\right) \cdot \widetilde{O}\left(\frac{\zeta^8}{d^3 R^6}\right)$
6: Let $\mathcal{O}_{\mathcal{D}_+^\star}$ be an oracle that draws a sample $(x,y)$ from the truncated regression model and returns $y$
7: Let $\mathcal{O}_{\mathcal{D}}$ be an oracle that draws a sample $(x,y)$ from the truncated regression model and returns $z \sim \mathcal{N}(\langle \widehat{w}, x \rangle, 1)$; this oracle can be implemented efficiently
8: Call Algorithm 1 with parameters $s, q, k, \varepsilon, \delta/3$ and oracles $\mathcal{O}_{\mathcal{D}_+^\star}, \mathcal{O}_{\mathcal{D}}$
9: Let $\widetilde{S}$ be the output of the call
10: Set $L := \widetilde{O}\left(R \cdot \frac{(\sigma+\beta)^9}{\rho^6 \alpha^3} \cdot \log \frac{1}{\zeta} \cdot \log \frac{1}{\delta}\right)$
11: Set $S := \widetilde{S} \cap [-L, L]$

▷ *Phase II: Repeatedly run PSGD and aggregate (Theorem E.2)*
12: Set $K := O(\log 1/\delta)$ and $T := \exp\left(\widetilde{O}\left(\frac{(\sigma+\beta)^8}{\rho^6 \alpha^3}\right)\right) \cdot \widetilde{O}\left(\frac{R^2}{\zeta^2} \cdot d\right)$
13: **for** $i \in [K]$ **do**
14:     Call Algorithm 5 with projection set $\mathcal{P}$, initialization $\widehat{w}$, set $S$ and parameters $T, \varepsilon$
15:     Let $\overline{w}_i$ be the output of this call
16: **end for**
17: Find some $\ell \in [K]$ such that $\|\overline{w}_\ell - \overline{w}_i\|_2 \leq 2\zeta/3$ for at least $3/5$ of the points $\overline{w}_1, \ldots, \overline{w}_K$

18: **Return** $\overline{w}_\ell$

---

## D. Additional Preliminaries

### D.1. Useful Lemmas from Previous Works

We begin by citing a few helpful lemmas from previous works. The following lemma upper bounds the distance between the mean of an untruncated Gaussian distribution and the mean of the same distribution after it is truncated to some set $S$.

**Lemma D.1** (Lemma 6 of (Daskalakis et al., 2018)). *Let $\mu_S$ be the mean of a one-dimensional truncated Gaussian $\mathcal{N}(\mu, 1, S)$, and let $a := \mathcal{N}(S; \mu, 1)$ be the mass that the untruncated Gaussian $\mathcal{N}(\mu, 1)$ places on $S$. Then $|\mu - \mu_S| \leq \sqrt{2 \log 1/a} + 1$.*

Next, we will make use of the following lemma, which lower bounds the variance of a one-dimensional truncated Gaussian.

**Lemma D.2** (Lemma 9 of (Daskalakis et al., 2019)). *Consider a one-dimensional truncated Gaussian $\mathcal{N}(\mu, 1, S)$, and let $a := \mathcal{N}(S; \mu, 1)$ be the mass that the untruncated Gaussian $\mathcal{N}(\mu, 1)$ places on $S$. Then $\mathrm{Var}_{z \sim \mathcal{N}(\mu, 1, S)}[z] \geq a^2/12$.*

Finally, the following lemma is useful for bounding the change of the survival probability (see Definition 3) under shifts on the parameter vector $w$ or the feature vector $x$.

**Lemma D.3** (Lemma 2 of (Daskalakis et al., 2019)). *For any $x, x', w, w' \in \mathbb{R}^d$ and any $S \subseteq R$, it holds that*

$$p(x, w, S) \geq p(x, w', S)^2 \cdot e^{-\langle w - w', x \rangle^2 - 2} \qquad \text{and} \qquad p(x, w, S) \geq p(x', w, S)^2 \cdot e^{-\langle w, x - x' \rangle^2 - 2} .$$

### D.2. PAC Learning under Smoothness Algorithm of (Lee et al., 2026)

Given an arbitrary hypothesis class $\mathcal{H}$, the following algorithm PAC learns a target hypothesis $h^\star \in \mathcal{H}$ under a base distribution $\mathcal{D}^\star$, given sample access to the conditional distribution $\mathcal{D}^\star_+ := \mathcal{D}|_{h^\star(x)=1}$ and to a distribution $\mathcal{D}$ which is smooth w.r.t. $\mathcal{D}^\star$. While the sample complexity of the algorithm scales polynomially with the inverse accuracy $1/\varepsilon$ and the VC dimension of $\mathcal{H}$, to actually implement the algorithm, we must be able to solve a structural ERM problem at each of its iterations, and this is computationally intractable in general.

---

**Algorithm 3** Iterative Pessimistic ERM (Algorithm 1 of (Lee et al., 2026))

---

**Require:** Parameters $\varepsilon, \delta, s, q$, hypothesis class $\mathcal{H}$
**Ensure:** Sample access to distributions $\mathcal{D}^\star_+$ and $\mathcal{D}$

1: Set $n := \widetilde{O}\left((\varepsilon s)^{-2q} \cdot (\mathrm{VC}(\mathcal{H}) + \log 1/\delta)\right)$
2: Create sets $P \subseteq \mathcal{X}$ and $U \subseteq \mathcal{X}$ by drawing $n$ i.i.d. samples from $\mathcal{D}^\star_+$ and $\mathcal{D}$ respectively
3: Set $T := (\varepsilon s)^{-q}$
4: **for** $i \in [T]$ **do**
5:     Define the survival set $S_i := H_1 \cap H_2 \cap \ldots \cap H_{i-1}$ if $i > 1$, and, otherwise, $S_i := \mathbb{R}^d$
6:     Find a hypothesis $h_i \in \mathcal{H}$ that minimizes $|\{x \in U \cap S_i \mid h_i(x) = 1\}|$ subject to satisfying $h_i(x) = 1$ for all $x \in P$
7:     Define $H_i := \{x : h_i(x) = 1\}$
8: **end for**

9: **Return** $H := H_1 \cap \ldots \cap H_T$

---

## E. Detailed Proof of Main Result (Theorem 3.1)

In this section, we will formally prove Theorem 3.1, which our main result for learning $w^\star$ in our truncated regression setting. We gave an overview of the proof in Section 3, but here we will rigorously present all technical parts and intermediate lemmas needed to prove the theorem. To begin with, we state the following definition that will be used repeatedly throughout the proofs.

**Definition 3** (Survival Probability). *Given a measurable $S \subseteq \mathbb{R}$ and two vectors $x, w \in \mathbb{R}^d$, define the survival probability of $x$ and parameter $w$ with respect to the survival set $S$ as $p(x, w; S) := \mathcal{N}(S; \langle w, x \rangle, 1)$.*

We will often use the shorthand $p_S(x) := p(x, w^\star; S)$. With this definition, the observed distribution of the features (see

Equation (1)) can be rewritten as

$$\mathcal{D}_{\mathrm{obs}}(x) := \frac{p(x, w^\star; S^\star) \cdot \mathcal{D}(x)}{\mathbb{E}_{x \sim \mathcal{D}}\left[p(x, w^\star; S^\star)\right]} = \frac{p_{S^\star}(x) \cdot \mathcal{D}(x)}{\mathbb{E}_{x \sim \mathcal{D}}\left[p_{S^\star}(x)\right]}. \tag{3}$$

Next, recall from our discussion in the main body that we are working under the following assumptions, which we state again using our new notation.

**Assumption 1** (Survival Probability). *There is a known $\alpha \in (0,1)$ such that $\mathbb{E}_{x \sim \mathcal{D}}[p(x, w^\star; S^\star)] \geq \alpha$.*

**Assumption 2** (Sub-Gaussianity of $x$). *There is a known $\sigma \geq 1$ such that $x \sim \mathcal{D}$ is $\sigma$-sub-Gaussian. That is, for any unit vector $v$, it holds that $\|\langle x - \mathbb{E}_{\mathcal{D}} x, v\rangle\|_{\psi_2} \leq \sigma$.*

**Assumption 3** (Identifiability). *There is a constant $\rho > 0$ such that $\mathbb{E}_{x \sim \mathcal{D}_{\mathrm{obs}}}\left[xx^\top\right] \succeq \rho^2 \cdot I$.*

In Assumption 2 as given in the main body, we additionally assume that $\|w^\star\|$ is bounded by a known constant $R$. Also, in the statement of Theorem 3.1, we assume a constant upper bound on $\|\mathbb{E}_{\mathcal{D}} x\|_2$. Moving forward, in order to keep dependencies clear, we will not assume any such bounds and will instead explicitly write these norms. We will also often use the shorthand $\mu := \mathbb{E}_{\mathcal{D}} x$. Hence, $\|\mu\|_2$ and $\|w^\star\|_2$ will appear repeatedly in our results, and of course if an upper bound is known for either of them it can be substituted. Furthermore, we will also state the following assumption, which is essentially redundant.

**Assumption 4** (Moment Bounds). *There exist known constants $\Lambda, M \geq 0$ such that, for any unit vector $v$,*

$$\mathbb{E}_{x \sim \mathcal{D}}\left[xx^\top\right] \preceq \Lambda^2 \cdot I \qquad and \qquad \mathbb{E}_{x \sim \mathcal{D}}\left[\langle x, v\rangle^4\right] \leq M^4.$$

To explain why we describe this assumption as redundant, note that, by standard properties of sub-Gaussian random variables (see Proposition 2.6.6 of (Vershynin, 2018)), it holds for any unit vector $v$ that $\mathbb{E}_{x \sim \mathcal{D}} \langle x - \mu, v\rangle^2 \lesssim \sigma^2$, and therefore $\mathbb{E}_{x \sim \mathcal{D}} \langle x, v\rangle^2 \lesssim \mathbb{E}_{x \sim \mathcal{D}} \langle x - \mu, v\rangle^2 + \langle \mu, v\rangle^2 \lesssim \sigma^2 + \|\mu\|_2^2$. Similarly, we can derive $\mathbb{E}_{x \sim \mathcal{D}} \langle x, v\rangle^4 \lesssim \sigma^4 + \|\mu\|_2^4$. Hence, both $\Lambda$ and $M$ in Assumption 4 can be substituted by $O(\sigma + \|\mu\|_2)$. The reason why we choose to explicitly define $\Lambda$ and $M$ is because (i) they allow us to more concisely present many of our results and calculations in the proofs, and (ii) some of our results do not require the full power of sub-Gaussianity, but continue to hold under only the moment bounds given in Assumption 4. Hence, we will repeatedly use $\Lambda$ and $M$ in our results, and the reader can substitute each of them with $O(\sigma + \|\mu\|_2)$ if desired.

Finally, recall from our discussion in Section 3.2 that, in order to be able to approximately learn the truncation set $S^\star$, we require a known upper bound $k$ on the number of intervals it consists of. It will be convenient for us to state this requirement as an additional assumption.

**Assumption 5** (Union of Intervals). *The survival set $S^\star$ is a union of at most $k$ intervals in $\mathbb{R}$, for a known integer $k$.*

With all the preliminary discussion out of the way, we now turn to presenting the proof of Theorem 3.1. Recall that our main algorithm is divided into two steps: first, we learn an approximation $S \approx S^\star$ of the truncation set, and then given this approximation we run PSGD on the perturbed log-likelihood $\mathscr{L}_S$ to obtain an estimate for $w^\star$. For each of these steps, we will state a theorem quantifying the guarantees of our algorithm. For the set-learning step, we have the following.

**Theorem E.1** (Set Learning Guarantees). *Suppose that Assumptions 1 to 5 hold. For $\varepsilon, \delta \in (0, 1/2)$, let*

$$n := \widetilde{O}\left(\varepsilon^{-\widetilde{\Theta}\left(\frac{\Lambda^4}{\rho^4 \alpha^2}(\sigma + \|\mu\|_2)^2\right)} \cdot (k + \log 1/\delta)\right).$$

*There exists an algorithm which draws $2n$ independent samples from the truncated regression model and outputs a set $S$, such that, with probability at least $1 - \delta$*

$$\mathbb{E}_{\mathcal{D}_{\mathrm{obs}}}\left[p(x, w^\star; S^\star \triangle S)\right] \leq \varepsilon.$$

*Moreover, the algorithm runs in time $O(n \log n)$, its output $S$ is a union of at most $k\varepsilon^{-\widetilde{\Theta}\left(\frac{\Lambda^2}{\rho^4 \alpha^2}(\sigma + \|\mu\|_2)^2\right)}$ intervals, and further it satisfies $S \subseteq [-R, R]$, where*

$$R \leq \widetilde{O}\left((1 + \|w^\star\|_2) \cdot \left(\|\mu\|_2 + \left(\sigma + \frac{\Lambda}{\alpha}\right)\sqrt{\log \frac{1}{\varepsilon}}\right)\right)$$

*and $\mu := \mathbb{E}_{\mathcal{D}} x$.*

As we discussed in Section 3, our main algorithm for set learning is Algorithm 1, which is an efficient implementation of Algorithm 3 of (Lee et al., 2026) for the special case where our hypothesis class contains unions of at most $k$ intervals on the real line. Note that, by our definition for the distribution $\mathcal{D}_y(w)$ in Section 3.2, for any measurable $T \subseteq \mathbb{R}$, we can write the mass $\mathcal{D}_y(w; T)$ that $\mathcal{D}_y(w)$ places on $T$ as

$$\mathcal{D}_y(w; T) := \int \mathbb{1}\left\{y \in T\right\} \mathcal{N}(y; \langle w, x \rangle, 1) \mathcal{D}_{\text{obs}}(x) \, dx \, dy = \underset{x \sim \mathcal{D}_{\text{obs}}}{\mathbb{E}} \left[ p(x, w; T) \right] \,,$$

where we used the definition of the survival probability $p(x, w; T)$, see Definition 3. In words, the mass that $\mathcal{D}_y(w)$ places on a set $T$ is, by definition, the probability that $y \in T$ if we draw $x \sim \mathcal{D}_{\text{obs}}$ and then draw $y \sim \mathcal{N}(\langle w, x \rangle, 1)$. Hence, our objective of learning a set $S$ such that $\mathcal{D}_y(w; S^\star \triangle S) \leq \varepsilon$ is equivalent to the relation $\mathbb{E}_{x \sim \mathcal{D}_{\text{obs}}}[p(x, w; S^\star \triangle S)] \leq \varepsilon$ appearing in the statement of Theorem E.1 above. However, Theorem E.1 is stronger than Theorem 3.4 which we presented in Section 3.1, in the sense that it also guarantees that the output set $S$ is fully contained within a bounded radius from the origin; this will be useful to us for technical reasons in our proofs.

Given an approximation $S$ as in Theorem E.1, the guarantees of the second part of our algorithm (*i.e.*, estimating $w^\star$) are given in the following.

**Theorem E.2** (Parameter Estimation Given $S \approx S^\star$)**.** *Suppose that Assumptions 1 to 5 hold, define $\mu := \mathbb{E}_{\mathcal{D}} x$, and let $\zeta > 0$ and $\delta \in (0, 1/2)$. Let $S \subset \mathbb{R}$ be a set satisfying the guarantees of Theorem E.1 for some $\varepsilon$ such that*

$$\varepsilon \leq e^{-\operatorname{poly}\left(\Lambda, M, \frac{1}{\rho}, \frac{1}{\alpha}\right)} \cdot \operatorname{poly}\left(\zeta, \frac{1}{d}, \frac{1}{\sigma + \|\mu\|_2}, \frac{1}{1 + \|w^\star\|_2}\right) \,.$$

*There exists an algorithm that, given sample access to the truncated regression model (Definition 1), takes as input the set $S$[9], the error $\varepsilon$ and the constants $\zeta, \delta$, and outputs a vector $\overline{w}$. The algorithm draws $N$ i.i.d. samples from the truncated regression model, where*

$$N = e^{\widetilde{O}\left(\frac{M^4 \Lambda^4}{\rho^6 \alpha^3}\right)} \cdot \widetilde{O}\left(\frac{d \cdot (\sigma + \|\mu\|_2)^4 \cdot (1 + \|w^\star\|_2)^2}{\zeta^2} \cdot \log \frac{1}{\delta}\right) \,.$$

*With probability at least $1 - \delta$, the output of the algorithm satisfies*

$$\|\overline{w} - w^\star\|_2 \leq \zeta$$

*and furthermore the algorithm terminates in time*

$$e^{\operatorname{poly}\left(\Lambda, M, \frac{1}{\rho}, \frac{1}{\alpha}\right)} \cdot \operatorname{poly}\left(\|w^\star\|_2, \|\mu\|_2, \sigma, d, k, \frac{1}{\zeta}, \log \frac{1}{\delta}\right) \,.$$

In other words, if we choose our set learning error $\varepsilon$ to be sufficiently small in the first step of the algorithm, then we can efficiently approximate $w^\star$ to high accuracy. Combining Theorem E.1 and Theorem E.2 immediately yields the desired Theorem 3.1, simply by first running the algorithm of Theorem E.1 with error $\varepsilon$ as given in Theorem E.2, and then running the algorithm of Theorem E.2. Hence, in the remainder of this section, we will present the proofs of these two intermediate theorems.

### E.1. Proof of Set Estimation Guarantees (Theorem E.1)

As we discussed in Section 3, our approach for learning the truncation set $S^\star$ relies on (i) an efficient algorithm (Algorithm 1) for positive-only learning unions of intervals (Theorem 3.4) and (ii) a way to obtain a distribution which is smooth w.r.t. the untruncated distribution of the response $y$ (Lemma 3.5). In this section, we first prove the correctness and efficiency of Algorithm 1 in Appendix E.1.1, thus showing Theorem 3.4, and then we prove our key Lemma 3.5 for establishing smoothness of a distribution in Appendix E.1.2. Afterwards, in Appendix E.1.3, we examine how we can make use of a warm start $w$ which is a constant distance away from $w^\star$, in order to actually sample from such a smooth distribution. Finally, in Appendix E.1.4, we combine our previous results with some additional analysis to establish our desired Theorem E.1.

---

[9]In particular, we require knowledge of the endpoints of the $\leq k/\varepsilon$ intervals constituting $S$. A simple membership oracle to $S$ does not suffice to run the subroutine of Lemma E.12 later on.

E.1.1. PROOF OF THEOREM 3.4 (CORRECTNESS OF SET LEARNING ALGORITHM UNDER SMOOTHNESS)

We begin by proving Theorem 3.4, which is our main result for positive-only learning of unions of intervals under smoothness.

*Proof of Theorem 3.4.* Consider running Algorithm 3 (whose guarantees are given in Theorem 3.3) in this setting, and we will show that this is equivalent to running Algorithm 1. Hence, the correctness of our learner follows from the correctness of Algorithm 3. We begin by giving an efficient procedure for implementing the ERM step of Algorithm 3. At any iteration $i$ of Algorithm 3, let $y^{(1)} \leq \ldots \leq y^{(n)}$ be the (ordered) positive samples of $P \subset \mathbb{R}$, and let $Q := U \cap S_i \subset \mathbb{R}$ be the unlabeled points that survive. Recall that our objective is to find a union of at most $k$ intervals which contain all points in $P$ and as few points of $Q$ as possible. Towards this, for any $1 \leq i \leq n - 1$, let $Q^{(i)} := Q \cap (y^{(i)}, y^{(i+1)})$ be the points of $Q$ between $y^{(i)}$ and $y^{(i+1)}$, and further let $Q^{(0)} := Q \cap (-\infty, y^{(0)})$ and $Q^{(n)} := Q \cap (y^{(n)}, \infty)$, so that $Q = \cup_{i=0}^{n} Q^{(i)}$.

We can assume w.l.o.g. that the optimal solution to our ERM problem is a union of $k$ disjoint intervals with endpoints in $P$. Indeed, if a solution contains an interval with endpoints not in $P$, we can simply shrink the interval until we hit the closest points of $P$, and this maintains feasiblity (*i.e.,*all points of $P$ are still covered by the solution) while making the solution no worse (*i.e.,*we can only cover fewer points of $Q$). Also, if a solution contains fewer than $k$ intervals, we can pick any one of these intervals which covers at least 3 points of $R$[10], and split it into two intervals by removing a sub-interval of the form $(y^{(i)}, y^{(i+1)})$, again maintaining feasibility while making the solution no worse.

Therefore, to optimally solve our ERM problem, it suffices to consider unions of $k$ disjoint intervals with endpoints in $P$, and we need to find the one which covers the least points of $Q$. It is clear that we can do this via the following greedy procedure: we start with the interval $[y^{(1)}, y^{(n)}]$, and we remove the $k - 1$ sub-intervals of the form $(y^{(i)}, y^{(i+1)})$ corresponding to the $k - 1$ sets $Q^{(i)}$ with the most points.

We now claim that, when our hypothesis class is $\mathcal{H}_k$, Algorithm 3 can be implemented in a *single* iteration, instead of requiring $1/(\varepsilon s)^q$ iterations. To see this, let $U^{(i)} := U \cap (y^{(i)}, y^{(i+1)})$ be the unlabeled points of $U$ which are between $y^{(i)}$ and $y^{(i+1)}$, and define the re-indexing $j_1, \ldots, j_{n-1}$ so that $|U^{(j_1)}| \geq \ldots \geq |U^{(j_{n-1})}|$. Consider running the first iteration of Algorithm 3, and implementing the ERM step as detailed above. That is, we start from the interval $[y^{(1)}, y^{(n)}]$, and we remove the sub-intervals corresponding to the sets $U^{(j_1)}, \ldots, U^{(j_{k-1})}$. Then, the set $H_1$ defined in Algorithm 3 will be

$$H_1 := \left[y^{(1)}, y^{(n)}\right] \setminus \left(\bigcup_{\ell=1}^{k-1} \left(y^{(j_\ell)}, y^{(j_{\ell+1})}\right)\right).$$

Consider subsequently running the second iteration of Algorithm 3, and again implementing the ERM step as above. Now, we wish to minimize the points of $U \cap H_1$ that are included in our solution. We again start from $[y^{(1)}, y^{(n)}]$, however, since $H_1$ already does not contain the sub-intervals corresponding to the sets $U^{(j_1)}, \ldots, U^{(j_{k-1})}$, we instead remove the sub-intervals corresponding to the next $k - 1$ biggest sets, *i.e.,*$U^{(j_k)}, \ldots, U^{(j_{2k-2})}$. Thus, the set $H_2$ defined in Algorithm 3 will be

$$H_2 := \left[y^{(1)}, y^{(n)}\right] \setminus \left(\bigcup_{\ell=k}^{2k-2} \left(y^{(j_\ell)}, y^{(j_{\ell+1})}\right)\right).$$

In the next iteration, we will minimize the points of $U \cap H_1 \cap H_2$ contained in our solution, and so on. Hence, we can inductively apply the above argument and obtain that

$$H_i := \left[y^{(1)}, y^{(n)}\right] \setminus \left(\bigcup_{\ell=(i-1)(k-1)+1}^{i(k-1)} \left(y^{(j_\ell)}, y^{(j_{\ell+1})}\right)\right).$$

Therefore, the final set returned by Algorithm 3 will be

$$\bigcap_{i=1}^{T} H_i = \left[y^{(1)}, y^{(n)}\right] \setminus \left(\bigcup_{\ell=1}^{T(k-1)} \left(y^{(j_\ell)}, y^{(j_{\ell+1})}\right)\right).$$

---

[10]Such an interval always exists since $r < n$.

where $T = 1/(\varepsilon s)^q$. Overall, to implement Algorithm 3 for the hypothesis class $\mathcal{H}_k$, it suffices to find the $\frac{k-1}{(\varepsilon s)^q}$ sub-intervals $(y^{(i)}, y^{(i+1)})$ containing the most unlabeled points of $U$. This can clearly be done by performing a single pass through the data to count the unlabeled points in each sub-interval, and then sorting these counts, as described in Algorithm 1. Finally, the sample complexity follows immediately from Theorem 3.3 and from the fact that $\mathrm{VC}(\mathcal{H}_k) = k + 1$. $\qquad\square$

### E.1.2. PROOF OF LEMMA 3.5 (SMOOTHNESS PROPERTY IN TRUNCATED LINEAR REGRESSION)

Next, we prove Lemma 3.5, which shows how to obtain the smooth distribution required by Theorem E.1 in our truncated regression setting. Recall from our discussion in Section 3.2 that, for any $w \in \mathbb{R}$ we define the distribution $\mathcal{D}_y(w)$ as follows: consider the distribution over $(x, y) \in \mathbb{R}^d \times \mathbb{R}$ where the marginal of $x$ is the $\mathcal{D}_{\mathrm{obs}}$ and the marginal of $y$ conditioned on $x$ is an untruncated Gaussian $\mathcal{N}(\langle w, x \rangle, 1)$. Then $\mathcal{D}_y(w)$ is the unconditional distribution of $y \in \mathbb{R}$.

To show Lemma 3.5, we need to relate the mass $\mathcal{D}_y(w^\star, T)$ that $\mathcal{D}_y(w^\star)$ places on some set $T \subseteq \mathbb{R}$ to the mass $\mathcal{D}_y(w, T)$ that $\mathcal{D}_y(w)$ places (for some $w \in \mathbb{R}^d$) on the same set $T$. Concretely, we need to ensure that there does not exist a set $T$ such that $\mathcal{D}_y(w^\star)$ places significantly more mass on $T$ compared to $\mathcal{D}_y(w)$. At a high level, there are two reasons why this might not hold: (i) the mean of $\mathcal{D}_y(w)$ can be much further away from $T$ compared to the mean of $\mathcal{D}_y(w^\star)$, and (ii) the distribution $\mathcal{D}_{\mathrm{obs}}$ of the features $x$ projected on $w$ might be significantly different compared to projecting on $w^\star$. As we will see in the proof, the first issue is taken care of by upper and lower bounds on the tail of the (Gaussian) noise (which ensure that $\mathcal{D}_y(w, T)$ will be lower bounded even if the mean of $\mathcal{D}_y(w^\star)$ is significantly shifted), while the second is resolved by sub-Gaussianity of $x$ (which ensures concentration around the mean of $\mathcal{D}_{\mathrm{obs}}$ in all directions).

**Lemma 3.5** (Smoothness). *Suppose that Assumptions 1 and 2 hold. Let $w \in \mathbb{R}^d$ be such that $\|w - w^\star\|_2 \leq D$, and assume that $\|\mathbb{E}_\mathcal{D} x\|_2 \leq \beta$ for $\beta > 0$. For any measurable $T \subseteq \mathbb{R}$ such that $\mathbb{E}_{x \sim \mathcal{D}_{\mathrm{obs}}}[\mathcal{N}(T; \langle w^\star, x \rangle, 1)] > 0$:*

$$\mathcal{D}_y(w^\star; T) \leq (1/s) \cdot [\mathcal{D}_y(w; T)]^{1/q}$$

*for constants $s = \Theta(1)$ and $q = \Theta\left(D^2\left(\sigma^2 \log 1/\alpha + \beta^2\right)\right)$.*

*Proof of Lemma 3.5.* By Lemma D.3, it follows that

$$\begin{aligned}
\mathcal{D}_y(w; T) &:= \int \mathbb{1}\{y \in T\} \mathcal{N}(y; \langle w, x \rangle, 1) \mathcal{D}_{\mathrm{obs}}(x) \, \mathrm{d}x \, \mathrm{d}y \\
&= \mathbb{E}_{\mathcal{D}_{\mathrm{obs}}}[p(x, w; T)] \\
&\gtrsim \mathbb{E}_{\mathcal{D}_{\mathrm{obs}}}\left[p(x, w^\star; T)^2 \exp\left(-\langle w - w^\star, x \rangle^2\right)\right].
\end{aligned}$$

We denote $p_T(x) := p(x, w^\star; T)$ and $\widetilde{w} := w - w^\star$ for brevity. Since the quantity inside the expectation is positive, for any $t > 0$ we have that

$$\begin{aligned}
\mathcal{D}_y(w; T) &\gtrsim \mathbb{E}_{\mathcal{D}_{\mathrm{obs}}}\left[p_T(x)^2 \cdot \exp\left(-\langle \widetilde{w}, x \rangle^2\right) \cdot \mathbb{1}\{|\langle \widetilde{w}, x \rangle| \leq t\}\right] \\
&\geq e^{-t^2} \mathbb{E}_{\mathcal{D}_{\mathrm{obs}}}\left[p_T(x)^2 \cdot \mathbb{1}\{|\langle \widetilde{w}, x \rangle| \leq t\}\right] \\
&= e^{-t^2}\left(\mathbb{E}_{\mathcal{D}_{\mathrm{obs}}}[p_T(x)^2] - \mathbb{E}_{\mathcal{D}_{\mathrm{obs}}}[p_T(x)^2 \cdot \mathbb{1}\{|\langle \widetilde{w}, x \rangle| > t\}]\right) \\
&\geq e^{-t^2}\left(\mathbb{E}_{\mathcal{D}_{\mathrm{obs}}}[p_T(x)^2] - \sqrt{\mathbb{E}_{\mathcal{D}_{\mathrm{obs}}}[p_T(x)^4] \Pr_{\mathcal{D}_{\mathrm{obs}}}(|\langle \widetilde{w}, x \rangle| > t)}\right)
\end{aligned}$$

where the last step is by the Cauchy–Schwarz inequality. Note that, by Lemma F.5 and standard sub-Gaussian concentration (see Proposition 2.6.1 of (Vershynin, 2018)), we have that for any $t > 0$

$$\Pr(|\langle \widetilde{w}, x \rangle| > t) \leq 2 \exp\left(-\frac{ct^2}{K^2 \|\widetilde{w}\|_2^2}\right) \leq 2 \exp\left(-\frac{ct^2}{K^2 D^2}\right)$$

where $K = \widetilde{\Theta}\left(\sqrt{\sigma^2 \log\frac{1}{\alpha} + \|\mu\|_2^2}\right)$ and $c$ is some absolute constant. Therefore, the above gives

$$\mathcal{D}_y(w; T) \gtrsim e^{-t^2}\left(\mathop{\mathbb{E}}_{\mathcal{D}_{\text{obs}}}\left[p_T(x)^2\right] - \exp\left(-\frac{ct^2}{2K^2D^2}\right)\sqrt{\mathop{\mathbb{E}}_{\mathcal{D}_{\text{obs}}}\left[p_T(x)^4\right]}\right).$$

We now pick

$$t := \Theta\left(KD\sqrt{\log\left(\frac{\mathbb{E}_{\mathcal{D}_{\text{obs}}}\left[p_T(x)^4\right]^{1/2}}{\mathbb{E}_{\mathcal{D}_{\text{obs}}}\left[p_T(x)^2\right]}\right)}\right)$$

where $t > 0$ due to Jensen's inequality. Substituting our choice into the above relation gives

$$\begin{aligned}
\mathcal{D}_y(w; T) &\gtrsim \left(\frac{\mathbb{E}_{\mathcal{D}_{\text{obs}}}\left[p_T(x)^2\right]}{\mathbb{E}_{\mathcal{D}_{\text{obs}}}\left[p_T(x)^4\right]^{1/2}}\right)^{\Theta(K^2D^2)} \cdot \mathop{\mathbb{E}}_{\mathcal{D}_{\text{obs}}}\left[p_T(x)^2\right] \\
&\geq \left(\mathop{\mathbb{E}}_{\mathcal{D}_{\text{obs}}}\left[p_T(x)^2\right]\right)^{\Theta(K^2D^2)+1} \\
&\geq \left(\mathop{\mathbb{E}}_{\mathcal{D}_{\text{obs}}}\left[p_T(x)\right]\right)^{\Theta(K^2D^2+1)} \\
&= [\mathcal{D}_y(w^\star; T)]^{\Theta(K^2D^2+1)}
\end{aligned}$$

where the second line is because $p_T(x) \leq 1$, the third is by Jensen's inequality, and the last is by recalling the definition of $\mathcal{D}_y$. $\qquad\square$

Hence, using Lemma 3.5 we can employ our set learning algorithm by using

- $y \sim \mathcal{D}_+^\star$ as the outcomes of true truncated linear regression model with parameter $w^\star$, *i.e.*, $\mathcal{D}_+^\star = \mathcal{D}_y(w^\star)$, and,

- $y \sim \mathcal{D}$ as the distribution $\mathcal{N}(\langle w, x\rangle, 1)$ where $x$ is drawn from the true truncated linear regression model with parameter $w^\star$. Hence $\mathcal{D} = \mathcal{D}_y(w)$ with $x \sim \mathcal{D}_{\text{obs}}$.

Note that the quality of the smoothness depends crucially on the distance between $w$ and $w^\star$.

### E.1.3. SET LEARNING GIVEN WARM START

In order to find the smooth distribution required to run the algorithm of Theorem 3.4, by Lemma 3.5, it suffices to find a warm start $\widehat{w}$ which is a constant distance away from $w^\star$, and we show how to do this in Lemma E.8 later on. Combining the above, we obtain the following intermediate result for approximately learning $S^\star$.

**Corollary E.3** (Estimating the truncation set). *Suppose Assumptions 1 to 5 hold. For $\varepsilon, \delta \in (0, 1/2)$, let*

$$n := \widetilde{O}\left(\varepsilon^{-\widetilde{\Theta}\left(\frac{\Lambda^2}{\rho^4\alpha^2}\left(\sigma+\|\mu\|_2\right)^2\right)} \cdot \left(k + \log\frac{1}{\delta}\right)\right)$$

*where $\mu := \mathbb{E}_{\mathcal{D}} x$. There exists an algorithm which draws $2n$ independent samples from the truncated regression model and outputs a set $S$, such that, with probability at least $1 - \delta$:*

$$\mathop{\mathbb{E}}_{x\sim\mathcal{D}_{\text{obs}}}\left[p(x, w^\star; S^\star\triangle S)\right] \leq \varepsilon.$$

*Moreover, the algorithm runs in time $O(n\log n)$, and its output $S$ is a union of at most $k\varepsilon^{-\widetilde{\Theta}\left(\frac{\Lambda^2}{\rho^4\alpha^2}\left(\sigma+\|\mu\|_2\right)^2\right)}$ intervals.*

### E.1.4. PUTTING EVERYTHING TOGETHER

We now show how to strengthen this result to obtain Theorem E.1, which also ensures that $S$ is contained within a bounded radius from the origin. To do so, we first establish that the truncated regression model places most of the mass of the response $y$ on an interval of bounded radius.

**Lemma E.4.** *Suppose that Assumptions 1, 2 and 4 hold. Let $\widehat{\mu} := \mathbb{E}_{\mathcal{D}_{\mathrm{obs}}} x$, and define the interval*

$$T := [\langle w^\star, \widehat{\mu} \rangle - L, \langle w^\star, \widehat{\mu} \rangle + L] \quad \text{where} \quad L \leq \widetilde{O}\left( (1 + \|w^\star\|_2) \left( \sigma + \frac{\Lambda}{\alpha} \right) \sqrt{\log \frac{1}{\varepsilon}} \right).$$

*Then it holds that*

$$\mathbb{E}_{x \sim \mathcal{D}_{\mathrm{obs}}} [p_T(x)] \geq 1 - \frac{\varepsilon}{2}$$

*where we denote $p_T(x) := p(x, w^\star, T)$ for short.*

*Proof.* First, by Lemma F.5 and standard subgaussian concentration (see Proposition 2.6.1 of (Vershynin, 2018)),

$$\Pr_{\mathcal{D}_{\mathrm{obs}}} \left( |\langle w^\star, x - \widehat{\mu} \rangle| \leq r \right) \geq 1 - \frac{\varepsilon}{4}$$

for $r = \widetilde{O}\left( (1 + \|w^\star\|_2) \left( \sigma + \frac{\Lambda}{\alpha} \right) \sqrt{\log \frac{1}{\varepsilon}} \right)$. Now, let $\mathcal{P}$ be the distribution over $(x, y) \in \mathbb{R}^d \times \mathbb{R}$ where the marginal of $x$ is $\mathcal{D}_{\mathrm{obs}}$ and the marginal of $y$ conditioned on $x$ is an *untruncated* Gaussian $\mathcal{N}(\langle w^\star, x \rangle, 1)$. It is immediate that

$$\mathbb{E}_{\mathcal{D}_{\mathrm{obs}}} [p_T(x)] = \Pr_{\mathcal{P}} (y \in T)$$

$$\geq \Pr_{\mathcal{P}} (y \in T \mid |\langle w^\star, x - \widehat{\mu} \rangle| \leq r) \cdot \Pr_{\mathcal{D}_{\mathrm{obs}}} (|\langle w^\star, x - \widehat{\mu} \rangle| \leq r)$$

$$\geq \left( 1 - \frac{\varepsilon}{4} \right) \cdot \Pr_{\mathcal{P}} (y \in T \mid |\langle w^\star, x - \widehat{\mu} \rangle| \leq r).$$

To bound this last probability, observe that since $T$ is symmetric around $\langle w^\star, \widehat{\mu} \rangle$, and the marginal of $y$ conditioned on $x$ is $\mathcal{N}(\langle w^\star, x \rangle, 1)$, the probability that $y \in T$ is minimized when $\langle w^\star, x \rangle$ is as far away from $\langle w^\star, \widehat{\mu} \rangle$ as possible. Hence, the conditional probability above is at least the probability that $y \in T$ when $\langle w^\star, x \rangle = \langle w^\star, \widehat{\mu} \rangle + r$ a.s., that is

$$\Pr_{\mathcal{P}} (y \in T \mid |\langle w^\star, x - \widehat{\mu} \rangle| \leq r) \geq \Pr_{y \sim \mathcal{N}(\langle w^\star, \widehat{\mu} \rangle + r, 1)} (y \in T)$$

$$= \Pr_{z \sim \mathcal{N}(0,1)} (-L - r \leq z \leq L - r)$$

$$\geq \Pr_{z \sim \mathcal{N}(0,1)} (|z| \leq L - r)$$

$$\geq 1 - 2 \exp\left( -\frac{(L - r)^2}{2} \right)$$

$$\geq 1 - \frac{\varepsilon}{4}$$

where the second line is by definition of the set $T$, the second-to-last is by Gaussian concentration, and the last is because $L = O\left( r + \sqrt{\log 1/\varepsilon} \right)$. Therefore, combining our last two displays yields

$$\mathbb{E}_{\mathcal{D}_{\mathrm{obs}}} [p_T(x)] \geq \left( 1 - \frac{\varepsilon}{4} \right)^2$$

and the lemma follows by using the inequality $(1 - t/2)^2 \geq 1 - t$. $\qquad \square$

Combining our above results, we can finally obtain a set estimation algorithm which satisfies all of our desired guarantees.

*Proof of Theorem E.1.* Denote $I := [-R, R]$ and $p_T(x) := p(x, w^\star; T)$ for brevity. We first use the algorithm of Corollary E.3 to learn a set $S'$ such that $\mathbb{E}_{\mathcal{D}_{\mathrm{obs}}} [p_{S' \triangle S^\star}(x)] \leq \frac{\varepsilon}{2}$, and then we return $S := S' \cap I$. We will now analyze the quality of the estimator $S$, *i.e.*, the loss $\mathbb{E}_{\mathcal{D}_{\mathrm{obs}}} [p_{S \triangle S^\star}(x)]$.

Let $T \subseteq \mathbb{R}$ be the interval and $L$ be the radius defined in Lemma E.4. Also, let $\widehat{\mu} := \mathbb{E}_{\mathcal{D}_{\mathrm{obs}}} x$. By Lemma F.3, it follows that:

$$|\langle w^\star, \widehat{\mu} \rangle| + L \leq |\langle w^\star, \mu \rangle| + \frac{\Lambda}{\alpha} + L \leq \|w^\star\|_2 \|\mu\|_2 + \frac{\Lambda}{\alpha} + L \leq R.$$

Therefore, $T \subseteq I$, and by Lemma E.4 it follows that $\mathbb{E}_{\mathcal{D}_{\mathrm{obs}}} [p_I(x)] \geq 1 - \frac{\varepsilon}{2}$. Thus, to prove the claim, observe that

$$\begin{aligned} \mathbb{E}_{\mathcal{D}_{\mathrm{obs}}} [p_{S \triangle S^\star}(x)] &= \mathbb{E}_{\mathcal{D}_{\mathrm{obs}}} \left[ p_{(S' \cap I) \triangle S^\star}(x) \right] \\ &= \mathbb{E}_{\mathcal{D}_{\mathrm{obs}}} \left[ p_{(S' \cap I) \cup S^\star}(x) \right] - \mathbb{E}_{\mathcal{D}_{\mathrm{obs}}} [p_{S' \cap I \cap S^\star}(x)]. \end{aligned}$$

To bound the first term, we have

$$p_{(S' \cap I) \cup S^\star}(x) = p_{(S' \cup S^\star) \cap (S^\star \cup I)}(x) \leq p_{S' \cup S^\star}(x)$$

and for the second term we have

$$\begin{aligned} p_{I \cap (S' \cap S^\star)}(x) &\geq p_I(x) + p_{S' \cap S^\star}(x) - p_{I \cap (S' \cap S^\star)}(x) \\ &= p_I(x) + p_{S' \cup S^\star}(x) - p_{S' \triangle S^\star}(x) - p_{I \cap (S' \cap S^\star)}(x) \\ &\geq p_I(x) + p_{S' \cup S^\star}(x) - p_{S' \triangle S^\star}(x) - 1 \end{aligned}$$

so substituting into our above display we obtain

$$\mathbb{E}_{\mathcal{D}_{\mathrm{obs}}} [p_{S \triangle S^\star}(x)] \leq \mathbb{E}_{\mathcal{D}_{\mathrm{obs}}} [p_{S' \triangle S^\star}(x)] - \mathbb{E}_{\mathcal{D}_{\mathrm{obs}}} [p_I(x)] + 1 \leq \varepsilon$$

since $\mathbb{E}_{\mathcal{D}_{\mathrm{obs}}} [p_{S' \triangle S^\star}(x)] \leq \varepsilon/2$. $\qquad\square$

## E.2. Proof of Parameter Estimation Guarantees (Theorem E.2)

Throughout this section, we will assume that we have computed an approximation $S \approx S^\star$ satisfying the guarantees of Theorem E.1 with some set learning error $\varepsilon$, using the machinery of the previous section. Given such an $S$, we will show how to obtain an estimate for the desired parameter vector $w^\star$, thus proving Theorem E.2. As we discussed in Section 3.3, our algorithm is PSGD on the following perturbed negative log-likelihood objective:

$$\mathscr{L}_S(w) := - \mathbb{E}_{x \sim \mathcal{D}_{\mathrm{obs}}} \left[ \mathbb{E}_{y \sim \mathcal{N}(\langle w^\star, x \rangle, 1, S \cap S^\star)} [\log \mathcal{N}(y; \langle w, x \rangle, 1, S) \mid x] \right]. \qquad \text{(Perturbed NLL)}$$

In all that follows, we will drop the inner conditioning when it is clear from context. The reason why the inner expectation is over $\mathcal{N}(\langle w^\star, x \rangle, 1, S \cap S^\star)$ instead of $\mathcal{N}(\langle w^\star, x \rangle, 1, S^\star)$ (from which our samples $y$ are actually drawn conditioned on $x$) is because, unless $S^\star \subseteq S$, in the second case the distribution would place mass in regions where the quantity inside the expectation is undefined. Furthermore, note that the perturbed NLL is different from simply substituting $S$ for $S^\star$ in the expression of the negative log-likelihood, *i.e.*, it is different from the expression

$$- \mathbb{E}_{x \sim \mathcal{D}_{\mathrm{obs}}} \left[ \mathbb{E}_{y \sim \mathcal{N}(\langle w^\star, x \rangle, 1, S)} [\log \mathcal{N}(y; \langle w, x \rangle, 1, S) \mid x] \right].$$

The reason for this choice is that, to run our PSGD algorithm, we will need to implement a gradient sampler for $\mathscr{L}_S$, and this will require us to approximately sample from the distribution in the expression of $\mathscr{L}_S$. That is, we will need to approximately sample from the distribution over $(x, y) \in \mathbb{R}^d \times \mathbb{R}$ where the marginal of $x$ is $\mathcal{D}_{\mathrm{obs}}$ and the marginal of $y$ conditioned on $x$ is $\mathcal{N}(\langle w^\star, x \rangle, 1, S \cap S^\star)$, and we will see that this is indeed possible. However, if we instead used the latter display for the expression of the perturbed likelihood, the marginal of $y$ conditioned on $x$ in the distribution we wish to sample from would be $\mathcal{N}(\langle w^\star, x \rangle, 1, S)$. Unless $S \subseteq S^\star$, it is unclear how we would be able to sample from the desired distribution in this case; indeed, we only have sample access to the truncated regression model, and this model only places mass for $y$ inside of $S^\star$.

As we discussed, in the general case, the perturbed log-likelihood is not a log-likelihood and, hence, may not have any desirable typical properties of log-likelihood, such as a global minimum at $w^\star$. Through the analysis in the following sections, we will establish that the perturbed NLL indeed satisfies properties that allow us to optimize it and obtain an estimate for $w^\star$.

### E.2.1. GRADIENT AND HESSIAN OF PERTURBED NLL

To begin with, we give the expressions for the gradient and Hessian of the perturbed log-likelihood. These expressions can be verified by direct computation, and we present the details in Appendix E.3.1.

**Lemma E.5** (Gradient and Hessian of $\mathscr{L}_S$). *For any measurable set $S \subseteq \mathbb{R}$, and any $w \in \mathbb{R}^d$*

$$\boldsymbol{\nabla}\mathscr{L}_S(w) = \mathop{\mathbb{E}}_{x \sim \mathcal{D}_{\mathrm{obs}}} \left[ - \mathop{\mathbb{E}}_{y \sim \mathcal{N}(\langle w^\star, x \rangle, 1, S^\star \cap S)} [y \cdot x] + \mathop{\mathbb{E}}_{z \sim \mathcal{N}(\langle w, x \rangle, 1, S)} [z \cdot x] \right],$$

$$\boldsymbol{\nabla}^2 \mathscr{L}_S(w) = \mathop{\mathbb{E}}_{x \sim \mathcal{D}_{\mathrm{obs}}} \left[ \mathop{\mathrm{Var}}_{z \sim \mathcal{N}(\langle w, x \rangle, 1, S)} [z] \cdot xx^\top \right].$$

Note that $\boldsymbol{\nabla}^2 \mathscr{L}_S(w) \succeq 0$ for all $w \in \mathbb{R}^d$, hence $\mathscr{L}_S$ is convex, and we can indeed hope to optimize it via PSGD. However, since we care about converging to the actual minimizer of $\mathscr{L}_S$ (and not just converging in function value to the minimum), we additionally need to ensure that $\mathscr{L}_S$ is *strongly* convex in an area around its minimizer.

### E.2.2. LOCAL STRONG CONVEXITY OF $\mathscr{L}_S$

In order to show strong convexity of $\mathscr{L}_S$ around its minimizer, we begin by establishing a lower bound on the eigenvalues of its Hessian.

**Lemma E.6.** *Suppose Assumptions 1, 3 and 4 hold. For any $w \in \mathbb{R}^d$ and any set $S \subseteq \mathbb{R}$ satisfying Equation (2) for some $\varepsilon > 0$, it holds that*

$$\boldsymbol{\nabla}^2 \mathscr{L}_S(w) \succeq \Omega \left( \frac{\alpha^5 \rho^{10}}{\Lambda^8} \cdot e^{-\frac{10M^4}{\rho^2 \alpha} \|w - w^\star\|_2^2} - \frac{M^{4/3} \Lambda^{2/3} \varepsilon^{1/3}}{\alpha^{2/3}} \right) \cdot I.$$

The proof of Lemma E.6 is given in Appendix E.3.2. A couple of observations are in order. First, the above bound is only useful for sufficiently small values of $\varepsilon$. This is not an issue, since $\varepsilon$ is the set estimation accuracy we require from the algorithm of Corollary E.3, and therefore we can select it appropriately. Secondly, and more importantly, the above bound is only useful for parameter vectors $w$ that are a constant distance away from the true vector $w^\star$. In order to leverage the convergence of PSGD for strongly convex functions, Lemma E.6 implies that we need to ensure that PSGD remains within a constant distance of $w^\star$. A simple way to do so is to find some point $\widehat{w}$ a constant distance away from $w^\star$, and define our projection set to be a ball of constant radius around $\widehat{w}$. However, it is not trivial to find such a point $\widehat{w}$ in the first place. In the following section, we will see an efficient algorithm to find such a warm-start point.

### E.2.3. EFFICIENTLY FINDING A WARM START

Our objective is to efficiently find an initial point $w_0$ that is a constant distance away from $w^\star$. To do so, we first recall the standard linear regression setting without truncation, where we are given $n$ datapoints $\left\{ \left( x^{(i)}, y^{(i)} \right) \right\}_{i \in [n]}$, and our objective is to find the Ordinary Least-Squares (OLS) minimizer, *i.e.*

$$\widehat{w} := \mathop{\arg\min}_{w \in \mathbb{R}^d} \frac{1}{n} \sum_{i \in [n]} \left( \left\langle w, x^{(i)} \right\rangle - y^{(i)} \right)^2.$$

In this setting, it is standard that writing down the first-order optimality condition for $\widehat{w}$ gives the following closed-form expression

$$\widehat{w} := \left( \frac{1}{n} \sum_i x^{(i)} x^{(i)\top} \right)^{-1} \cdot \left( \frac{1}{n} \sum_i y^{(i)} \cdot x^{(i)} \right). \tag{4}$$

Our main claim in this section is that, in our truncated regression setting, the OLS minimizer given in Equation (4) is a good enough approximation of the true parameter vector $w^\star$, in the sense that $\|\widehat{w} - w^\star\|_2 \leq O(1)$, *in spite of* the presence of truncation.[11] The following lemma shows that this is indeed true, at least in the infinite-sample regime.

---

[11]Note that, by Assumption 3, it follows that $\mathbb{E}_{\mathcal{D}_{\mathrm{obs}}} \left[ xx^\top \right]$ is fully dimensional, and so if $n \geq d$ then $\frac{1}{n} \sum_i x^{(i)} x^{(i)\top}$ is invertible w.p. 1, hence $\widehat{w}$ above is well-defined.

**Lemma E.7.** *Suppose that Assumptions 1, 3 and 4 hold, and define*

$$\widetilde{w} := \left( \underset{x \sim \mathcal{D}_{\mathrm{obs}}}{\mathbb{E}} \left[ xx^\top \right] \right)^{-1} \cdot \underset{x \sim \mathcal{D}_{\mathrm{obs}}}{\mathbb{E}} \left[ \underset{y \sim \mathcal{N}(\langle w^\star, x \rangle, 1, S^\star)}{\mathbb{E}} [y] \cdot x \right] . \tag{5}$$

*Then it holds that*

$$\|\widetilde{w} - w^\star\|_2 \le \frac{3\Lambda}{2\rho^2 \alpha} .$$

*Proof.* Note that, by Assumption 3, the matrix $\mathbb{E}_{\mathcal{D}_{\mathrm{obs}}} \left[ xx^\top \right]$ is invertible, so $\widetilde{w}$ is well-defined. Furthermore, observe that $\mathbb{E}_{\mathcal{D}_{\mathrm{obs}}} \left[ xx^\top \right] \cdot w^\star = \mathbb{E}_{\mathcal{D}_{\mathrm{obs}}} \left[ \langle w^\star, x \rangle \cdot x \right]$. Therefore

$$\widetilde{w} - w^\star = \left( \underset{x \sim \mathcal{D}_{\mathrm{obs}}}{\mathbb{E}} \left[ xx^\top \right] \right)^{-1} \cdot \underset{x \sim \mathcal{D}_{\mathrm{obs}}}{\mathbb{E}} \left[ \left( \underset{y \sim \mathcal{N}(\langle w^\star, x \rangle, 1, S^\star)}{\mathbb{E}} [y] - \langle w^\star, x \rangle \right) \cdot x \right]$$

By Assumption 3 we have that $\mathbb{E}_{\mathcal{D}_{\mathrm{obs}}} \left[ xx^\top \right] \succeq \rho^2 \cdot I$, which implies that $0 \prec \left( \mathbb{E}_{\mathcal{D}_{\mathrm{obs}}} \left[ xx^\top \right] \right)^{-1} \preceq \frac{1}{\rho^2} \cdot I$. Thus, taking norms above and applying Lemma F.4, we obtain

$$\|\widetilde{w} - w^\star\|_2 \le \left\| \underset{x \sim \mathcal{D}_{\mathrm{obs}}}{\mathbb{E}} \left[ xx^\top \right] \right\|_{\mathrm{op}}^{-1} \cdot \frac{3\Lambda}{2\alpha} \le \frac{3\Lambda}{2\rho^2 \alpha} .$$

$\square$

In order to convert Lemma E.7 into a finite-sample guarantee for $\widehat{w}$ in Equation (4), we will need to use concentration of the distribution of features, and in fact sub-Gaussianity (Assumption 2) suffices to show the following lemma, whose proof is given in Appendix E.3.3.

**Lemma E.8.** *Suppose that Assumptions 1 to 4 hold. Let $\left\{ \left( x^{(i)}, y^{(i)} \right) \right\}_{i \in [n]}$ be i.i.d. samples from the truncated regression model, and suppose that*

$$n \ge \widetilde{\Omega} \left( \left( 1 + \|w^\star\|_2 \right)^2 \cdot \left( \sigma + \|\mu\|_2 \right)^4 \cdot \frac{\Lambda^4}{\rho^8 \alpha^2} \cdot d \cdot \log \frac{1}{\delta} \right) \tag{6}$$

*where $\mu := \mathbb{E}_{\mathcal{D}} x$. Let $\widehat{w}$ be defined as in Equation (4). Then with probability at least $1 - \delta$*

$$\|\widehat{w} - w^\star\|_2 \le \frac{6\Lambda}{\rho^2 \alpha} .$$

Therefore, Lemma E.7 shows that we can efficiently compute our initial point $\widehat{w}$ from Equation (4), using a number of samples $n$ almost-linear in the dimension, and then we can define the projection set for PSGD as

$$\mathcal{P} := B \left( \widehat{w}, \frac{6\Lambda}{\rho^2 \alpha} \right) \tag{7}$$

so that $w^\star \in \mathcal{P}$ w.p. $1 - \delta$. By Lemma E.6, we obtain that for all $w \in \mathcal{P}$ that

$$\boldsymbol{\nabla}^2 \mathscr{L}_S(w) \succeq \Omega \left( \frac{\alpha^5 \rho^{10}}{\Lambda^8} \cdot e^{-\frac{45 M^4 \Lambda^4}{\rho^6 \alpha^3}} - \frac{M^{4/3} \Lambda^{2/3} \varepsilon^{1/3}}{\alpha} \right) \cdot I .$$

where $\varepsilon$ is the set learning accuracy we require from the algorithm of Theorem E.1. Hence, we can choose $\varepsilon$ to be a sufficiently small constant, so that the first term above dominates the second, and obtain the following strong convexity guarantee.

**Corollary E.9.** *Suppose that Assumptions 1 to 4 hold. Let $\left\{ \left( x^{(i)}, y^{(i)} \right) \right\}_{i \in [n]}$ be i.i.d. samples from the truncated regression model, for $n$ satisfying Equation (6). Let $\widehat{w}$ be defined as in Equation (4), and $\mathcal{P}$ be defined as in Equation (7). Finally, let $S$ be a set satisfying the guarantee of Theorem E.1, with accuracy parameter $\varepsilon$ such that*

$$\varepsilon \le \frac{\alpha^{17} \rho^{30}}{2 \Lambda^{26} M^4} \cdot e^{-\frac{135 M^4 \Lambda^4}{\rho^6 \alpha^3}} . \tag{8}$$

*Then, with probability at least $1 - \delta$, it holds that $w^\star \in \mathcal{P}$, and further, for all $w \in \mathcal{P}$*

$$\nabla^2 \mathscr{L}_S(w) \succeq \Omega \left( \frac{\alpha^5 \rho^{10}}{\Lambda^8} \cdot e^{-\frac{45M^4\Lambda^4}{\rho^6\alpha^3}} \right) \cdot I . \tag{9}$$

Overall, we have shown how to construct a projection set $\mathcal{P}$ such that the perturbed log-likelihood $\mathscr{L}_S$ is strongly convex everywhere in $\mathcal{P}$, and so we can indeed run PSGD using this projection set in order to minimize $\mathscr{L}_S$.f However, recall that the minimizer of $\mathscr{L}_S$ in $\mathcal{P}$ is not, in general, the desired parameter vector $w^\star$. Hence, we now turn our attention to showing that this minimizer satisfies a desirable "closeness" property with respect to $w^\star$.

E.2.4. CLOSENESS OF $w_{\min}$ AND $w^\star$

In what follows, we use $w_{\min}$ to denote the minimizer of $\mathscr{L}_S$ in the projection set $\mathcal{P}$ given in Equation (7), *i.e.*

$$w_{\min} := \arg\min_{w \in \mathcal{P}} \mathscr{L}_S(w)$$

As we discussed, $w_{\min}$ will not, in general, be the desired $w^\star$. However, in this section, we will show that we can make $w_{\min}$ be arbitrarily close to $w^\star$, by picking a sufficiently small set learning accuracy $\varepsilon$ in Corollary E.3. To control the distance $\|w_{\min} - w^\star\|_2$, we will first bound the norm of $\nabla \mathscr{L}_S$ at $w^\star$ (which is in $\mathcal{P}$ w.h.p.), and then we will leverage standard consequences of the strong convexity of $\mathscr{L}_S$ over $\mathcal{P}$. Towards this, our main technical result is the following lemma, whose proof is given in Appendix E.3.4.

**Lemma E.10** (Bound on Norm of Gradient). *Suppose that Assumptions 1, 2 and 4 hold, and let $S$ be a measurable set satisfying the guarantee of Theorem E.1, for some $\varepsilon > 0$ satisfying $\varepsilon < 1/8 \cdot \alpha^3$. Then*

$$\|\nabla \mathscr{L}_S(w^\star)\|_2 \leq \widetilde{O}\left( \frac{\Lambda^2}{\alpha} \cdot (1 + \|w^\star\|_2) \cdot (\|\mu\|_2 + \sigma) \sqrt{d} \varepsilon^{1/6} \right)$$

*where $\mu := \mathbb{E}_\mathcal{D} x$.*

Given the bound of Lemma E.10, it is now straightforward to control the desired distance of $w_{\min}$ from $w^\star$, as follows.

**Lemma E.11** (Closeness). *Suppose that Assumptions 1 to 4 hold, and let $S$ be a measurable set satisfying the guarantee of Theorem E.1 for some $\varepsilon > 0$ satisfying Equation (8). Let $\mathcal{P}$ be the projection set defined in Equation (7) for some $\widehat{w} \in \mathbb{R}^d$, and let $w_{\min}$ be the minimizer of $\mathscr{L}_S$ over $\mathcal{P}$. If $w^\star \in \mathcal{P}$, then it holds that*

$$\|w^\star - w_{\min}\|_2 \leq e^{\widetilde{O}\left(\frac{M^4\Lambda^4}{\rho^6\alpha^3}\right)} \cdot \widetilde{O}\left( (1 + \|w^\star\|_2) \cdot (\|\mu\|_2 + \sigma) \sqrt{d} \varepsilon^{1/6} \right)$$

*where $\mu := \mathbb{E}_\mathcal{D} x$.*

*Proof.* Let $\kappa := e^{-\widetilde{O}\left(\frac{M^4\Lambda^4}{\rho^6\alpha^3}\right)}$, such that by the guarantee on $\varepsilon$ and Corollary E.9 it holds that $\nabla^2 \mathscr{L}_S(w) \succeq \kappa \cdot I$ for any $w \in \mathcal{P}$. By monotonicity of the gradient of a strongly convex function, we have

$$\langle \nabla \mathscr{L}_S(w^\star) - \nabla \mathscr{L}_S(w_{\min}), w^\star - w_{\min} \rangle \geq \kappa \cdot \|w^\star - w_{\min}\|_2^2$$

First-order optimality of $w_{\min}$ implies that

$$\langle \nabla \mathscr{L}_S(w_{\min}), w^\star - w_{\min} \rangle \geq 0$$

and therefore we obtain

$$\|w^\star - w_{\min}\|_2^2 \leq \frac{1}{\kappa} \langle \nabla \mathscr{L}_S(w^\star), w^\star - w_{\min} \rangle \leq \frac{1}{\kappa} \|\nabla \mathscr{L}_S(w^\star)\|_2 \|w^\star - w_{\min}\|_2 .$$

The claim now follows by Lemma E.10. $\qquad\square$

Therefore, Lemma E.11 establishes that we can make $w_{\min}$ be arbitrarily close to $w^\star$, by appropriately picking the set learning accuracy $\varepsilon$ (of course, smaller values of $\varepsilon$ imply increased sample complexity and runtime in Corollary E.3). As such, in order to find a parameter vector arbitrarily close to $w^\star$, we can simply run PSGD on $\mathscr{L}_S$ on $\mathcal{P}$ (over which $\mathscr{L}_S$ is strongly convex) until we converge to the minimizer $w_{\min}$.

Having established this result, we now turn to algorithmic considerations for PSGD.

E.2.5. EFFICIENT GRADIENT SAMPLER

To run the PSGD algorithm we need, at each iteration, to produce a vector $g(w)$ such that $\mathbb{E}[g(w)] = \boldsymbol{\nabla}\mathscr{L}_S(w)$, where $w \in \mathbb{R}$ is the point we are considering at the current iteration. Recall from Lemma E.5 that

$$\boldsymbol{\nabla}\mathscr{L}_S(w) = \mathop{\mathbb{E}}_{x \sim \mathcal{D}_{\mathrm{obs}}} \left[ - \mathop{\mathbb{E}}_{y \sim \mathcal{N}(\langle w^\star, x \rangle, 1, S^\star \cap S)} [y \cdot x] + \mathop{\mathbb{E}}_{z \sim \mathcal{N}(\langle w, x \rangle, 1, S)} [z \cdot x] \right].$$

Implementing an exact gradient sampler is challenging. For example, to sample from the distribution of the first term, one approach would be to draw samples $(x, y)$ from our truncated regression model, and then perform rejection sampling to ensure that $y \in S$. This would indeed give the correct marginal of $y$ given $x$ (i.e., $\mathcal{N}(\langle w^\star, x \rangle, 1, S^\star \cap S)$), however the marginal of $x$ would not be $\mathcal{D}_{\mathrm{obs}}$, but a biased version of it. Furthermore, for the second term, one could first draw $x \sim \mathcal{D}_{\mathrm{obs}}$ from the truncated regression model to ensure that the marginal of $x$ is correct, and then use rejection sampling to draw $z \sim \mathcal{N}(\langle w, x \rangle, 1, S)$ conditionally on $x$. This would indeed give an unbiased sample for the second term; however, because we have not assumed that every $x$ in the support of $\mathcal{D}_{\mathrm{obs}}$ has at least constant survival probability $p(x, w, S)$, we would be unable to bound the runtime of the rejection sampling procedure.

Therefore, instead of implementing an exact gradient sampler, we will instead focus on generating *approximate* gradient samples $g(w)$ such that $\mathbb{E}[g(w)] = \boldsymbol{\nabla}\mathscr{L}_S(w) + b$, for some controlled bias vector $b$. We will make use of the following result.

**Lemma E.12** (Lemma 13 of (Daskalakis et al., 2019)). *Let $\mu \in R$, and let $S \subset R$ be a union of $k$ closed intervals, such that $\mathcal{N}(S; \mu, 1) \geq a$ for some $a > 0$. Then, for any $\zeta > 0$, there exists an algorithm that runs in time $\mathrm{poly}(\log(1/a, \zeta), k)$ and returns a sample from a distribution with support $\subseteq S$ and TV distance from $\mathcal{N}(\mu, 1, S)$ at most $\zeta$.*

Our method for obtaining approximate gradient samples is given in Algorithm 4.

---

**Algorithm 4** Approximate Gradient Sampler

---

**Require:** Parameter vector $w \in \mathbb{R}^d$, set $S \subseteq R$, accuracy parameter $\zeta$
**Ensure:** Sample access to the truncated regression model

    Draw independent $(\widehat{x}, \widehat{y}), (\widetilde{x}, \widetilde{y})$ from the truncated regression model
    Obtain sample $\widetilde{z}$ by calling the algorithm of Lemma E.12 for $S$, $\mu := \langle w, \widetilde{x} \rangle$ and $\zeta$
    **Return** $g(w) := \widetilde{z} \cdot \widetilde{x} - \widehat{y} \cdot \widehat{x}$

---

To give some intuition about our gradient sampler, let

1. $P$ be a distribution over $(x, y) \in \mathbb{R}^d \times \mathbb{R}$ such that the marginal of $x$ is $\mathcal{D}_{\mathrm{obs}}$ and the marginal of $y$ given $x$ is $\mathcal{N}(\langle w^\star, x \rangle, 1, S^\star \cap S)$. Note that $P$ depends only on $S$ (but not $w$).

2. $Q$ be a distribution over $(x, z) \in \mathbb{R}^d \times \mathbb{R}$ such that the marginal of $x$ is $\mathcal{D}_{\mathrm{obs}}$ and the marginal of $z$ given $x$ is $\mathcal{N}(\langle w, x \rangle, 1, S)$. Note that $Q$ depends both on $S$ and $w$.

If we could sample from $P$ and $Q$, we would have an exact sampler for the desired gradient and we would be done. Our sampler samples from two other distributions $P', Q'$ that serve as proxies to $P, Q$. In more detail, let

1. $P'$ be the distribution on $(\widehat{x}, \widehat{y})$, which corresponds to sampling from the true model with parameters $w^\star, S^\star$. We prove that the total variation distance between $P'$ and $P$ is small by using the fact that $S$ is a good approximation of $S^\star$.

2. $Q'$ be the distribution on $(\widetilde{x}, \widetilde{z})$, which corresponds to drawing first $x \sim \mathcal{D}_{\mathrm{obs}}$ and then drawing the outcome from the Gaussian with mean $\langle w, x \rangle$ and variance 1 conditional on $S$, using the algorithm of Lemma E.12. We prove that $Q'$ is close in total variation distance to $Q$, by relying on the guarantees of Lemma E.12.

These two results (see Lemma E.27) allow us to control the bias of our gradient sampler.

To analyze our algorithm, we will first bound its runtime with high probability, and then we will control the bias of its samples. Clearly, the runtime is dominated by the call to the algorithm of Lemma E.12, and by the guarantee of the lemma

it suffices to lower bound the survival probability $p(\widetilde{x}, w; S)$ of the sample $\widetilde{x}$ drawn by Algorithm 4. As we show in the following Lemma E.13, this probability might be exponentially small, however this is not an issue since the algorithm in Lemma E.12 runs in time polylogarithmic in the inverse survival probability.

**Lemma E.13.** *Suppose that Assumptions 1 to 4 hold. Let $S \subseteq \mathbb{R}$ be a set satisfying the guarantees of Theorem E.1 for some $\varepsilon < \alpha$. Let $\mathcal{P}$ be the projection set defined in Equation (7) for some $\widehat{w} \in \mathbb{R}^d$, and suppose that $w^\star \in \mathcal{P}$. Then, for any $w \in \mathcal{P}$ and any $\eta \in (0, 1)$, if $x \sim \mathcal{D}_{\mathrm{obs}}$, it holds with probability at least $1 - \eta$ that*

$$p(x, w; S) \geq \exp\left\{-O\left(\left(1 + \|w^\star\|_2^2\right) \cdot \frac{\Lambda^{12}\sigma^2}{\rho^8\alpha^6(\alpha - \varepsilon)^2} \cdot \left(d + \log\frac{1}{\eta}\right)\right)\right\}.$$

The proof of Lemma E.13 is given in Appendix E.3.5. Given this result, the following is immediate from Theorem E.1 and Lemmas E.12 and E.13.

**Corollary E.14.** *Suppose that Assumptions 1 to 5 hold. Let $S \subset R$ be a set satisfying the guarantees of Theorem E.1 for some $\varepsilon < \alpha/2$. Finally, let $\mathcal{P}$ be the projection set defined in Equation (7) for some $\widehat{w} \in \mathbb{R}^d$, and let $w \in \mathcal{P}$. Then, for any $\eta \in (0, 1)$ with probability at least $1 - \eta$, a call to Algorithm 4 with input $w$, $S$, and some $\zeta > 0$ terminates in time*

$$\mathrm{poly}\left(\|w^\star\|_2, d, \log\frac{1}{\zeta}, k, \frac{1}{\varepsilon}, \log\frac{1}{\eta}\right)^{\mathrm{poly}\left(\Lambda, \sigma, \frac{1}{\rho}, \frac{1}{\alpha}, \|\mu\|_2\right)},$$

*where $\mu := \mathbb{E}_{\mathcal{D}} x$.*

Hence, we have shown a high-probability bound on the runtime of Algorithm 4, and we use Corollary E.14 for an appropriate choice of the parameter $\eta$ to obtain our desired results. Recall, however, that Algorithm 4 is a *biased* gradient sampler; indeed, it is easy to see that in general $\mathbb{E}[g(w)] \neq \boldsymbol{\nabla}\mathscr{L}_S(w)$, where $g(w)$ is the output of the algorithm for input $w$. Nevertheless, the following Lemma E.15 establishes that the bias $\|\mathbb{E}[g(w)] - \boldsymbol{\nabla}\mathscr{L}_S(w)\|_2$ is bounded, and can be controlled by setting the set learning accuracy $\varepsilon$ to be appropriately small. The proof of Lemma E.15 is given in Appendix E.3.5.

**Lemma E.15.** *Suppose that Assumptions 1 to 4 hold. Let $S \subseteq \mathbb{R}$ be a set satisfying the guarantees of Theorem E.1 for some $\varepsilon < \alpha$. Then, for any $w \in \mathbb{R}^d$ and $\zeta < \sqrt{\varepsilon}$, if $g(w)$ is the output of a call to Algorithm 4 with input $w$, $S$, $\zeta$, it holds that*

$$\|\mathbb{E}[g(w)] - \boldsymbol{\nabla}\mathscr{L}_S(w)\|_2 \leq \widetilde{O}\left(\left(1 + \|w^\star\|_2\right) \cdot \left(\sigma^2 + \|\mu\|_2^2\right) \cdot \frac{\Lambda^2}{\alpha^{5/4}} \cdot \sqrt{d} \cdot \varepsilon^{1/4}\right)$$

*where $\mu := \mathbb{E}_{\mathcal{D}} x$.*

As we will see shortly, the convergence guarantees of PSGD essentially generalize even when our gradient samples have bounded bias, and so the above result suffices for our purposes. In order to obtain convergence guarantees for the PSGD algorithm, the last property we require from our gradient sampler is that the second moment of the (biased) stochastic gradients is bounded by a constant at each step of the execution. This is established in the following Lemma E.16, whose proof is given in Appendix E.3.5.

**Lemma E.16.** *Suppose that Assumptions 1 to 4 hold. Let $S \subseteq \mathbb{R}$ be a set satisfying the guarantees of Theorem E.1 for some $\varepsilon < \alpha^2/4$. Also, let $\mathcal{P}$ be the projection set defined in Equation (7) for some $\widehat{w} \in \mathbb{R}^d$, and suppose that $w^\star \in \mathcal{P}$. Then, for any $w \in \mathcal{P}$ and $\zeta < \sqrt{\varepsilon}$, if $g(w)$ is the output of a call to Algorithm 4 with input $w$, $S$, $\zeta$, it holds that*

$$\mathbb{E}\left[\|g(w)\|_2^2 \mid w\right] \leq \widetilde{O}\left(\left(1 + \|w^\star\|_2^2\right) \cdot \left(\|\mu\|_2^4 + \sigma^4\right) \cdot \frac{M^4\Lambda^4 d}{\rho^4\alpha^2}\right)$$

*where $\mu := \mathbb{E}_{\mathcal{D}} x$.*

### E.2.6. ANALYSIS OF PSGD WITH BIASED GRADIENTS

Having established the desired properties of our gradient sampler, we can now turn to analyzing the convergence of PSGD with biased gradients, given in Algorithm 5.

---

**Algorithm 5** Projected Stochastic Gradient Descent

---

**Require:** Projection set $\mathcal{P} \subset \mathbb{R}^d$, initialization $w^{(0)} \in \mathcal{P}$, set $S \subseteq \mathbb{R}$, parameters $T, \varepsilon$
**Ensure:** Sample access to the truncated regression model

$\quad$ Set $\kappa := e^{-\widetilde{O}\left(\frac{M^4 \Lambda^4}{\rho^6 \alpha^3}\right)}$
$\quad$ **for** $t \in [T]$ **do**
$\quad\quad$ Set $\eta_t := \frac{1}{\kappa \cdot t}$
$\quad\quad$ Let $g^{(t)}$ be the output of Algorithm 4 with input $w^{(t-1)}$, $S$ and $\zeta := \frac{\sqrt{\varepsilon}}{2}$
$\quad\quad$ Update $w^{(t)} := \Pi_{\mathcal{P}}\left(w^{(t-1)} - \eta_t \cdot g^{(t)}\right)$
$\quad$ **end for**
$\quad$ **Return** $\frac{1}{T} \sum_{t \in [T]} w^{(t)}$

---

The following guarantee follows directly by the proof of Lemma 6 of (Cherapanamjeri et al., 2023).

**Lemma E.17.** *Let $f \colon \mathbb{R}^d \to \mathbb{R}$ be a convex function, $\mathcal{P} \subset \mathbb{R}^d$ be a convex set with diameter $D$, and fix some initial point $w^{(0)} \in \mathcal{P}$. Let $w^{(1)}, \ldots, w^{(T)}$ be the iterates generated by running PSGD (Algorithm 5) for $T$ steps, using gradient estimates $g^{(1)}, \ldots, g^{(T)}$. Suppose that the following hold for all $t \in [T]$:*

1. **Strong Convexity:** *$f$ is $\kappa$-strongly convex over $\mathcal{P}$*

2. **Bounded Gradient Second Moment:** $\mathbb{E}\left[\left\|g(w^{(t)})\right\|_2^2 \mid w^{(t-1)}\right] \leq \mu^2$

3. **Bounded Gradient Bias:** $\left\|\mathbb{E}\left[g(w^{(t)}) \mid w^{(t-1)}\right] - \nabla \mathscr{L}_S(w^{(t)})\right\|_2 \leq \beta$

*Then, the average iterate $\overline{w} := \frac{1}{T} \sum_{t \in [T]} w^{(t)}$ satisfies*

$$\mathbb{E}\left[f(\overline{w})\right] - f(w_{\min}) \leq \frac{\mu^2}{\kappa T}(1 + \log T) + \beta D$$

*where $w_{\min} := \arg\min_{w \in \mathcal{P}} f(w)$.*

Using this convergence guarantee and our above analysis, it is straightforward to show that PSGD converges to the minimizer of the perturbed log-likelihood in $\mathcal{P}$ with constant probability.

**Corollary E.18.** *Suppose that Assumptions 1 to 5 hold, define $\mu := \mathbb{E}_{\mathcal{D}}\, x$, and let $\zeta > 0$. Let $S \subset \mathbb{R}$ be a set satisfying the guarantees of Theorem E.1 for some $\varepsilon$ such that*

$$\varepsilon \leq e^{-\widetilde{O}\left(\frac{M^4 \Lambda^4}{\rho^6 \alpha^3}\right)} \cdot \widetilde{O}\left(\frac{\zeta^8}{d^2 \cdot (\sigma + \|\mu\|_2)^{16} \cdot (1 + \|w^\star\|_2)^4}\right)$$

*so that $\varepsilon$ satisfies Equation (8). Finally, let $\mathcal{P}$ be the projection set defined in Equation (7) for some $\widehat{w} \in \mathbb{R}^d$, and suppose that $w^\star \in \mathcal{P}$. Consider running $T$ iterations of PSGD (Algorithm 5) over $\mathcal{P}$ with arbitrary initialization $w^{(0)} \in \mathcal{P}$, where $T$ satisfies*

$$T \geq e^{\widetilde{O}\left(\frac{M^4 \Lambda^4}{\rho^6 \alpha^3}\right)} \cdot \widetilde{O}\left(\frac{d \cdot (\sigma + \|\mu\|_2)^4 \cdot (1 + \|w^\star\|_2)^2}{\zeta^2}\right) \ .$$

*Let $w^{(1)}, \ldots, w^{(T)}$ be the iterates generated. Then, with probability at least $2/3$, the average iterate $\overline{w} := \frac{1}{T} \sum_{t \in [T]} w^{(t)}$ satisfies*

$$\|\overline{w} - w_{\min}\|_2 \leq \frac{\zeta}{9}$$

*where $w_{\min} := \arg\min_{w \in \mathcal{P}} \mathscr{L}_S(w)$. Moreover, for any $\eta \in (0, 1)$, with probability at least $1 - T \cdot \eta$, each iteration of PSGD terminates in time*

$$\text{poly}\left(\|w^\star\|_2, d, k, \frac{1}{\zeta}, \log \frac{1}{\eta}\right)^{\text{poly}\left(\Lambda, M, \sigma, \frac{1}{\rho}, \frac{1}{\alpha}, \|\mu\|_2\right)} \ .$$

*Proof.* Since $w^\star \in \mathcal{P}$ and $\mathcal{P}$ is a ball of diameter $O\left(\frac{\Lambda}{\rho^2 \alpha}\right)$, the projection step of each iteration can be implemented in time $\text{poly}\log\left(\|w^\star\|_2 + \frac{\Lambda}{\rho^2 \alpha}\right)$, and so the bottleneck for the runtime is the call to Algorithm 4 to obtain a gradient sample. Hence, the runtime bound follows immediately by applying Corollary E.14 and a union bound over all iterations, and then substituting our bound on $\varepsilon$.

To argue about the convergence guarantee, we will rely on Lemma E.17, and so we begin by checking the assumptions of the lemma: assumption (1) follows from Corollary E.9 for $\kappa := e^{-\widetilde{O}\left(\frac{M^4 \Lambda^4}{\rho^6 \alpha^3}\right)}$, assumption (2) follows from Lemma E.16, while assumption (3) follows from Lemma E.15. Since the projection set $\mathcal{P}$ has diameter $O\left(\frac{\Lambda}{\rho^2 \alpha}\right)$, by Lemma E.17 we obtain

$$\mathbb{E}\left[f(\overline{w})\right] - f(w_{\min}) \le \widetilde{O}\left(\sigma^4 + \|\mu\|_2^4\right) \cdot e^{\widetilde{O}\left(\frac{M^4 \Lambda^4}{\rho^6 \alpha^3}\right)} \cdot \left(\frac{1 + \log T}{T} \cdot d \cdot \left(1 + \|w^\star\|_2^2\right) + \varepsilon^{1/4} \cdot d^{1/2}\left(1 + \|w^\star\|_2\right)\right)$$

Applying Markov's inequality and hiding constants in the big-O notation, it follows that the above bound holds without the expectation, with probability at least $2/3$. Since $\mathscr{L}_S$ is $\kappa$-strongly convex over $\mathcal{P}$ and $w_{\min}$ is its minimizer, standard properties imply

$$f(\overline{w}) - f(w_{\min}) \ge \frac{\kappa}{2} \|\overline{w} - w_{\min}\|_2^2$$

and the convergence guarantee follows by combining the above and substituting our bounds for $\varepsilon$ and $T$. $\qquad \square$

To convert this constant-probability guarantee into a high-probability one, it suffices to apply a standard boosting technique, similar to (Daskalakis et al., 2018). Namely, we run PSGD a total of $K := O(\log 1/\delta)$ independent times to obtain average iterates $\{\overline{w}_1, \ldots, \overline{w}_K\}$. The guarantee of Corollary E.18 along with a Chernoff bound imply that, with probability at least $1 - \delta$, at least $3/5$ of the points $\overline{w}_i$ satisfy $\|\overline{w}_i - w_{\min}\|_2 \le \zeta/3$. Conditioned on this event, we can return any point $\overline{w}_\ell$ such that $\|\overline{w}_\ell - \overline{w}_i\|_2 \le 2\zeta/3$ for at least $3/5$ of the points $\overline{w}_i$; by the triangle inequality and our conditioning event, any point satisfying $\|\overline{w}_i - w_{\min}\|_2 \le \zeta/3$ also satisfies this property, and so a desired point $\overline{w}_\ell$ always exists. We claim that this point $\overline{w}_\ell$ satisfies $\|\overline{w}_\ell - w_{\min}\|_2 \le \zeta$. Suppose otherwise; then the triangle inequality gives us that, for any $\overline{w}_i$, either $\|\overline{w}_i - w_{\min}\|_2 > \zeta/3$ or $\|\overline{w}_\ell - \overline{w}_i\|_2 > 2\zeta/3$. However, this is a contradiction, since at most $2/5$ of $\overline{w}_i$ can satisfy the first inequality (due to the conditioning event) and also at most $2/5$ of $\overline{w}_i$ can satisfy the second inequality by definition of $\overline{w}_\ell$.

In the above procedure, we run a total of $T \cdot O(\log 1/\delta)$ iterations of PSGD, each of which requires only a single sample from the truncated regression model in the call to the approximate gradient sampler (Algorithm 4). By a union bound, the runtime guarantee of Corollary E.18 holds for each of these iterations with probability at least $1 - T \cdot O(\log 1/\delta) \cdot \eta$, and so if $\delta$ is bounded away from 1 we can pick $\eta := \frac{\delta}{T \cdot O(\log 1/\delta)} < 1$ to make the total failure probability at most $1 - \delta/2$. Hence, after substituting the bound for $T$ from Corollary E.18, our above discussion can be summarized in the following statement.

**Corollary E.19.** *Suppose that the assumptions of Corollary E.18 hold, and let $\zeta > 0$, $\delta \in (0, 1/2)$ be positive constants. There exists an algorithm that takes as input the projection set $\mathcal{P}$, an arbitrary initialization $w^{(0)} \in \mathcal{P}$, the truncation set $S \subset \mathbb{R}$ and the constants $\delta, \zeta$, and outputs a point $\overline{w} \in \mathcal{P}$. The algorithm draws $N$ i.i.d. samples from the truncated regression model, where*

$$N = e^{\widetilde{O}\left(\frac{M^4 \Lambda^4}{\rho^6 \alpha^3}\right)} \cdot \widetilde{O}\left(\frac{d \cdot (\sigma + \|\mu\|_2)^4 \cdot (1 + \|w^\star\|_2)^2}{\zeta^2} \cdot \log \frac{1}{\delta}\right) \ .$$

*With probability at least $1 - \delta/2$, the output of the algorithm satisfies*

$$\|\overline{w} - w_{\min}\|_2 \le \zeta \ ,$$

*and furthermore the algorithm terminates in time*

$$\text{poly}\left(\|w^\star\|_2, d, k, \frac{1}{\zeta}, \log \frac{1}{\delta}\right)^{\text{poly}\left(\Lambda, M, \sigma, \frac{1}{\rho}, \frac{1}{\alpha}, \|\mu\|_2\right)} \ .$$

To conclude, note that by Lemma E.11, conditioned on the event that $w^\star \in \mathcal{P}$, we can ensure that $\|w_{\min} - w^\star\|_2 \le \zeta$ by setting

$$\varepsilon \le e^{-\widetilde{O}\left(\frac{M^4 \Lambda^4}{\rho^6 \alpha^3}\right)} \cdot \widetilde{O}\left(\frac{\zeta^6}{d^3 \cdot (\sigma + \|\mu\|_2)^6 \cdot (1 + \|w^\star\|_2)^6}\right) \ .$$

To ensure that $w^\star \in \mathcal{P}$ with probability at least $1 - \delta/2$, by Corollary E.9, it suffices to draw $n$ i.i.d. samples from the truncated regression model and define $\mathcal{P}$ as in Equation (7) with $\widehat{w}$ as in Equation (4), where $n$ satisfies Equation (6), that is

$$n = \widetilde{O}\left((1 + \|w^\star\|_2)^2 \cdot (\sigma + \|\mu\|_2)^4 \cdot \frac{\Lambda^4}{\rho^8 \alpha^2} \cdot d \cdot \log \frac{1}{\delta}\right) \ .$$

Hence, the sample complexity to find the initial point $\widehat{w}$ and define our projection set is dominated by the sample complexity of repeatedly running PSGD to approximate $w^\star$, as given in Corollary E.19. The proof of Theorem E.2 now follows by our above discussion and Corollary E.19.

### E.3. Missing Proofs for Parameter Estimation

#### E.3.1. PROOF OF LEMMA E.5 (GRADIENT AND HESSIAN OF $\mathscr{L}_S$)

In this section we prove the expressions for the gradient and Hessian of the perturbed log-likelihood given in Lemma E.5, which we restate below.

**Lemma E.5** (Gradient and Hessian of $\mathscr{L}_S$)**.** *For any measurable set $S \subseteq \mathbb{R}$, and any $w \in \mathbb{R}^d$*

$$\boldsymbol{\nabla}\mathscr{L}_S(w) = \mathop{\mathbb{E}}_{x \sim \mathcal{D}_{\mathrm{obs}}}\left[-\mathop{\mathbb{E}}_{y \sim \mathcal{N}(\langle w^\star, x \rangle, 1, S^\star \cap S)}[y \cdot x] + \mathop{\mathbb{E}}_{z \sim \mathcal{N}(\langle w, x \rangle, 1, S)}[z \cdot x]\right],$$

$$\boldsymbol{\nabla}^2\mathscr{L}_S(w) = \mathop{\mathbb{E}}_{x \sim \mathcal{D}_{\mathrm{obs}}}\left[\mathop{\mathrm{Var}}_{z \sim \mathcal{N}(\langle w, x \rangle, 1, S)}[z] \cdot xx^\top\right].$$

*Proof.* For any $w, x \in \mathbb{R}^d$ and any $S \subset \mathbb{R}$, define

$$L_S(w; x) := -\mathop{\mathbb{E}}_{y \sim \mathcal{N}(\langle w^\star, x \rangle, 1, S \cap S^\star)}[\log \mathcal{N}(y; \langle w, x \rangle, 1, S)]$$

so that $\mathscr{L}_S(w) = \mathbb{E}_{x \sim \mathcal{D}_{\mathrm{obs}}}[L_S(w; x)]$. It holds that

$$L_S(w; x) = -\mathop{\mathbb{E}}_{y \sim \mathcal{N}(\langle w^\star, x \rangle, 1, S^\star)}[\log \mathcal{N}(y; \langle w, x \rangle, 1, S) \mid y \in S]$$

$$= -\mathop{\mathbb{E}}_{y \sim \mathcal{N}(\langle w^\star, x \rangle, 1, S^\star)}\left[\log \frac{\mathcal{N}(y; \langle w, x \rangle, 1)}{\mathcal{N}(S; \langle w, x \rangle, 1)} \,\middle|\, y \in S\right] \ .$$

Fix some $x \in \mathbb{R}^d$, and let $\ell(w)$ be the expression inside the expectation. We have that

$$\ell(w) = \log \frac{\exp\left(\frac{-(y - \langle w, x \rangle)^2}{2}\right)}{\int_S \exp\left(-\frac{(z - \langle w, x \rangle)^2}{2}\right) \mathrm{d}z} = -\frac{(y - \langle w, x \rangle)^2}{2} - \log \int_S \exp\left(-\frac{(z - \langle w, x \rangle)^2}{2}\right) \mathrm{d}z$$

$$= -\frac{y^2}{2} + y \cdot \langle w, x \rangle - \log \int_S \exp\left(-\frac{z^2}{2} + z \cdot \langle w, x \rangle\right) \mathrm{d}z \ .$$

Taking the gradient we obtain

$$\boldsymbol{\nabla}\ell(w) = y \cdot x - \frac{\int_S z \exp\left(-\frac{z^2}{2} + z \cdot \langle w, x \rangle\right) \mathrm{d}z}{\int_S \exp\left(-\frac{z^2}{2} + z \cdot \langle w, x \rangle\right) \mathrm{d}z} \cdot x = y \cdot x - \mathop{\mathbb{E}}_{z \sim \mathcal{N}(\langle w, x \rangle, 1, S)}[z \cdot x] \ .$$

Substituting into the definition of $L_S(w)$ and exchanging expectation and differentiation

$$\boldsymbol{\nabla}L_S(w) = -\mathop{\mathbb{E}}_{y \sim \mathcal{N}(\langle w^\star, x \rangle, 1, S^\star)}[\boldsymbol{\nabla}\ell(w) \mid y \in S] = -\mathop{\mathbb{E}}_{y \sim \mathcal{N}(\langle w^\star, x \rangle, 1, S \cap S^\star)}[y \cdot x] + \mathop{\mathbb{E}}_{z \sim \mathcal{N}(\langle w, x \rangle, 1, S)}[z \cdot x] \ .$$

The desired expression for $\boldsymbol{\nabla}\mathscr{L}_S(w)$ follows directly. For the Hessian, note that

$$\boldsymbol{\nabla}_w\left(\mathop{\mathbb{E}}_{z \sim \mathcal{N}(\langle w, x \rangle, 1, S)}[z]\right) = \boldsymbol{\nabla}_w\left(\frac{\int_S z \exp\left(-\frac{z^2}{2} + z \cdot \langle w, x \rangle\right) \mathrm{d}z}{\int_S \exp\left(-\frac{z^2}{2} + z \cdot \langle w, x \rangle\right) \mathrm{d}z}\right)$$

$$= \frac{\int_S z^2 \exp\left(-\frac{z^2}{2} + z \cdot \langle w, x \rangle\right) \mathrm{d}z}{\int_S \exp\left(-\frac{z^2}{2} + z \cdot \langle w, x \rangle\right) \mathrm{d}z} \cdot x - \left(\frac{\int_S z \exp\left(-\frac{z^2}{2} + z \cdot \langle w, x \rangle\right) \mathrm{d}z}{\int_S \exp\left(-\frac{z^2}{2} + z \cdot \langle w, x \rangle\right) \mathrm{d}z}\right)^2 \cdot x$$

$$= \left(\underset{z \sim \mathcal{N}(\langle w, x \rangle, 1, S)}{\mathbb{E}}\left[z^2\right] - \left(\underset{z \sim \mathcal{N}(\langle w, x \rangle, 1, S)}{\mathbb{E}}\left[z\right]\right)^2\right) \cdot x = \underset{z \sim \mathcal{N}(\langle w, x \rangle, 1, S)}{\mathrm{Var}}\left[z\right] \cdot x.$$

Therefore, by our previous expression for $\nabla L_S(w)$ we have

$$\nabla^2 L_S(w) = \nabla_w \left(\underset{z \sim \mathcal{N}(\langle w, x \rangle, 1, S)}{\mathbb{E}}\left[z\right]\right) x^\top = \underset{z \sim \mathcal{N}(\langle w, x \rangle, 1, S)}{\mathrm{Var}}\left[z\right] \cdot xx^\top.$$

Again, the desired expression for $\nabla^2 \mathscr{L}_S(w)$ follows directly. $\qquad\square$

### E.3.2. PROOF OF LEMMA E.6

We continue by establishing the lower bound on the eigenvalues of $\nabla^2 \mathscr{L}_S$ given by Lemma E.6.

**Lemma E.6.** *Suppose Assumptions 1, 3 and 4 hold. For any $w \in \mathbb{R}^d$ and any set $S \subseteq \mathbb{R}$ satisfying Equation (2) for some $\varepsilon > 0$, it holds that*

$$\nabla^2 \mathscr{L}_S(w) \succeq \Omega\left(\frac{\alpha^5 \rho^{10}}{\Lambda^8} \cdot e^{-\frac{10M^4}{\rho^2 \alpha} \|w - w^\star\|_2^2} - \frac{M^{4/3} \Lambda^{2/3} \varepsilon^{1/3}}{\alpha^{2/3}}\right) \cdot I.$$

*Proof.* By Lemmas D.2 and D.3, we have

$$\underset{z \sim \mathcal{N}(\langle w, x \rangle, 1, S)}{\mathrm{Var}}\left[z\right] \geq \frac{p\left(x, w; S\right)^2}{12} \quad \text{and} \quad p(x, w, S) \geq p(x, w^\star, S)^2 \cdot \exp\left(-\langle w - w^\star, x \rangle^2 - 2\right).$$

Let $v$ be an arbitrary unit vector, and denote $p_S(x) := p(x, w^\star; S)$ for brevity. By applying the above two inequalities to our expression for the Hessian in Lemma E.5, we obtain that

$$v^\top \nabla^2 \mathscr{L}_S(w) v \gtrsim \underset{x \sim \mathcal{D}_{\mathrm{obs}}}{\mathbb{E}}\left[p_S(x)^4 \cdot e^{-2\langle w - w^\star, x \rangle^2} \cdot \langle x, v \rangle^2\right].$$

To lower bound the above, observe that $p_S(x) = p_{S^\star \cup S}(x) - p_{S^\star \setminus S}(x) \geq p_{S^\star}(x) - p_{S^\star \setminus S}(x) \geq 0$, so:

$$v^\top \nabla^2 \mathscr{L}_S(w) v \gtrsim \underset{x \sim \mathcal{D}_{\mathrm{obs}}}{\mathbb{E}}\left[\left(p_{S^\star}(x) - p_{S^\star \setminus S}(x)\right)^4 \cdot e^{-2\langle w - w^\star, x \rangle^2} \cdot \langle x, v \rangle^2\right]$$

$$\geq \underset{x \sim \mathcal{D}_{\mathrm{obs}}}{\mathbb{E}}\left[p_{S^\star}(x)^4 \cdot e^{-2\langle w - w^\star, x \rangle^2} \cdot \langle x, v \rangle^2\right] - 4 \underset{x \sim \mathcal{D}_{\mathrm{obs}}}{\mathbb{E}}\left[p_{S^\star}(x)^3 p_{S^\star \setminus S}(x) \cdot e^{-2\langle w - w^\star, x \rangle^2} \cdot \langle x, v \rangle^2\right]$$

$$\tag{10}$$

where we used the inequality $(a - b)^4 \geq a^4 - 4a^3 b$. We will bound each of the above expectations separately. To bound the first expectation, note that

$$\underset{\mathcal{D}_{\mathrm{obs}}}{\mathbb{E}}\left[p_{S^\star}(x)^4 \cdot e^{-2\langle w - w^\star, x \rangle^2} \cdot \langle x, v \rangle^2\right]$$

$$\geq \underset{\mathcal{D}}{\mathbb{E}}\left[p_{S^\star}(x)^5 \cdot e^{-2\langle w - w^\star, x \rangle^2} \cdot \langle x, v \rangle^2\right]$$

$$\geq \frac{1}{\mathbb{E}_{\mathcal{D}}\left[e^{-2\langle w - w^\star, x \rangle^2} \cdot \langle x, v \rangle^2\right]^4} \cdot \left(\underset{\mathcal{D}}{\mathbb{E}}\left[p_{S^\star}(x) \cdot e^{-2\langle w - w^\star, x \rangle^2} \cdot \langle x, v \rangle^2\right]\right)^5$$

$$= \frac{1}{\mathbb{E}_{\mathcal{D}}\left[e^{-2\langle w - w^\star, x \rangle^2} \cdot \langle x, v \rangle^2\right]^4} \cdot \left(\underset{\mathcal{D}_{\mathrm{obs}}}{\mathbb{E}}\left[e^{-2\langle w - w^\star, x \rangle^2} \cdot \langle x, v \rangle^2\right] \cdot \underset{\mathcal{D}}{\mathbb{E}}\left[p_{S^\star}(x)\right]\right)^5$$

$$\geq \frac{\alpha^5}{\Lambda^8} \cdot \underset{\mathcal{D}_{\mathrm{obs}}}{\mathbb{E}}\left[e^{-2\langle w - w^\star, x \rangle^2} \cdot \langle x, v \rangle^2\right]^5$$

where the second line is by Fact F.1, the third by Hölder's inequality, the fourth by substituting Equation (3), and the last by Assumptions 1 and 4. We can complete the bound by Jensen's inequality

$$
\mathbb{E}_{\mathcal{D}_{\mathrm{obs}}} \left[ e^{-2\langle w-w^\star, x\rangle^2} \cdot \langle x, v\rangle^2 \right] \geq \mathbb{E}_{\mathcal{D}_{\mathrm{obs}}} \left[ \langle x, v\rangle^2 \right] \cdot \exp\left( -\frac{2\,\mathbb{E}_{\mathcal{D}_{\mathrm{obs}}}\left[ \langle w-w^\star, x\rangle^2 \cdot \langle x, v\rangle^2 \right]}{\mathbb{E}_{\mathcal{D}_{\mathrm{obs}}}\left[ \langle x, v\rangle^2 \right]} \right) \geq \rho^2 e^{-\frac{2M^4}{\rho^2 \alpha}\|w-w^\star\|_2^2}
$$

where in the last step we used the Cauchy–Schwarz inequality along with Assumptions 3 and 4 and Fact F.1. So overall, we obtain

$$
\mathbb{E}_{\mathcal{D}_{\mathrm{obs}}} \left[ p_{S^\star}(x)^4 \cdot e^{-2\langle w-w^\star, x\rangle^2} \cdot \langle x, v\rangle^2 \right] \geq \frac{\alpha^5 \rho^{10}}{\Lambda^8} \cdot e^{-\frac{10M^4}{\rho^2 \alpha}\|w-w^\star\|_2^2} .
$$

To bound the second expectation appearing in Equation (10), let $\gamma := \left( \frac{M^4 \varepsilon \alpha}{4\Lambda^4} \right)^{1/3}$ and observe that

$$
\begin{aligned}
\mathbb{E}_{x\sim\mathcal{D}_{\mathrm{obs}}} & \left[ p_{S^\star}(x)^3 p_{S^\star\setminus S}(x) \cdot e^{-2\langle w-w^\star, x\rangle^2} \cdot \langle x, v\rangle^2 \right] \\
& \leq \mathbb{E}_{\mathcal{D}_{\mathrm{obs}}} \left[ p_{S^\star\setminus S}(x) \cdot e^{-2\langle w-w^\star, x\rangle^2} \cdot \langle x, v\rangle^2 \right] \\
& \leq \mathbb{E}_{\mathcal{D}_{\mathrm{obs}}} \left[ p_{S^\star\triangle S}(x) \cdot \langle x, v\rangle^2 \right] \\
& = \mathbb{E}_{\mathcal{D}_{\mathrm{obs}}} \left[ p_{S^\star\triangle S}(x) \cdot \langle x, v\rangle^2 \cdot \mathbb{1}\left\{ p_{S^\star\triangle S}(x) \leq \gamma \right\} \right] + \mathbb{E}_{\mathcal{D}_{\mathrm{obs}}} \left[ p_{S^\star\triangle S}(x) \cdot \langle x, v\rangle^2 \cdot \mathbb{1}\left\{ p_{S^\star\triangle S}(x) \geq \gamma \right\} \right] \\
& \leq \gamma \cdot \mathbb{E}_{\mathcal{D}_{\mathrm{obs}}} \left[ \langle x, v\rangle^2 \right] + \mathbb{E}_{\mathcal{D}_{\mathrm{obs}}} \left[ \langle x, v\rangle^2 \cdot \mathbb{1}\left\{ p_{S^\star\triangle S}(x) \geq \gamma \right\} \right] \\
& \leq \gamma \cdot \mathbb{E}_{\mathcal{D}_{\mathrm{obs}}} \left[ \langle x, v\rangle^2 \right] + \sqrt{\mathbb{E}_{\mathcal{D}_{\mathrm{obs}}}\left[ \langle x, v\rangle^4 \right] \mathbb{E}_{\mathcal{D}_{\mathrm{obs}}}\left[ \mathbb{1}\left\{ p_{S^\star\triangle S}(x) \geq \gamma \right\} \right]} \\
& \leq \gamma \cdot \mathbb{E}_{\mathcal{D}_{\mathrm{obs}}} \left[ \langle x, v\rangle^2 \right] + \sqrt{\mathbb{E}_{\mathcal{D}_{\mathrm{obs}}}\left[ \langle x, v\rangle^4 \right] \cdot \frac{\mathbb{E}_{\mathcal{D}_{\mathrm{obs}}}[p_{S^\star\triangle S}(x)]}{\gamma}} \\
& \leq \frac{\gamma \cdot \Lambda^2}{\alpha} + M^2 \sqrt{\frac{\varepsilon}{\gamma\alpha}} .
\end{aligned}
$$

The first five lines follow by trivial bounds. The third-to-last is by the Cauchy–Schwarz inequality, the second-to-last is by Markov's inequality, and the last line is by Assumption 4 and Fact F.1 and the fact that $S$ satisfies Equation (2). Finally, plugging our last two displays into Equation (10) and substituting our choice of $\gamma$, we obtain that

$$
v^\top \boldsymbol{\nabla}^2 \mathscr{L}_S(w) v \gtrsim \frac{\alpha^5 \rho^{10}}{\Lambda^8} \cdot e^{-\frac{10M^4}{\rho^2 \alpha}\|w-w^\star\|_2^2} - \frac{M^{4/3}\Lambda^{2/3}\varepsilon^{1/3}}{\alpha^{2/3}} .
$$

$\square$

**Remark E.20** (Improving exponential dependencies). In the case where the truncation set $S^\star$ is known (hence $\varepsilon = 0$), our strong convexity constant has exponential dependence on the survival probablitity $\alpha$ and on the moment bounds $\rho, M$. This is different compared to (Daskalakis et al., 2019), who achieve a polynomial dependence. The technical reason for this is that (Daskalakis et al., 2019) show strong convexity over a carefully designed projection set, such that all $w$ in that projection set satisfy a useful algebraic property; our projection set $\mathcal{P}$ (see Corollary E.9) is much simpler, and only bounds the distance $\|w-w^\star\|_2$ of $w \in \mathcal{P}$. The reason for using this simpler projection set as opposed to the one in (Daskalakis et al., 2019) is that we also correspondingly make a simpler assumption on the survival probability (namely, Assumption 1) in contrast to their (arguably) more complicated assumption which, in turn, also enabled them to obtain a better dependency on $\rho$ and $M$. We believe that replacing Assumption 3 with their assumption and $\mathcal{P}$ with their projection set in our analysis would yield polynomial dependence on $\rho$ and $M$.

E.3.3. EFFICIENT WARM-START

In this section, we prove the results given in Appendix E.2.3. Recall that our objective is to show that $\widehat{w}$ is a constant distance away from $w^\star$, where $\widehat{w}$ is defined in Equation (4) as

$$\widehat{w} := \left( \frac{1}{n} \sum_i x^{(i)} x^{(i)\top} \right)^{-1} \cdot \left( \frac{1}{n} \sum_i y^{(i)} \cdot x^{(i)} \right) .$$

In Appendix E.2.3, we showed the following guarantee for the infinite-sample setting.

**Lemma E.7.** *Suppose that Assumptions 1, 3 and 4 hold, and define*

$$\widetilde{w} := \left( \underset{x \sim \mathcal{D}_{\mathrm{obs}}}{\mathbb{E}} \left[ xx^\top \right] \right)^{-1} \cdot \underset{x \sim \mathcal{D}_{\mathrm{obs}}}{\mathbb{E}} \left[ \underset{y \sim \mathcal{N}(\langle w^\star, x \rangle, 1, S^\star)}{\mathbb{E}} [y] \cdot x \right] . \tag{5}$$

*Then it holds that*

$$\| \widetilde{w} - w^\star \|_2 \leq \frac{3\Lambda}{2\rho^2 \alpha} .$$

To convert this into a finite-sample guarantee for $\widehat{w}$, we will show that the two terms appearing in the right-hand side of Equation (4) concentrate rapidly to their expected values. We begin by establishing concentration of the vector $\frac{1}{n} \sum_i y^{(i)} \cdot x^{(i)}$.

**Lemma E.21.** *Let $\left\{ \left( x^{(i)}, y^{(i)} \right) \right\}_{i \in [n]}$ be i.i.d. samples such that the marginal of $x^{(i)}$ is $\mathcal{D}_{\mathrm{obs}}$ and the marginal of $y^{(i)}$ conditioned on $x^{(i)}$ is $\mathcal{N}(\langle w^\star, x^{(i)} \rangle, 1, S^\star)$. Suppose that Assumptions 1 and 2 hold, and furthermore*

$$n \geq \widetilde{\Omega} \left( (1 + \| w^\star \|_2)^2 \cdot \left( \sigma^2 \log \frac{1}{\alpha} + \| \mu \|_2^2 \right)^2 \cdot \frac{d}{\zeta^2} \log \frac{1}{\delta} \right)$$

*where $\mu := \mathbb{E}_{\mathcal{D}} x$. Then, with probability at least $1 - \delta$, it holds that*

$$\left\| \frac{1}{n} \sum_i y^{(i)} \cdot x^{(i)} - \mathbb{E} \left[ y \cdot x \right] \right\|_2 \leq \zeta .$$

*Proof.* By Lemma F.8 and the Bernstein inequality for subexponential random variables (see Theorem 2.9.1 of (Vershynin, 2018)), it follows that, for any unit vector $v$ and any $t > 0$

$$\Pr \left( \left| \left\langle v, \frac{1}{n} \sum_i y^{(i)} \cdot x^{(i)} - \mathbb{E} \left[ y \cdot x \right] \right\rangle \right| \geq t \right) \leq 2 \exp \left( -c \min \left( \frac{nt^2}{K^2}, \frac{nt}{K} \right) \right)$$

where $K = \Theta \left( (1 + \| w^\star \|_2) \cdot \left( \sigma^2 \log \frac{1}{\alpha} + \| \mu \|_2^2 \right) \right) \geq 1$ and $c$ is an absolute constant. Therefore, for $t = \frac{\zeta}{\sqrt{d}}$, we have

$$\Pr \left( \left| \left\langle v, \frac{1}{n} \sum_i y^{(i)} \cdot x^{(i)} - \mathbb{E} \left[ y \cdot x \right] \right\rangle \right| \geq \frac{\zeta}{\sqrt{d}} \right) \leq 2 \exp \left( -c \min \left( \frac{n\zeta^2}{K^2 d}, \frac{n\zeta}{K\sqrt{d}} \right) \right) \leq \frac{\delta}{d}$$

since $n = \Omega \left( \frac{K^2 d}{\zeta^2} \log \frac{d}{\delta} \right)$. The lemma now follows by choosing any orthonormal basis in $\mathbb{R}^d$ and requiring that the above event does not hold for each vector in that basis, which by a union bound happens with probability at least $1 - \delta$. $\qquad \square$

Next, we show concentration of the matrix $\frac{1}{n} \sum_i x^{(i)} x^{(i)\top}$, in operator norm.

**Lemma E.22.** *Let $\left\{ x^{(i)} \right\}_{i \in [n]}$ be i.i.d. samples from $\mathcal{D}_{\mathrm{obs}}$. Suppose that Assumptions 1, 2 and 4 hold, and*

$$n \geq \widetilde{\Omega} \left( \frac{\Lambda^4}{\rho^4 \alpha^2} (\sigma + \| \mu \|_2)^4 \frac{\left( d + \log \frac{1}{\delta} \right)}{\zeta^2} \right) .$$

*Then, with probability at least $1 - \delta$, it holds that*

$$\left\| \frac{1}{n} \sum_i x^{(i)} x^{(i)\top} - \mathbb{E}\left[ xx^\top \right] \right\|_{\text{op}} \leq \zeta \, .$$

*Proof.* For any vector $v$, if $x \sim \mathcal{D}_{\text{obs}}$, then by Lemma F.5 and Assumption 3 we have

$$\frac{\| \langle x, v \rangle \|_{\psi_2}}{\sqrt{\mathbb{E}_{\mathcal{D}_{\text{obs}}} \langle x, v \rangle^2}} \leq \frac{1}{\rho} \cdot O\left( \sqrt{\sigma^2 \log \frac{1}{\alpha} + \|\mu\|_2^2} \right) =: K \, .$$

Therefore, by Theorem 4.7.1 and Remark 4.7.3 of (Vershynin, 2018), there exists some absolute constant $C$ such that, with probability at least $1 - \delta$

$$\left\| \frac{1}{n} \sum_i x^{(i)} x^{(i)\top} - \mathop{\mathbb{E}}_{\mathcal{D}_{\text{obs}}}\left[ xx^\top \right] \right\|_{\text{op}} \leq CK^2 \left( \sqrt{\frac{d + \log \frac{2}{\delta}}{n}} + \frac{d + \log \frac{2}{\delta}}{n} \right) \cdot \left\| \mathop{\mathbb{E}}_{\mathcal{D}_{\text{obs}}}\left[ xx^\top \right] \right\|_{\text{op}} \, .$$

Since $\left\| \mathbb{E}_{\mathcal{D}_{\text{obs}}}\left[ xx^\top \right] \right\|_{\text{op}} \leq \frac{\Lambda^2}{\alpha}$ by Assumption 4 and Fact F.1, the right-hand side of the above becomes equal to $\zeta$ by setting $n \geq O\left( \frac{K^4 \Lambda^4 (d + \log \frac{1}{\delta})}{\alpha^2 \zeta^2} \right)$ and the lemma follows by the definition of $K$ above. $\qquad\square$

Lastly, we combine our above results to show that the finite-sample estimate $\widehat{w}$ is a constant distance away from the infinite-sample estimate $\widetilde{w}$, with high probability.

**Lemma E.23.** *Suppose that Assumptions 1 to 4 hold. Let $\left\{ \left( x^{(i)}, y^{(i)} \right) \right\}_{i \in [n]}$ be i.i.d. samples from the truncated regression model, and suppose that*

$$n \geq \widetilde{\Omega}\left( (1 + \|w^\star\|_2)^2 \cdot (\sigma + \|\mu\|_2)^4 \cdot \frac{\Lambda^4}{\rho^8 \alpha^2} \cdot d \cdot \log \frac{1}{\delta} \right)$$

*where $\mu := \mathbb{E}_{\mathcal{D}} x$. Let $\widehat{w}, \widetilde{w}$ be defined as in Equations (4) and (5). Then with probability at least $1 - \delta$*

$$\| \widehat{w} - \widetilde{w} \|_2 \leq \frac{9\Lambda}{2\rho^2 \alpha} \, .$$

*Proof.* For brevity, we denote

$$\Sigma := \mathop{\mathbb{E}}_{x \sim \mathcal{D}_{\text{obs}}}\left[ xx^\top \right], \quad \nu := \mathop{\mathbb{E}}_{x \sim \mathcal{D}_{\text{obs}}}\left[ \mathop{\mathbb{E}}_{y \sim \mathcal{N}(\langle w^\star, x \rangle, 1, S^\star)}[y] \cdot x \right]$$

and

$$\widehat{\Sigma} := \frac{1}{n} \sum_i x^{(i)} x^{(i)\top}, \quad \widehat{\nu} := \frac{1}{n} \sum_i y^{(i)} \cdot x^{(i)}$$

so that $\widehat{w} = \widehat{\Sigma}^{-1} \widehat{\nu}$ and $\widetilde{w} = \Sigma^{-1} \nu$. Given our lower bound for the number of samples $n$, we can apply a union bound over the conclusions of Lemmas E.21 and E.22 to obtain that, with probability $1 - \delta$

$$\left\| \Sigma - \widehat{\Sigma} \right\|_{\text{op}} \leq \frac{\rho^2}{2(1 + \|w^\star\|_2)} \quad \text{and} \quad \| \nu - \widehat{\nu} \|_2 \leq \rho^2 \, .$$

We condition on both of the above events holding. Letting $\lambda_d(A)$ denote the minimum eigenvalue of a symmetric matrix $A \in \mathbb{S}^{d \times d}$, the first event holding implies that $\left| \lambda_d(\widehat{\Sigma} - \Sigma) \right| \leq \frac{\rho^2}{2(1 + \|w^\star\|_2)}$, and therefore Weyl's inequality gives

$$\lambda_d(\widehat{\Sigma}) \geq \lambda_d(\Sigma) + \lambda_d(\widehat{\Sigma} - \Sigma) \geq \rho^2 \left( 1 - \frac{1}{2(1 + \|w^\star\|_2)} \right)$$

where we used the fact that $\Sigma \succeq \rho^2 \cdot I$ by Assumption 3. Now, observe that

$$
\begin{aligned}
\widehat{w} - \widetilde{w} &= \widehat{\Sigma}^{-1}\widehat{\nu} - \Sigma^{-1}\nu \\
&= \widehat{\Sigma}^{-1}\left[(\widehat{\nu} - \nu) + \left(\Sigma - \widehat{\Sigma}\right)\Sigma^{-1}\nu\right] \\
&= \widehat{\Sigma}^{-1}\left[(\widehat{\nu} - \nu) + \left(\Sigma - \widehat{\Sigma}\right)\widetilde{w}\right] .
\end{aligned}
$$

Applying the triangle inequality and $\left\|\widehat{\Sigma}^{-1}\right\|_{\mathrm{op}} = \frac{1}{\lambda_d(\widehat{\Sigma})}$ (since $\lambda_d(\widehat{\Sigma}) > 0$), we have that

$$
\|\widehat{w} - \widetilde{w}\|_2 \leq \frac{1}{\lambda_d(\widehat{\Sigma})}\left(\|\widehat{\nu} - \nu\|_2 + \left\|\Sigma - \widehat{\Sigma}\right\|_{\mathrm{op}}\|\widetilde{w}\|\right) .
$$

To bound the above, note that, by Lemma E.7, we have

$$
\|\widetilde{w}\|_2 \leq \|w^\star\|_2 + \frac{3\Lambda}{2\rho^2\alpha} .
$$

Therefore, combining all our above bounds, we finally obtain

$$
\begin{aligned}
\|\widehat{w} - \widetilde{w}\|_2 &\leq \frac{1}{\rho^2\left(1 - \frac{1}{2(1 + \|w^\star\|_2)}\right)}\left(\rho^2 + \frac{\rho^2\|w^\star\|_2}{2(1 + \|w^\star\|_2)} + \frac{3\Lambda}{4\alpha(1 + \|w^\star\|_2)}\right) \\
&\leq \frac{2}{\rho^2}\left(\rho^2 + \frac{\rho^2}{2} + \frac{3\Lambda}{4\alpha}\right) \\
&= \frac{3\Lambda}{2\rho^2\alpha} + 3
\end{aligned}
$$

and the lemma follows. $\qquad \square$

Our final result of this section now follows immediately.

**Lemma E.8.** *Suppose that Assumptions 1 to 4 hold. Let $\left\{\left(x^{(i)}, y^{(i)}\right)\right\}_{i \in [n]}$ be i.i.d. samples from the truncated regression model, and suppose that*

$$
n \geq \widetilde{\Omega}\left((1 + \|w^\star\|_2)^2 \cdot (\sigma + \|\mu\|_2)^4 \cdot \frac{\Lambda^4}{\rho^8\alpha^2} \cdot d \cdot \log\frac{1}{\delta}\right) \tag{6}
$$

*where $\mu := \mathbb{E}_{\mathcal{D}}\, x$. Let $\widehat{w}$ be defined as in Equation (4). Then with probability at least $1 - \delta$*

$$
\|\widehat{w} - w^\star\|_2 \leq \frac{6\Lambda}{\rho^2\alpha} .
$$

*Proof.* The result follows from Lemmas E.7 and E.23 and the triangle inequality. $\qquad \square$

### E.3.4. CLOSENESS OF $w_{\min}$ AND $w^\star$

In this section, we prove Lemma E.10, which, as we discussed in Appendix E.2.4, is the key technical result that allows us to control the distance between $w^\star$ and the minimizer $w_{\min}$ of $\mathscr{L}_S$ over the projection set $\mathcal{P}$. We begin with two auxiliary lemmas. First, we show a simple concentration inequality for truncated Gaussians in one dimension.

**Proposition E.24.** *Let $\left\{y^{(i)}\right\}_{i \in [n]}$ be $n$ i.i.d. samples from the truncated gaussian $\mathcal{N}(\nu, 1, T)$, for some $\nu \in \mathbb{R}$ and some measurable $T \subseteq \mathbb{R}$. Denote by $a := \mathcal{N}(T; \nu, 1)$ the mass that the untruncated gaussian places on $T$. Assume that*

$$
n \geq \Omega\left(\frac{1 + \log\frac{1}{a}}{\zeta^2} \cdot \sqrt{\log\frac{1}{\delta}}\right) .
$$

*Then with probability at least $1 - \delta$ it holds that*

$$
\left|\frac{1}{n}\sum_{i \in [n]} y^{(i)} - \mathbb{E}[y]\right| \leq \zeta .
$$

*Proof.* By Lemma F.7 and standard subgaussian concentration (see Proposition 2.6.1 of (Vershynin, 2018)), it follows that for any $\zeta > 0$

$$\Pr\left(\left|\frac{1}{n}\sum_{i\in[n]} y^{(i)} - \mathbb{E}[y]\right| \geq \zeta\right) \leq 2\exp\left(-\frac{cn\zeta^2}{1+\log 1/a}\right)$$

where $c > 0$ is some absolute constant. The claim follows by the lower bound on $n$. □

We now show a change-of-measure inequality between two truncated Gaussians with the same mean but different truncation sets, in the case where the truncation set of one Gaussian is a subset of the truncation set of the other.

**Proposition E.25** (Change-of-Measure between Truncated Gaussians)**.** *Let $\nu \in \mathbb{R}$, and let $S, T \subset \mathbb{R}$ be measurable sets, such that $S \subseteq T$. Define $a := \mathcal{N}(S; \nu, 1)$ and $b := \mathcal{N}(T; \nu, 1)$ the mass that the untruncated Gaussian $\mathcal{N}(\nu, 1)$ places on $S$ and $T$ respectively. Then for any event $\mathcal{E}$ it holds that*

$$\Pr_{z\sim\mathcal{N}(\nu,1,S)}(\mathcal{E}) \leq \frac{\mathcal{N}(T;\nu,1)}{\mathcal{N}(S;\nu,1)} \cdot \Pr_{z\sim\mathcal{N}(\nu,1,T)}(\mathcal{E}).$$

*Proof.* By definition of the truncated Gaussian we have

$$\begin{aligned}
\Pr_{z\sim\mathcal{N}(\nu,1,S)}(\mathcal{E}) &= \int \mathbb{1}_{\mathcal{E}}(z) \cdot \mathcal{N}(z;\nu,1,S)\,\mathrm{d}z \\
&= \int \mathbb{1}_{\mathcal{E}}(z) \cdot \frac{\mathcal{N}(z;\nu,1)\cdot\mathbb{1}\{z\in S\}}{\mathcal{N}(S;\nu,1)}\,\mathrm{d}z \\
&= \frac{\mathcal{N}(T;\nu,1)}{\mathcal{N}(S;\nu,1)} \int \mathbb{1}_{\mathcal{E}}(z) \cdot \frac{\mathcal{N}(z;\nu,1)\cdot\mathbb{1}\{z\in S\}}{\mathcal{N}(T;\nu,1)}\,\mathrm{d}z \\
&\leq \frac{\mathcal{N}(T;\nu,1)}{\mathcal{N}(S;\nu,1)} \int \mathbb{1}_{\mathcal{E}}(z) \cdot \frac{\mathcal{N}(z;\nu,1)\cdot\mathbb{1}\{z\in T\}}{\mathcal{N}(T;\nu,1)}\,\mathrm{d}z \\
&= \frac{\mathcal{N}(T;\nu,1)}{\mathcal{N}(S;\nu,1)} \int \mathbb{1}_{\mathcal{E}}(z) \cdot \mathcal{N}(z;\nu,1,T)\,\mathrm{d}z \\
&= \frac{\mathcal{N}(T;\nu,1)}{\mathcal{N}(S;\nu,1)} \cdot \Pr_{z\sim\mathcal{N}(\nu,1,T)}(\mathcal{E})
\end{aligned}$$

where the inequality is because $S \subseteq T$. □

Having established our helper lemmas, we now turn to showing the following key bound on the distance of the expectations of two truncated Gaussians with different truncation sets.

**Lemma E.26.** *Let $\nu \in \mathbb{R}$ and $S, T \subseteq \mathbb{R}$ be measurable sets such that $S \subseteq T$. Let $p_S := \mathcal{N}(S; \nu, 1)$ and $p_T := \mathcal{N}(T; \nu, 1)$ be the mass that the Gaussian $\mathcal{N}(\nu, 1)$ places on $S, T$ respectively. Then it holds that*

$$\left|\mathbb{E}_{y\sim\mathcal{N}(\nu,1,S)}[y] - \mathbb{E}_{z\sim\mathcal{N}(\nu,1,T)}[z]\right| \leq O\left(\sqrt{\left(1+\log\frac{1}{p_T}\right)\cdot\log\frac{p_T}{p_S}}\right).$$

*Proof.* Our argument is similar to that of Lemma B.4 of (Lee et al., 2024). Let $\zeta > 0$ and $\delta \in (0,1)$ be arbitrary, and let $\left\{y^{(i)}\right\}_{i\in[n]}$ be $n$ i.i.d. samples from the truncated Gaussian $\mathcal{N}(\nu, 1, S)$, where

$$n \geq \Omega\left(\left(1+\log\frac{1}{p_T}\right)\left(1+\log\frac{1}{p_S}\right)\frac{1}{\zeta^2}\log\frac{1}{\delta}\right).$$

Given Lemma F.7, standard sub-Gaussian concentration (see Proposition 2.6.1 of (Vershynin, 2018)) gives, for some absolute constant $c > 0$

$$\Pr\left(\left|\frac{1}{n}\sum_{i\in[n]} y^{(i)} - \mathbb{E}[y]\right| \leq \zeta\right) \geq 1 - 2\exp\left(-\frac{cn\zeta^2}{1+\log\frac{1}{p_S}}\right) \geq 1 - \frac{\delta}{2}$$

since $n \geq \Omega\left(\left(1 + \log\frac{1}{p_S}\right)\frac{1}{\zeta^2}\log\frac{1}{\delta}\right)$. Condition on this high-probability event. To bound the desired distance, note that an application of the triangle inequality yields

$$\left|\mathop{\mathbb{E}}_{y\sim\mathcal{N}(\nu,1,S)}[y] - \mathop{\mathbb{E}}_{z\sim\mathcal{N}(\nu,1,T)}[z]\right| \leq \left|\frac{1}{n}\sum_{i\in[n]}y^{(i)} - \mathop{\mathbb{E}}_{y\sim\mathcal{N}(\nu,1,T)}[y]\right| + \zeta.$$

So, to prove the claim, we focus on bounding this latter difference. Towards this, again by sub-Gaussian concentration we have, for any $t > 0$ and some absolute constant $c > 0$

$$\mathop{\Pr}_{y^{(1)},\dots,y^{(n)}\sim\mathcal{N}(\nu,1,T)}\left(\left|\frac{1}{n}\sum_i y^{(i)} - \mathop{\mathbb{E}}_{y\sim\mathcal{N}(\nu,1,T)}[y]\right| \geq t\right) \leq 2\exp\left(-\frac{cnt^2}{1+\log\frac{1}{p_T}}\right).$$

Since $S \subseteq T$ and $y^{(1)},\dots,y^{(n)}$ are independent, we can repeatedly apply Proposition E.25 to obtain

$$\mathop{\Pr}_{y^{(1)},\dots,y^{(n)}\sim\mathcal{N}(\nu,1,S)}\left(\left|\frac{1}{n}\sum_i y^{(i)} - \mathop{\mathbb{E}}_{y\sim\mathcal{N}(\nu,1,T)}[y]\right| \geq t\right) \leq 2\left(\frac{p_T}{p_S}\right)^n \cdot \exp\left(-\frac{cnt^2}{1+\log\frac{1}{p_T}}\right).$$

If we pick $t := \sqrt{\frac{1+\log\frac{1}{p_T}}{c}\cdot\log\frac{p_T}{p_S} + \zeta^2}$, the right hand side becomes $2\exp\left(-\frac{cn\zeta^2}{1+\log\frac{1}{p_T}}\right)$, which is at most $\delta/2$ since $n \geq \Omega\left(\left(1 + \log\frac{1}{p_T}\right)\frac{1}{\zeta^2}\log\frac{1}{\delta}\right)$. Therefore, with probability at least $1 - \delta/2$, it holds that

$$\left|\frac{1}{n}\sum_i y^{(i)} - \mathop{\mathbb{E}}_{y\sim\mathcal{N}(\nu,1,T)}[y]\right| \leq \sqrt{\frac{1+\log\frac{1}{p_T(x)}}{c}\cdot\log\frac{p_T(x)}{p_S(x)} + \zeta^2}.$$

Thus, by a union bound, with probability $1 - \delta$ both this event and our previous conditioning event hold, and combining the above yields

$$\left|\mathop{\mathbb{E}}_{y\sim\mathcal{N}(\nu,1,S)}[y] - \mathop{\mathbb{E}}_{z\sim\mathcal{N}(\nu,1,T)}[z]\right|^2 \leq O\left(\left(1+\log\frac{1}{p_T}\right)\cdot\log\frac{p_T}{p_S}\right) + 3\zeta^2.$$

Since this holds for any $\zeta > 0$ and $\delta \in (0,1)$, the lemma follows by taking the limit as $\zeta, \delta \to 0^+$. $\qquad\square$

Armed with the distance bound of Lemma E.26, we are ready to show the main result of this section.

**Lemma E.10** (Bound on Norm of Gradient). *Suppose that Assumptions 1, 2 and 4 hold, and let $S$ be a measurable set satisfying the guarantee of Theorem E.1, for some $\varepsilon > 0$ satisfying $\varepsilon < 1/8 \cdot \alpha^3$. Then*

$$\|\boldsymbol{\nabla}\mathscr{L}_S(w^\star)\|_2 \leq \widetilde{O}\left(\frac{\Lambda^2}{\alpha}\cdot(1+\|w^\star\|_2)\cdot(\|\mu\|_2+\sigma)\sqrt{d}\varepsilon^{1/6}\right)$$

*where $\mu := \mathbb{E}_{\mathcal{D}}\, x$.*

*Proof.* Let $p_T(x) := p(x, w^\star; T)$ for brevity. To bound the norm of the above, we define the following set of "good" points in $\mathbb{R}^d$

$$\mathcal{A} := \left\{\, x \in \mathbb{R}^d \,\middle|\, p_{S^\star}(x) \geq \alpha\varepsilon^{1/3} \quad\text{and}\quad p_{S^\star\triangle S}(x) \leq \varepsilon^{2/3} \,\right\}.$$

We begin by bounding the mass that $\mathcal{D}_{\text{obs}}$ places in $\mathcal{A}$. Note that, by Fact F.1

$$
\begin{aligned}
\mathop{\Pr}_{x\sim\mathcal{D}_{\text{obs}}}\left(p_{S^\star}(x) \leq \alpha\varepsilon^{1/3}\right) &= \mathop{\mathbb{E}}_{x\sim\mathcal{D}_{\text{obs}}}\left[\mathbb{1}\left\{p_{S^\star}(x) \leq \alpha\varepsilon^{1/3}\right\}\right] \\
&\leq \frac{1}{\alpha}\cdot\mathop{\mathbb{E}}_{x\sim\mathcal{D}}\left[p_{S^\star}(x)\cdot\mathbb{1}\left\{p_{S^\star}(x) \leq \alpha\varepsilon^{1/3}\right\}\right] \\
&\leq \varepsilon^{1/3}\cdot\mathop{\mathbb{E}}_{x\sim\mathcal{D}}\left[\mathbb{1}\left\{p_{S^\star}(x) \leq \alpha\varepsilon^{1/3}\right\}\right] \\
&\leq \varepsilon^{1/3}
\end{aligned}
$$

and furthermore Markov's inequality yields

$$\Pr_{x \sim \mathcal{D}_{\mathrm{obs}}} \left( p_{S^\star \triangle S}(x) \geq \varepsilon^{2/3} \right) \leq \frac{\mathbb{E}_{x \sim \mathcal{D}_{\mathrm{obs}}}[p_{S^\star \triangle S}(x)]}{\varepsilon^{2/3}} \leq \varepsilon^{1/3}$$

where we used the guarantee on $S$ by Theorem E.1. Therefore, by a union bound it follows that $\Pr_{x \sim \mathcal{D}_{\mathrm{obs}}}(x \notin \mathcal{A}) \leq 2\varepsilon^{1/3}$. Now, towards proving the claim, recall by Lemma E.5 that

$$\boldsymbol{\nabla}\mathcal{L}_S(w^\star) = \mathbb{E}_{x \sim \mathcal{D}_{\mathrm{obs}}} \left[ - \mathbb{E}_{y \sim \mathcal{N}(\langle w^\star, x\rangle, 1, S^\star \cap S)}[y \cdot x] + \mathbb{E}_{z \sim \mathcal{N}(\langle w^\star, x\rangle, 1, S)}[z \cdot x] \right].$$

An application of the law of total expectation and the triangle inequality then gives

$$\|\boldsymbol{\nabla}\mathcal{L}_S(w^\star)\|_2 \leq \left\| \mathbb{E}_{x \sim \mathcal{D}_{\mathrm{obs}}} \left[ \left( \mathbb{E}_{z \sim \mathcal{N}(\langle w^\star, x\rangle, 1, S)}[z \cdot x] - \mathbb{E}_{y \sim \mathcal{N}(\langle w^\star, x\rangle, 1, S^\star \cap S)}[y \cdot x] \right) \cdot \mathbb{1}\{x \in \mathcal{A}\} \right] \right\|_2$$
$$+ \left\| \mathbb{E}_{x \sim \mathcal{D}_{\mathrm{obs}}} \left[ \left( \mathbb{E}_{z \sim \mathcal{N}(\langle w^\star, x\rangle, 1, S)}[z \cdot x] - \mathbb{E}_{y \sim \mathcal{N}(\langle w^\star, x\rangle, 1, S^\star \cap S)}[y \cdot x] \right) \cdot \mathbb{1}\{x \notin \mathcal{A}\} \right] \right\|_2. \quad (11)$$

To bound the first norm above, note that by the triangle inequality and Lemma E.26 we have

$$\left\| \mathbb{E}_{x \sim \mathcal{D}_{\mathrm{obs}}} \left[ \left( \mathbb{E}_{z \sim \mathcal{N}(\langle w^\star, x\rangle, 1, S)}[z \cdot x] - \mathbb{E}_{y \sim \mathcal{N}(\langle w^\star, x\rangle, 1, S^\star \cap S)}[y \cdot x] \right) \cdot \mathbb{1}\{x \in \mathcal{A}\} \right] \right\|_2$$
$$\leq \mathbb{E}_{x \sim \mathcal{D}_{\mathrm{obs}}} \left[ \left| \mathbb{E}_{z \sim \mathcal{N}(\langle w^\star, x\rangle, 1, S)}[z] - \mathbb{E}_{y \sim \mathcal{N}(\langle w^\star, x\rangle, 1, S^\star \cap S)}[y] \right| \cdot \|x\|_2 \cdot \mathbb{1}\{x \in \mathcal{A}\} \right]$$
$$\lesssim \mathbb{E}_{x \sim \mathcal{D}_{\mathrm{obs}}} \left[ \sqrt{\left(1 + \log \frac{1}{p_S(x)}\right) \cdot \log \frac{p_S(x)}{p_{S \cap S^\star}(x)}} \cdot \|x\|_2 \cdot \mathbb{1}\{x \in \mathcal{A}\} \right].$$

An application of the Cauchy–Schwarz inequality, combined with Assumption 4, yields

$$\left\| \mathbb{E}_{x \sim \mathcal{D}_{\mathrm{obs}}} \left[ \left( \mathbb{E}_{z \sim \mathcal{N}(\langle w^\star, x\rangle, 1, S)}[z \cdot x] - \mathbb{E}_{y \sim \mathcal{N}(\langle w^\star, x\rangle, 1, S^\star \cap S)}[y \cdot x] \right) \cdot \mathbb{1}\{x \in \mathcal{A}\} \right] \right\|_2$$
$$\lesssim \Lambda\sqrt{d} \cdot \sqrt{\mathbb{E}_{x \sim \mathcal{D}_{\mathrm{obs}}} \left[ \left(1 + \log \frac{1}{p_S(x)}\right) \cdot \log \frac{p_S(x)}{p_{S \cap S^\star}(x)} \cdot \mathbb{1}\{x \in \mathcal{A}\} \right]}.$$

To bound the latter, observe that the following simple relations hold

$$p_S(x) = p_{S \cup S^\star}(x) - p_{S \setminus S^\star}(x) \geq p_{S^\star}(x) - p_{S \triangle S^\star}(x),$$
$$p_S(x) = p_{S \setminus S^\star}(x) + p_{S \cap S^\star}(x) \leq p_{S \triangle S^\star}(x) + p_{S^\star}(x),$$
$$p_{S \cap S^\star}(x) = p_{S^\star}(x) - p_{S \setminus S^\star}(x) \geq p_{S^\star}(x) - p_{S \triangle S^\star}(x)$$

and so, for any $x \in \mathcal{A}$, it holds that $\alpha\varepsilon^{1/3} - \varepsilon^{2/3} \leq p_S(x) \leq \alpha\varepsilon^{1/3} + \varepsilon^{2/3}$ and $p_{S \cap S^\star}(x) \geq \alpha\varepsilon^{1/3} - \varepsilon^{2/3}$. Note that, by our assumption on $\varepsilon$, we have $\alpha\varepsilon^{1/3} - \varepsilon^{2/3} > 0$. Combining with the previous expression, we obtain the following bound

$$\left\| \mathbb{E}_{x \sim \mathcal{D}_{\mathrm{obs}}} \left[ \left( \mathbb{E}_{z \sim \mathcal{N}(\langle w^\star, x\rangle, 1, S)}[z \cdot x] - \mathbb{E}_{y \sim \mathcal{N}(\langle w^\star, x\rangle, 1, S^\star \cap S)}[y \cdot x] \right) \cdot \mathbb{1}\{x \in \mathcal{A}\} \right] \right\|_2$$
$$\leq O\left( \sqrt{\left(1 + \log \frac{1}{\alpha\varepsilon^{1/3} - \varepsilon^{2/3}}\right) \cdot \log \frac{\alpha + \varepsilon^{1/3}}{\alpha - \varepsilon^{1/3}}} \cdot \Lambda\sqrt{d} \right).$$

To simplify the bound, set $z := \frac{\varepsilon^{1/3}}{\alpha} < \frac{1}{2}$, and observe that the quantity under the square root is

$$O\left( \left( \log \frac{1}{\alpha\varepsilon} + \log \frac{1}{1 - z} \right) \cdot \log \frac{1 + z}{1 - z} \right).$$

Using the inequality $\ln \frac{1+z}{1-z} \leq 3z$ for $z \in [0, 1/2]$, we get that $\log \frac{1}{\alpha \varepsilon} \cdot \log \frac{1+z}{1-z} \lesssim z \cdot \log \frac{1}{\alpha \varepsilon}$, and furthermore using the inequalities $\ln(1+z) \leq z$, $z \ln \frac{1}{1-z} \leq z$ and $\ln \frac{1}{1-z} \leq \sqrt{z}$ for $z \in [0, 1/2]$ we have $\log \frac{1}{1-z} \cdot \log \frac{1+z}{1-z} = \log(1+z) \cdot \log \frac{1}{1-z} + \log\left(\frac{1}{1-z}\right)^2 \lesssim z$. Thus, recalling the definition of $z$, we can write the above bound as

$$\left\| \mathop{\mathbb{E}}_{x \sim \mathcal{D}_{\mathrm{obs}}} \left[ \left( \mathop{\mathbb{E}}_{z \sim \mathcal{N}(\langle w^\star, x \rangle, 1, S)} [z \cdot x] - \mathop{\mathbb{E}}_{y \sim \mathcal{N}(\langle w^\star, x \rangle, 1, S^\star \cap S)} [y \cdot x] \right) \cdot \mathbb{1}\{x \in \mathcal{A}\} \right] \right\|_2 \leq \widetilde{O}\left( \frac{\Lambda}{\alpha^{1/2}} \cdot \sqrt{d} \cdot \varepsilon^{1/6} \right) .$$

To bound the second norm appearing in Equation (11), applying the triangle inequality followed by Jensen's inequality yields

$$\left\| \mathop{\mathbb{E}}_{x \sim \mathcal{D}_{\mathrm{obs}}} \left[ \left( \mathop{\mathbb{E}}_{z \sim \mathcal{N}(\langle w^\star, x \rangle, 1, S)} [z \cdot x] - \mathop{\mathbb{E}}_{y \sim \mathcal{N}(\langle w^\star, x \rangle, 1, S^\star \cap S)} [y \cdot x] \right) \cdot \mathbb{1}\{x \notin \mathcal{A}\} \right] \right\|_2$$
$$\leq \mathop{\mathbb{E}}_{x \sim \mathcal{D}_{\mathrm{obs}}} \left[ \mathop{\mathbb{E}}_{z \sim \mathcal{N}(\langle w^\star, x \rangle, 1, S)} |z| \cdot \|x\|_2 \cdot \mathbb{1}\{x \in \mathcal{A}\} \right] + \mathop{\mathbb{E}}_{x \sim \mathcal{D}_{\mathrm{obs}}} \left[ \mathop{\mathbb{E}}_{y \sim \mathcal{N}(\langle w^\star, x \rangle, 1, S^\star \cap S)} |y| \cdot \|x\|_2 \cdot \mathbb{1}\{x \notin \mathcal{A}\} \right] .$$

By the guarantee of Theorem E.1, it follows that

$$\mathop{\mathbb{E}}_{z \sim \mathcal{N}(\langle w^\star, x \rangle, 1, S)} |z| , \quad \mathop{\mathbb{E}}_{y \sim \mathcal{N}(\langle w^\star, x \rangle, 1, S^\star \cap S)} |y| \leq R$$

where $R \leq \widetilde{O}\left( (1 + \|w^\star\|_2) \cdot \left( \|\mu\|_2 + \left( \sigma + \frac{\Lambda}{\alpha} \right) \sqrt{\log \frac{1}{\varepsilon}} \right) \right)$. Therefore, we obtain

$$\left\| \mathop{\mathbb{E}}_{x \sim \mathcal{D}_{\mathrm{obs}}} \left[ \left( \mathop{\mathbb{E}}_{z \sim \mathcal{N}(\langle w^\star, x \rangle, 1, S)} [z \cdot x] - \mathop{\mathbb{E}}_{y \sim \mathcal{N}(\langle w^\star, x \rangle, 1, S^\star \cap S)} [y \cdot x] \right) \cdot \mathbb{1}\{x \notin \mathcal{A}\} \right] \right\|_2$$
$$\leq 2R \cdot \mathop{\mathbb{E}}_{x \sim \mathcal{D}_{\mathrm{obs}}} [\|x\|_2 \cdot \mathbb{1}\{x \notin \mathcal{A}\}]$$
$$\leq 2R \cdot \sqrt{ \mathop{\mathbb{E}}_{x \sim \mathcal{D}_{\mathrm{obs}}} \left[ \|x\|_2^2 \right] \mathop{\mathbb{E}}_{x \sim \mathcal{D}_{\mathrm{obs}}} [\mathbb{1}\{x \notin \mathcal{A}\}] }$$
$$\leq 2\Lambda R \sqrt{d} \sqrt{ \mathop{\Pr}_{x \sim \mathcal{D}_{\mathrm{obs}}} (x \notin \mathcal{A}) } \leq 2\sqrt{2} \cdot \Lambda R \sqrt{d} \varepsilon^{1/6}$$

where we used the Cauchy–Schwarz inequality and Assumption 4. Substituting our above bounds into Equation (11) yields the lemma. $\qquad \square$

### E.3.5. EFFICIENT GRADIENT SAMPLER

Next we prove desirable properties of our efficient (biased) gradient sampler, namely, bounds on its runtime, bias, and on the second moment of its stochastic gradients.

**Proof of Lemma E.13 (Survival Probability Lower Bound).** We begin with the following lower bound for the survival probability of a sample drawn from the observed distribution $\mathcal{D}_{\mathrm{obs}}$.

**Lemma E.13.** *Suppose that Assumptions 1 to 4 hold. Let $S \subseteq \mathbb{R}$ be a set satisfying the guarantees of Theorem E.1 for some $\varepsilon < \alpha$. Let $\mathcal{P}$ be the projection set defined in Equation (7) for some $\widehat{w} \in \mathbb{R}^d$, and suppose that $w^\star \in \mathcal{P}$. Then, for any $w \in \mathcal{P}$ and any $\eta \in (0, 1)$, if $x \sim \mathcal{D}_{\mathrm{obs}}$, it holds with probability at least $1 - \eta$ that*

$$p(x, w; S) \geq \exp\left\{ -O\left( \left( 1 + \|w^\star\|_2^2 \right) \cdot \frac{\Lambda^{12} \sigma^2}{\rho^8 \alpha^6 (\alpha - \varepsilon)^2} \cdot \left( d + \log \frac{1}{\eta} \right) \right) \right\} .$$

*Proof.* We begin by bounding $\mathbb{E}_{\mathcal{D}_{\mathrm{obs}}} [p(x, w; S)]$. Letting $p_T(x) := p(x, w^\star; T)$ for brevity, we have by the first part of

Lemma D.3 that

$$\mathbb{E}_{\mathcal{D}_{\text{obs}}} \left[ p(x, w; S) \right] \geq \mathbb{E}_{x \sim \mathcal{D}_{\text{obs}}} \left[ p_S(x)^2 \cdot \exp \left( - \langle w - w^\star, x \rangle^2 - 2 \right) \right]$$

$$\geq e^{-2} \mathbb{E}_{x \sim \mathcal{D}_{\text{obs}}} \left[ p_S(x)^2 \right] \cdot \exp \left( - \frac{\mathbb{E}_{x \sim \mathcal{D}_{\text{obs}}} \left[ p_S(x)^2 \cdot \langle w - w^\star, x \rangle^2 \right]}{\mathbb{E}_{x \sim \mathcal{D}_{\text{obs}}} \left[ p_S(x)^2 \right]} \right)$$

$$\geq e^{-2} \mathbb{E}_{x \sim \mathcal{D}_{\text{obs}}} \left[ p_S(x)^2 \right] \cdot \exp \left( - \frac{\Lambda^2 \| w - w^\star \|_2^2}{\mathbb{E}_{x \sim \mathcal{D}_{\text{obs}}} \left[ p_S(x)^2 \right]} \right)$$

$$\geq e^{-2} \mathbb{E}_{x \sim \mathcal{D}_{\text{obs}}} \left[ p_S(x)^2 \right] \cdot \exp \left( - \frac{36 \Lambda^6}{\rho^4 \alpha^2 \cdot \mathbb{E}_{x \sim \mathcal{D}_{\text{obs}}} \left[ p_S(x)^2 \right]} \right)$$

where the second line is by Jensen's inequality, the third is by Assumption 4 and the fact that $p_S(x) \leq 1$, while the last is by the assumption that $w, w^\star \in \mathcal{P}$ and the definition of $\mathcal{P}$ in Equation (7). To complete the bound, note that $p_S(x) = p_{S \cup S^\star}(x) - p_{S \setminus S^\star}(x) \geq p_{S^\star}(x) - p_{S \triangle S^\star}(x)$, and therefore

$$\mathbb{E}_{x \sim \mathcal{D}_{\text{obs}}} \left[ p_S(x) \right] \geq \mathbb{E}_{x \sim \mathcal{D}_{\text{obs}}} \left[ p_{S^\star}(x) \right] - \mathbb{E}_{x \sim \mathcal{D}_{\text{obs}}} \left[ p_{S \triangle S^\star}(x) \right]$$

$$\geq \frac{\mathbb{E}_{x \sim \mathcal{D}} \left[ p_{S^\star}(x)^2 \right]}{\mathbb{E}_{x \sim \mathcal{D}} \left[ p_{S^\star}(x) \right]} - \varepsilon$$

$$\geq \mathbb{E}_{x \sim \mathcal{D}} \left[ p_{S^\star}(x) \right] - \varepsilon$$

$$\geq \alpha - \varepsilon$$

where the second step is by definition of $\mathcal{D}_{\text{obs}}$ and the guarantee of Theorem E.1, the third is by Jensen's inequality, and the last is by Assumption 1. Since $\varepsilon < \alpha$ by assumption, another application of Jensen's inequality gives $\mathbb{E}_{x \sim \mathcal{D}_{\text{obs}}} \left[ p_S(x)^2 \right] \geq \left( \mathbb{E}_{x \sim \mathcal{D}_{\text{obs}}} \left[ p_S(x) \right] \right)^2 \geq (\alpha - \varepsilon)^2$ and so we can combine our above bounds to obtain

$$\mathbb{E}_{\mathcal{D}_{\text{obs}}} \left[ p(x, w; S) \right] \geq e^{-2} \cdot (\alpha - \varepsilon)^2 \cdot \exp \left( - \frac{36 \Lambda^6}{\rho^4 \alpha^2 (\alpha - \varepsilon)^2} \right) = e^{- \widetilde{O} \left( \frac{\Lambda^6}{\rho^4 \alpha^2 (\alpha - \varepsilon)^2} \right)} .$$

Let $q := \widetilde{O} \left( \frac{\Lambda^6}{\rho^4 \alpha^2 (\alpha - \varepsilon)^2} \right)$ for brevity, so that $\mathbb{E}_{\mathcal{D}_{\text{obs}}} \left[ p(x, w; S) \right] \geq e^{-q}$. Having established this bound, we now turn to proving the lemma. Let $\widehat{\mu} := \mathbb{E}_{\mathcal{D}_{\text{obs}}} x$ and $\eta \in (0, 1)$, and define the following ball around $\widehat{\mu}$

$$\mathcal{B} := \left\{ x \in \mathbb{R}^d \ \middle| \ \| x - \widehat{\mu} \|_2 \leq CK \left( \sqrt{d} + \sqrt{q + \log \frac{2}{\eta}} \right) \right\}$$

where $K := O \left( \sqrt{\sigma^2 \log \frac{1}{\alpha} + \frac{\Lambda^2}{\alpha^2}} \right)$ is the sub-Gaussian norm of the mean-zero random vector $x - \widehat{\mu}$ (see Lemma F.5), and $C > 0$ is a constant. By Proposition 6.2.1 of (Vershynin, 2018), we can pick $C$ so that

$$\Pr_{x \sim \mathcal{D}_{\text{obs}}} (x \in \mathcal{B}) \geq 1 - \frac{e^{-q}}{2} \cdot \eta \geq \max \left( 1 - \frac{e^{-q}}{2}, 1 - \eta \right) .$$

We claim that there is $x^* \in \mathcal{B}$, s.t., $p(x^*, w; S) \geq \frac{e^{-q}}{2}$. Indeed, if $p(x, w; S) < \frac{e^{-q}}{2}$ for all $x \in \mathcal{B}$, then

$$\mathbb{E}_{\mathcal{D}_{\text{obs}}} \left[ p(x, w; S) \right] = \mathbb{E}_{\mathcal{D}_{\text{obs}}} \left[ p(x, w; S) \cdot \mathbb{1} \{ x \in \mathcal{B} \} \right] + \mathbb{E}_{\mathcal{D}_{\text{obs}}} \left[ p(x, w; S) \cdot \mathbb{1} \{ x \notin \mathcal{B} \} \right]$$

$$< \frac{e^{-q}}{2} + \mathbb{E}_{\mathcal{D}_{\text{obs}}} \left[ \mathbb{1} \{ x \notin \mathcal{B} \} \right]$$

$$\leq e^{-q}$$

where the second line is by the assumption and the trivial bounds $\mathbb{1} \{ \cdot \} \leq 1$ and $p(x, w; S) \leq 1$, while the last line is by our above bound on $\Pr_{x \sim \mathcal{D}_{\text{obs}}} (x \in \mathcal{B})$. However, this contradicts our previous lower bound of $\mathbb{E}_{\mathcal{D}_{\text{obs}}} \left[ p(x, w; S) \right] \geq e^{-q}$. Thus,

there exists indeed some $x^* \in \mathcal{B}$ such that $p(x^*, w; S) \geq \frac{e^{-q}}{2}$. Now, we can use the second part of Lemma D.3 to obtain, for any $x \in \mathcal{B}$

$$
\begin{aligned}
p(x, w; S) &\gtrsim p(x^*, w; S)^2 \cdot \exp\left(-\langle w, x - x^* \rangle^2\right) \\
&\gtrsim e^{-2q} \cdot \exp\left(-\|w\|_2^2 \|x - x^*\|_2^2\right) \\
&\geq e^{-2q} \cdot \exp\left(-O\left(\left(\|w^\star\|_2^2 + \frac{\Lambda^4}{\rho^4 \alpha^2}\right) \cdot K^2\left(d + q + \log\frac{1}{\eta}\right)\right)\right) \\
&\geq \exp\left\{-O\left(\left(1 + \|w^\star\|_2^2\right) \cdot \frac{\Lambda^{12} \sigma^2}{\rho^8 \alpha^6 (\alpha - \varepsilon)^2} \cdot \left(d + \log\frac{1}{\eta}\right)\right)\right\} .
\end{aligned}
$$

The second line is by the assumption on $x^*$ and the Cauchy–Schwarz inequality. The third line is by the assumption $w, w^\star \in \mathcal{P}$ and the definition of $\mathcal{P}$ in Equation (7), as well as the fact that $\|x - x^*\|_2 \leq O(K(\sqrt{d} + \sqrt{q + \log 1/\eta}))$ for any $x, x^* \in \mathcal{B}$. The last line is by substituting our above definitions of $q, K$. Since this holds for all $x \in \mathcal{B}$, the lemma now follows by our previous bound on the mass that $\mathcal{D}_{\text{obs}}$ places on $\mathcal{B}$. $\qquad \square$

As discussed in Appendix E.2.5, the above bound suffices to establish a high-probability bound on the runtime of the gradient sampler given in Algorithm 4.

**Proof of Lemma E.15 (Bound on the Bias of Gradients).** We continue by establishing the bound on the bias of Algorithm 4. Towards this, let $\widehat{\mathcal{D}}_S, \widetilde{\mathcal{D}}_{w,S}$ be the distributions of the samples $(\widehat{x}, \widehat{y}), (\widetilde{x}, \widetilde{z}) \in \mathbb{R}^d \times \mathbb{R}$ generated by Algorithm 4. Also, let $\widehat{\mathcal{Q}}_S$ be a distribution over $(x, y) \in \mathbb{R}^d \times \mathbb{R}$ such that the marginal of $x$ is $\mathcal{D}_{\text{obs}}$ and the marginal of $y$ given $x$ is $\mathcal{N}(\langle w^\star, x \rangle, 1, S^\star \cap S)$. Similarly, let $\widetilde{\mathcal{Q}}_{w,S}$ be a distribution over $(x, z) \in \mathbb{R}^d \times \mathbb{R}$ such that the marginal of $x$ is $\mathcal{D}_{\text{obs}}$ and the marginal of $z$ given $x$ is $\mathcal{N}(\langle w, x \rangle, 1, S)$. Clearly, if we could sample directly from $\widehat{\mathcal{Q}}_S, \widetilde{\mathcal{Q}}_{w,S}$, then we would be able to obtain an exact gradient sample. Instead, Algorithm 4 samples from $\widehat{\mathcal{D}}_S, \widetilde{\mathcal{D}}_{w,S}$, and we will show that the distributions $\widehat{\mathcal{D}}_S, \widehat{\mathcal{Q}}_S$ and $\widetilde{\mathcal{D}}_{w,S}, \widetilde{\mathcal{Q}}_{w,S}$ are close in TV distance.

**Lemma E.27.** *Suppose that Assumptions 1 to 4 hold. Let $S \subseteq \mathbb{R}$ be a set satisfying the guarantees of Theorem E.1 for some $\varepsilon < \alpha$. Then, with the above notation, for any $w \in \mathbb{R}^d$ it holds that*

$$
d_{\mathsf{TV}}(\widehat{\mathcal{D}}_S, \widehat{\mathcal{Q}}_S) \leq 2\sqrt{\frac{\varepsilon}{\alpha}} \qquad \text{and} \qquad d_{\mathsf{TV}}(\widetilde{\mathcal{D}}_{w,S}, \widetilde{\mathcal{Q}}_{w,S}) \leq \zeta .
$$

*Proof.* The second part of the lemma is straightforward to see. Indeed, for any $x \in \mathbb{R}^d$, let $\widetilde{\mathcal{D}}_{w,S}^{z|x}$ be the marginal of $z$ given $x$ under $\widetilde{\mathcal{D}}_{w,S}$, and similarly let $\widetilde{\mathcal{Q}}_{w,S}^{z|x} = \mathcal{N}(\langle w, x \rangle, 1, S)$ be the marginal of $z$ given $x$ under $\widetilde{\mathcal{Q}}_{w,S}$. Then Lemma E.12 guarantees that there exists an optimal coupling $\mathcal{C}_x$ of $\widetilde{\mathcal{D}}_{w,S}^{z|x}, \widetilde{\mathcal{Q}}_{w,S}^{z|x}$, such that $\Pr_{(\widetilde{z}, \widetilde{z}') \sim \mathcal{C}_x}(\widetilde{z} \neq \widetilde{z}') \leq \zeta$. Hence, we can construct a coupling of $\widetilde{\mathcal{D}}_{w,S}, \widetilde{\mathcal{Q}}_{w,S}$ as follows: first, sample $\widetilde{x} \sim \mathcal{D}_{\text{obs}}$ and set $\widetilde{x}' = \widetilde{x}$, and then sample $(\widetilde{z}, \widetilde{z}') \sim \mathcal{C}_{\widetilde{x}}$. By the description of Algorithm 4 and the definition of $\mathcal{C}_{\widetilde{x}}$, since $\widetilde{\mathcal{D}}_{w,S}, \widetilde{\mathcal{Q}}_{w,S}$ have the same $x$-marginal $\mathcal{D}_{\text{obs}}$, it is clear that $(\widetilde{x}, \widetilde{z}) \sim \widetilde{\mathcal{D}}_{w,S}$ and $(\widetilde{x}', \widetilde{z}') \sim \widetilde{\mathcal{Q}}_{w,S}$, so the above is indeed the coupling, and clearly $(\widetilde{x}, \widetilde{z}) \neq (\widetilde{x}', \widetilde{z}')$ iff $\widetilde{z} \neq \widetilde{z}'$, which happens with probability at most $\zeta$.

To show the first part of the claim, since $\widehat{\mathcal{D}}_S$ and $\widehat{\mathcal{Q}}_S$ have the same $x$-marginal $\mathcal{D}_{\text{obs}}$, we have

$$
\begin{aligned}
d_{\mathsf{TV}}(\widehat{\mathcal{D}}_S, \widehat{\mathcal{Q}}_S) &= \frac{1}{2} \int \left|\widehat{\mathcal{D}}_S(x, y) - \widehat{\mathcal{Q}}_S(x, y)\right| \mathrm{d}x \, \mathrm{d}y \\
&= \frac{1}{2} \int |\mathcal{N}(y; \langle w^\star, x \rangle, 1, S^\star) - \mathcal{N}(y; \langle w^\star, x \rangle, 1, S^\star \cap S)| \, \mathcal{D}_{\text{obs}}(x) \, \mathrm{d}x \, \mathrm{d}y \\
&= \mathop{\mathbb{E}}_{x \sim \mathcal{D}_{\text{obs}}} [d_{\mathsf{TV}}(\mathcal{N}(\langle w^\star, x \rangle, 1, S^\star), \mathcal{N}(\langle w^\star, x \rangle, 1, S^\star \cap S))] .
\end{aligned}
$$

Note that, for any $\mu \in \mathbb{R}$, since $S^\star \cap S \subseteq S^\star$, it is immediate that the density of $\mathcal{N}(\mu, 1, S^\star \cap S)$ is above that of $\mathcal{N}(\mu, 1, S^\star)$ in the set $S^\star \cap S$, and it is zero outside of that set. Therefore, for any $\mu \in \mathbb{R}$

$$
\begin{aligned}
d_{\mathsf{TV}}(\mathcal{N}(\mu, 1, S^\star), \mathcal{N}(\mu, 1, S^\star \cap S)) &= \sup_{T \subseteq \mathbb{R}} \left[ \mathcal{N}(T; \mu, 1, S^\star) - \mathcal{N}(T; \mu, 1, S^\star \cap S) \right] \\
&= \mathcal{N}(S^\star \setminus S; \mu, 1, S^\star) \\
&= \frac{\mathcal{N}(S^\star \setminus S; \mu, 1)}{\mathcal{N}(S^\star; \mu, 1)} \, .
\end{aligned}
$$

Combining the above, and setting $p_T(x) := p(x, w^\star; T) = \mathcal{N}(T; \langle w^\star, x \rangle, 1)$ for brevity, we obtain that

$$
d_{\mathsf{TV}}(\widehat{\mathcal{D}}_S, \widehat{\mathcal{Q}}_S) = \mathbb{E}_{x \sim \mathcal{D}_{\mathrm{obs}}} \left[ \frac{p_{S^\star \setminus S}(x)}{p_{S^\star}(x)} \right] \, .
$$

To bound this quantity, note that, by the guarantee of Theorem E.1 and Markov's inequality, we have for any $\eta > 0$ that

$$
\Pr_{x \sim \mathcal{D}_{\mathrm{obs}}} (p_{S^\star \triangle S}(x) \geq \eta) \leq \frac{\mathbb{E}_{x \sim \mathcal{D}_{\mathrm{obs}}}[p_{S^\star \triangle S}(x)]}{\eta} \leq \frac{\varepsilon}{\eta} \, .
$$

Thus, an application of the law of total expectation yields

$$
\begin{aligned}
d_{\mathsf{TV}}(\widehat{\mathcal{D}}_S, \widehat{\mathcal{Q}}_S) &= \mathbb{E}_{x \sim \mathcal{D}_{\mathrm{obs}}} \left[ \frac{p_{S^\star \setminus S}(x)}{p_{S^\star}(x)} \cdot \mathbb{1}\{p_{S^\star \triangle S}(x) < \eta\} \right] + \mathbb{E}_{x \sim \mathcal{D}_{\mathrm{obs}}} \left[ \frac{p_{S^\star \setminus S}(x)}{p_{S^\star}(x)} \cdot \mathbb{1}\{p_{S^\star \triangle S}(x) \geq \eta\} \right] \\
&\leq \eta \cdot \mathbb{E}_{x \sim \mathcal{D}_{\mathrm{obs}}} \left[ \frac{1}{p_{S^\star}(x)} \right] + \mathbb{E}_{x \sim \mathcal{D}_{\mathrm{obs}}} [\mathbb{1}\{p_{S^\star \triangle S}(x) \geq \eta\}] \\
&\leq \frac{\eta}{\mathbb{E}_{\mathcal{D}}[p_S^\star(x)]} + \frac{\varepsilon}{\eta} \\
&\leq \frac{\eta}{\alpha} + \frac{\varepsilon}{\eta} \, .
\end{aligned}
$$

In the second line, we used the trivial bounds $p_{S^\star \setminus S}(x) \leq p_{S^\star \triangle S}(x), p_{S^\star}(x)$ and $\mathbb{1}\{\cdot\} \leq 1$. In the third line, we substituted the definition of $\mathcal{D}_{\mathrm{obs}}$ from Equation (3), and the last line is by Assumption 1. Picking $\eta := \sqrt{\alpha \varepsilon}$ gives the first part of the lemma. $\qquad \square$

It is now straightforward to bound the bias of the gradient samples returned by Algorithm 4, thus establishing the second main result of Appendix E.2.5.

**Lemma E.15.** *Suppose that Assumptions 1 to 4 hold. Let $S \subseteq \mathbb{R}$ be a set satisfying the guarantees of Theorem E.1 for some $\varepsilon < \alpha$. Then, for any $w \in \mathbb{R}^d$ and $\zeta < \sqrt{\varepsilon}$, if $g(w)$ is the output of a call to Algorithm 4 with input $w, S, \zeta$, it holds that*

$$
\|\mathbb{E}[g(w)] - \nabla \mathscr{L}_S(w)\|_2 \leq \widetilde{O}\left( (1 + \|w^\star\|_2) \cdot \left( \sigma^2 + \|\mu\|_2^2 \right) \cdot \frac{\Lambda^2}{\alpha^{5/4}} \cdot \sqrt{d} \cdot \varepsilon^{1/4} \right)
$$

*where $\mu := \mathbb{E}_{\mathcal{D}} x$.*

*Proof.* Keeping the above notation, consider the product distributions $\widehat{\mathcal{D}}_S \times \widetilde{\mathcal{D}}_{w,S}$ and $\widehat{\mathcal{Q}}_S \times \widetilde{\mathcal{Q}}_{w,S}$, and the following random variables with their respective marginals: $(\widehat{x}, \widehat{y}) \sim \widehat{\mathcal{D}}_S$, $(\widehat{x}', \widehat{y}') \sim \widehat{\mathcal{Q}}_S$, $(\widetilde{x}, \widetilde{z}) \sim \widetilde{\mathcal{D}}_{w,S}$ and $(\widetilde{x}', \widetilde{z}') \sim \widetilde{\mathcal{Q}}_{w,S}$. By Lemma E.27 and a union bound, there exists some coupling $\mathcal{C}$ of these two product distributions, such that

$$
\Pr_{\mathcal{C}} ((\widehat{x}, \widehat{y}) \neq (\widehat{x}', \widehat{y}') \quad \text{or} \quad (\widetilde{x}, \widetilde{z}) \neq (\widetilde{x}', \widetilde{z}')) \leq 2\sqrt{\frac{\varepsilon}{\alpha}} + \zeta \, .
$$

Let $\mathcal{E}$ be this low-probability event. By the definitions of $\widehat{\mathcal{D}}_S, \widetilde{\mathcal{D}}_{w,S}, \widehat{\mathcal{Q}}_S, \widetilde{\mathcal{Q}}_{w,S}$, it is clear that

$$
\begin{aligned}
\mathbb{E}[g(w)] - \nabla \mathscr{L}_S(w) &= \mathbb{E}[\widetilde{z} \cdot \widetilde{x} - \widehat{y} \cdot \widehat{x}] - \mathbb{E}[\widetilde{z}' \cdot \widetilde{x}' - \widehat{y}' \cdot \widehat{x}'] \\
&= \mathbb{E}_{\mathcal{C}}[(\widetilde{z} \cdot \widetilde{x} - \widehat{y} \cdot \widehat{x}) - (\widetilde{z}' \cdot \widetilde{x}' - \widehat{y}' \cdot \widehat{x}')] \\
&= \mathbb{E}_{\mathcal{C}}[((\widetilde{z} \cdot \widetilde{x} - \widehat{y} \cdot \widehat{x}) - (\widetilde{z}' \cdot \widetilde{x}' - \widehat{y}' \cdot \widehat{x}')) \cdot \mathbb{1}_{\mathcal{E}}]
\end{aligned}
$$

where the last line is because, when $\mathcal{E}$ does not hold, then the two terms inside the expectation are equal by definition of $\mathcal{E}$. Therefore, the triangle inequality and Jensen's inequality yields

$$\|\mathbb{E}\left[g(w)\right] - \boldsymbol{\nabla}\mathscr{L}_S(w)\|_2 \leq \underset{\mathcal{C}}{\mathbb{E}}\left[|\widetilde{z}'| \cdot \|\widetilde{x}'\|_2 \cdot \mathbb{1}_{\mathcal{E}}\right] + \underset{\mathcal{C}}{\mathbb{E}}\left[|\widetilde{z}| \cdot \|\widetilde{x}\|_2 \cdot \mathbb{1}_{\mathcal{E}}\right] + \underset{\mathcal{C}}{\mathbb{E}}\left[|\widehat{y}'| \cdot \|\widehat{x}'\|_2 \cdot \mathbb{1}_{\mathcal{E}}\right] + \underset{\mathcal{C}}{\mathbb{E}}\left[\|\widehat{y} \cdot \widehat{x}\|_2 \cdot \mathbb{1}_{\mathcal{E}}\right].$$

For the first three terms, note that $\widetilde{z}', \widetilde{z}, \widehat{y}' \in S$ a.s. under $\mathcal{C}$, and since $S$ satisfies the guarantee of Theorem E.1, it holds a.s. that $|\widetilde{z}'|, |\widetilde{z}|, |\widehat{y}'| \leq R$, where $R \leq \widetilde{O}\left((1 + \|w^\star\|_2) \cdot \left(\|\mu\|_2 + \left(\sigma + \frac{\Lambda}{\alpha}\right)\sqrt{\log\frac{1}{\varepsilon}}\right)\right)$. Hence, the above becomes

$$\|\mathbb{E}\left[g(w)\right] - \boldsymbol{\nabla}\mathscr{L}_S(w)\|_2 \leq R\left(\underset{\mathcal{C}}{\mathbb{E}}\left[\|\widetilde{x}'\|_2 \cdot \mathbb{1}_{\mathcal{E}}\right] + \underset{\mathcal{C}}{\mathbb{E}}\left[\|\widetilde{x}\|_2 \cdot \mathbb{1}_{\mathcal{E}}\right] + \underset{\mathcal{C}}{\mathbb{E}}\left[\|\widehat{x}'\|_2 \cdot \mathbb{1}_{\mathcal{E}}\right] + \underset{\mathcal{C}}{\mathbb{E}}\left[\|\widehat{y} \cdot \widehat{x}\|_2 \cdot \mathbb{1}_{\mathcal{E}}\right]\right).$$

Applying the Cauchy–Schwarz inequality to each of the expectations appearing on the right-hand side, and recalling that the marginal of each one of $\widetilde{x}', \widetilde{x}, \widehat{x}'$ under $\mathcal{C}$ is $\mathcal{D}_{\text{obs}}$, we obtain

$$\|\mathbb{E}\left[g(w)\right] - \boldsymbol{\nabla}\mathscr{L}_S(w)\|_2 \leq 3R\sqrt{\underset{\mathcal{D}_{\text{obs}}}{\mathbb{E}}\|x\|_2^2 \cdot \underset{\mathcal{C}}{\mathbb{E}}\left[\mathbb{1}_{\mathcal{E}}\right]} + \sqrt{\underset{\widehat{\mathcal{D}}_S}{\mathbb{E}}\|\widehat{y} \cdot \widehat{x}\|_2^2 \cdot \underset{\mathcal{C}}{\mathbb{E}}\left[\mathbb{1}_{\mathcal{E}}\right]}.$$

Now, by Assumption 4 we have that $\mathbb{E}_{\mathcal{D}_{\text{obs}}}\|x\|_2^2 \leq \Lambda^2 d$, while by definition of $\mathcal{E}$ and the assumption on $\zeta$ we have $\mathbb{E}_{\mathcal{C}}\left[\mathbb{1}_{\mathcal{E}}\right] \leq 2\sqrt{\frac{\varepsilon}{\alpha}} + \zeta \lesssim \sqrt{\frac{\varepsilon}{\alpha}}$. Lastly, recall by definition of $\widehat{\mathcal{D}}_S$ that $(\widehat{x}, \widehat{y})$ is simply a sample from our truncated regression model, and Lemma F.8 implies that, for any unit vector $v$, it holds that $\|\langle v, \widehat{y} \cdot \widehat{x}\rangle\|_{\psi_1} \leq K$, where $K \leq O\left((1 + \|w^\star\|_2) \cdot \left(\sigma^2 \log\frac{1}{\alpha} + \|\mu\|_2^2\right)\right)$. Hence, Proposition 2.8.1 of (Vershynin, 2018) gives $\mathbb{E}\langle v, \widehat{y} \cdot \widehat{x}\rangle^2 \lesssim K^2$, which implies that $\mathbb{E}\|\widehat{y} \cdot \widehat{x}\|_2^2 \lesssim K^2 d$. Substituting our bounds into the above display yields the lemma. $\qquad\square$

**Proof of Lemma E.16 (Bound on the Second Moment of the Gradients).** Finally, we show Lemma E.16, which upper bounds the second moment of the stochastic gradients generated by Algorithm 4. We begin with the following auxilliary lemma, which essentially upper bounds the second moment of one of the two terms appearing in $g(w)$, where $g(w)$ is the output of Algorithm 4 on input $w$.

**Lemma E.28.** *Suppose that Assumptions 1 to 4 hold. Let $S \subseteq \mathbb{R}$ be a set satisfying the guarantees of Theorem E.1 for some $\varepsilon < \alpha^2/4$. Also, let $\mathcal{P}$ be the projection set defined in Equation (7), suppose that $w^\star \in \mathcal{P}$, and let $w \in \mathcal{P}$ be a parameter vector in that set. Consider the random vector $(x, z) \in \mathbb{R}^d \times \mathbb{R}$, where the marginal of $x$ is $\mathcal{D}_{\text{obs}}$, and the marginal of $z$ given $x$ is $\mathcal{N}(\langle w, x\rangle, 1, S)$. Then it holds that*

$$\mathbb{E}\|z \cdot x\|_2^2 \leq \widetilde{O}\left(\left(1 + \|w^\star\|_2^2\right)\left(1 + \left(\|\mu\|_2^2 + \sigma^2\right)\varepsilon^{1/4}\right) \cdot \frac{M^4\Lambda^4 d}{\rho^4\alpha^2}\right).$$

*Proof.* First, note that

$$\underset{z \sim \mathcal{N}(\langle w, x\rangle, 1, S)}{\mathbb{E}}\left[z^2 \mid x\right] \lesssim \underset{z \sim \mathcal{N}(\langle w, x\rangle, 1, S)}{\mathbb{E}}\left[(z - \mathbb{E}z)^2 \mid x\right] + \left(\underset{z \sim \mathcal{N}(\langle w, x\rangle, 1, S)}{\mathbb{E}}\left[z \mid x\right]\right)^2.$$

By Lemma F.7 and Proposition 2.8.1 of (Vershynin, 2018), it follows that the first term is bounded by $O\left(1 + \log\frac{1}{p(x, w; S)}\right)$, while by Lemma D.1 it follows that the second term is bounded by $O\left(\langle w, x\rangle^2 + \log\frac{1}{p(x, w; S)}\right)$. Therefore, we have that

$$\underset{z \sim \mathcal{N}(\langle w, x\rangle, 1, S)}{\mathbb{E}}\left[z^2 \mid x\right] \leq O\left(1 + \langle w, x\rangle^2 + \log\frac{1}{p(x, w; S)}\right)$$

$$\leq O\left(1 + \langle w^\star, x\rangle^2 + \langle w - w^\star, x\rangle^2 + \log\frac{1}{p(x, w; S)}\right).$$

Also, Lemma D.3 implies that $\log\frac{1}{p(x, w; S)} \lesssim 1 + \log\frac{1}{p_S(x)} + \langle w - w^\star, x\rangle^2$, where we denote $p_S(x) := p(x, w^\star; S)$ for brevity. Hence, we can write

$$\underset{z \sim \mathcal{N}(\langle w, x\rangle, 1, S)}{\mathbb{E}}\left[z^2 \mid x\right] \leq O\left(1 + \langle w^\star, x\rangle^2 + \langle w - w^\star, x\rangle^2 + \log\frac{1}{p_S(x)}\right).$$

Next, define the set $\mathcal{A} := \left\{ x \in \mathbb{R}^d \mid p_{S^\star \triangle S}(x) \le \sqrt{\varepsilon} \right\}$, and note that by the guarantee of Theorem E.1 and Markov's inequality

$$\Pr_{x \sim \mathcal{D}_{\mathrm{obs}}} (x \in \mathcal{A}) \ge 1 - \frac{\mathbb{E}_{x \sim \mathcal{D}_{\mathrm{obs}}} [p_{S^\star \triangle S}(x)]}{\sqrt{\varepsilon}} \ge 1 - \sqrt{\varepsilon}.$$

Also, observe that

$$\mathbb{E}_{x \sim \mathcal{D}_{\mathrm{obs}}} [p_{S^\star}(x)] = \frac{\mathbb{E}_{x \sim \mathcal{D}_{\mathrm{obs}}} [p_{S^\star}(x)^2]}{\mathbb{E}_{x \sim \mathcal{D}_{\mathrm{obs}}} [p_{S^\star}(x)]} \ge \mathbb{E}_{x \sim \mathcal{D}} [p_{S^\star}(x)] \ge \alpha$$

where the first step is by definition of $\mathcal{D}_{\mathrm{obs}}$, the second is by Jensen's inequality and the last is by Assumption 1. Hence, by the fact that $p_S(x) = p_{S \cup S^\star}(x) - p_{S \setminus S^\star}(x) \ge p_{S^\star}(x) - p_{S \triangle S^\star}(x)$, it follows that $p_S(x) \ge \alpha - \sqrt{\varepsilon} \ge \alpha^2/2$ for all $x \in \mathcal{A}$, given our assumption on $\varepsilon$.

Lastly, by the guarantee of Theorem E.1 and the fact that $z \in S$ a.s., it follows that $|z| \le R$ a.s., where $R \le \widetilde{O}\left( (1 + \|w^\star\|_2) \cdot \left( \|\mu\|_2 + \left(\sigma + \frac{\Lambda}{\alpha}\right) \sqrt{\log \frac{1}{\varepsilon}} \right) \right)$. Having established all the above relations, we now turn to bounding the desired expectation, starting by an application of the law of total expectation

$$\mathbb{E} \|z \cdot x\|_2^2 = \mathbb{E} \left[ \|z \cdot x\|_2^2 \cdot \mathbb{1}\{x \notin \mathcal{A}\} \right] + \mathbb{E} \left[ \|z \cdot x\|_2^2 \cdot \mathbb{1}\{x \in \mathcal{A}\} \right].$$

For the first term, by the Cauchy–Schwarz inequality we have

$$\mathbb{E} \left[ \|z \cdot x\|_2^2 \cdot \mathbb{1}\{x \notin \mathcal{A}\} \right] \le \sqrt{\mathbb{E} \left[ \|z \cdot x\|_2^4 \right] \cdot \mathbb{E} [\mathbb{1}\{x \notin \mathcal{A}\}]} \le R^2 M^2 d \varepsilon^{1/4}$$

where we used Assumption 4. For the second term, we have that

$$\begin{aligned}
\mathbb{E} \left[ \|z \cdot x\|_2^2 \cdot \mathbb{1}\{x \in \mathcal{A}\} \right] &= \mathbb{E} \left[ \mathbb{E} \left[ z^2 \mid x \right] \cdot \|x\|_2^2 \cdot \mathbb{1}\{x \in \mathcal{A}\} \right] \\
&\lesssim \mathbb{E} \left[ \left( 1 + \langle w^\star, x \rangle^2 + \langle w - w^\star, x \rangle^2 + \log \frac{1}{p_S(x)} \right) \cdot \|x\|_2^2 \cdot \mathbb{1}\{x \in \mathcal{A}\} \right] \\
&\le \mathbb{E} \left[ \left( 1 + \langle w^\star, x \rangle^2 + \langle w - w^\star, x \rangle^2 + \log \frac{2}{\alpha^2} \right) \cdot \|x\|_2^2 \right] \\
&\le \left( 1 + \log \frac{2}{\alpha^2} \right) \cdot \Lambda^2 d + \mathbb{E} \left[ \langle w^\star, x \rangle^2 \cdot \|x\|_2^2 \right] + \mathbb{E} \left[ \langle w - w^\star, x \rangle^2 \cdot \|x\|_2^2 \right] \\
&\le \left( 1 + \log \frac{2}{\alpha^2} \right) \cdot \Lambda^2 d + \left( \|w^\star\|_2^2 + \|w - w^\star\|_2^2 \right) \cdot M^4 d
\end{aligned}$$

where the second and third lines follow by our above relations, the second-to-last line is by Assumption 4, while the last line is by applying the Cauchy–Schwarz inequality followed by Assumption 4. Now, since $w, w^\star \in \mathcal{P}$ by assumption, it follows that $\|w - w^\star\|_2 \lesssim \frac{\Lambda^2}{\rho^2 \alpha}$ by Equation (7), and so we can write

$$\mathbb{E} \left[ \|z \cdot x\|_2^2 \cdot \mathbb{1}\{x \in \mathcal{A}\} \right] \le \widetilde{O} \left( \left( 1 + \|w^\star\|_2^2 + \frac{\Lambda^4}{\rho^4 \alpha^2} \right) \cdot M^4 d \right).$$

The lemma now follows by combining our bounds and recalling the definition of $R$. $\qquad\square$

Having established the above helper lemma, we now turn to showing our main result.

**Lemma E.16.** *Suppose that Assumptions 1 to 4 hold. Let $S \subseteq \mathbb{R}$ be a set satisfying the guarantees of Theorem E.1 for some $\varepsilon < \alpha^2/4$. Also, let $\mathcal{P}$ be the projection set defined in Equation (7) for some $\widehat{w} \in \mathbb{R}^d$, and suppose that $w^\star \in \mathcal{P}$. Then, for any $w \in \mathcal{P}$ and $\zeta < \sqrt{\varepsilon}$, if $g(w)$ is the output of a call to Algorithm 4 with input $w$, $S$, $\zeta$, it holds that*

$$\mathbb{E} \left[ \|g(w)\|_2^2 \mid w \right] \le \widetilde{O} \left( \left( 1 + \|w^\star\|_2^2 \right) \cdot \left( \|\mu\|_2^4 + \sigma^4 \right) \cdot \frac{M^4 \Lambda^4 d}{\rho^4 \alpha^2} \right)$$

*where $\mu := \mathbb{E}_{\mathcal{D}} x$.*

*Proof.* We will drop the conditioning on $w$ for brevity. Using the same notation as in the proof of Lemma E.15, we consider the product distributions $\widehat{\mathcal{D}}_S \times \widetilde{\mathcal{D}}_{w,S}$ and $\widehat{\mathcal{Q}}_S \times \widetilde{\mathcal{Q}}_{w,S}$, and the following random variables with their respective marginals: $(\widehat{x}, \widehat{y}) \sim \widehat{\mathcal{D}}_S$, $(\widetilde{x}, \widetilde{z}) \sim \widetilde{\mathcal{D}}_{w,S}$ and $(\widetilde{x}', \widetilde{z}') \sim \widetilde{\mathcal{Q}}_{w,S}$. Then it holds that $g(w) = \widetilde{z} \cdot \widetilde{x} - \widehat{y} \cdot \widehat{x}$, and we can write

$$\mathbb{E}\left[\|g(w)\|_2^2\right] = \mathbb{E}\left[\|\widetilde{z}\cdot\widetilde{x} - \widehat{y}\cdot\widehat{x}\|_2^2\right] \le 2\,\mathbb{E}\left[\|\widetilde{z}\cdot\widetilde{x}\|_2^2\right] + 2\,\mathbb{E}\left[\|\widehat{y}\cdot\widehat{x}\|_2^2\right].$$

To bound the second term, as in the last part of the proof of Lemma E.15, we have that $\mathbb{E}\|\widehat{y}\cdot\widehat{x}\|_2^2 \lesssim K^2 d$, where $K \le O\left((1 + \|w^\star\|_2)\cdot\left(\sigma^2\log\frac{1}{\alpha} + \|\mu\|_2^2\right)\right)$. For the first term, observe first that $\widetilde{z} \in S$ a.s., and since $S$ satisfies the guarantee of Theorem E.1, it follows that $|\widetilde{z}| \le R$ a.s., where $R \le \widetilde{O}\left((1 + \|w^\star\|_2)\cdot\left(\|\mu\|_2 + \left(\sigma + \frac{\Lambda}{\alpha}\right)\sqrt{\log\frac{1}{\varepsilon}}\right)\right)$. Also, by Lemma E.27, there exists a coupling $\mathcal{C}$ of $\widetilde{\mathcal{D}}_{w,S}$ and $\widetilde{\mathcal{Q}}_{w,S}$ such that $(\widetilde{x}, \widetilde{z}) = (\widetilde{x}', \widetilde{z}')$ with probability at least $1 - \zeta$ under $\mathcal{C}$. Letting $\mathcal{E}$ be this high-probability event, we can write

$$\begin{aligned}
\mathbb{E}\left[\|\widetilde{z}\cdot\widetilde{x}\|_2^2\right] &= \mathbb{E}_{\mathcal{C}}\left[\|\widetilde{z}'\cdot\widetilde{x}'\|_2^2 \cdot \mathbb{1}_{\mathcal{E}}\right] + \mathbb{E}_{\mathcal{C}}\left[\|\widetilde{z}\cdot\widetilde{x}\|_2^2 \cdot \mathbb{1}_{\overline{\mathcal{E}}}\right] \\
&\le \mathbb{E}\left[\|\widetilde{z}'\cdot\widetilde{x}'\|_2^2\right] + \sqrt{\mathbb{E}\left[\|\widetilde{z}\cdot\widetilde{x}\|_2^4\right]\cdot\mathbb{E}_{\mathcal{C}}\left[\mathbb{1}_{\overline{\mathcal{E}}}\right]} \\
&\le \mathbb{E}\|\widetilde{z}'\cdot\widetilde{x}'\|_2^2 + \sqrt{\zeta}R^2 M^2 d
\end{aligned}$$

where the second line is by the Cauchy–Schwarz inequality and the trivial bound $\mathbb{1}\{\cdot\} \le 1$, while the last line is by definition of $\mathcal{E}$, Assumption 4, and the fact that $|\widetilde{z}| \le R$. Therefore, we can combine the above, along with the assumption on $\zeta$, to obtain

$$\mathbb{E}\left[\|g(w)\|_2^2\right] \lesssim \mathbb{E}\|\widetilde{z}'\cdot\widetilde{x}'\|_2^2 + \left(K^2 + R^2 M^2 \varepsilon^{1/4}\right)d.$$

The lemma now follows by applying Lemma E.28 to bound the first term, and using the definitions of $K, R$. $\qquad\square$

# F. Auxiliary Results

## F.1. Useful Facts

Here, we establish some simple facts and relations that are used repeatedly in proofs.

**Fact F.1.** *Suppose Assumption 1 holds and $f : \mathbb{R}^d \to \mathbb{R}_{\ge 0}$ takes non-negative values. Then*

$$\mathbb{E}_{x\sim\mathcal{D}}[p_{S^\star}(x)\cdot f(x)] \le \mathbb{E}_{x\sim\mathcal{D}_{\mathrm{obs}}}[f(x)] \le \frac{1}{\alpha}\mathbb{E}_{x\sim\mathcal{D}}[p_{S^\star}(x)\cdot f(x)] \le \frac{1}{\alpha}\mathbb{E}_{x\sim\mathcal{D}}[f(x)].$$

*Proof.* All inequalities follow immediately from the definition of $\mathcal{D}_{\mathrm{obs}}$ in Equation (3), Assumption 1, and the fact that $p_{S^\star}(x) \le 1$. $\qquad\square$

**Fact F.2.** *Suppose Assumption 1 holds and $f : \mathbb{R}^d \to \mathbb{R}$. Then $|\mathbb{E}_{x\sim\mathcal{D}}[f(x)] - \mathbb{E}_{x\sim\mathcal{D}_{\mathrm{obs}}}[f(x)]| \le \frac{1}{\alpha}\mathbb{E}_{x\sim\mathcal{D}}|f(x)|.$*

*Proof.*

$$\begin{aligned}
\left|\mathbb{E}_{x\sim\mathcal{D}}[f(x)] - \mathbb{E}_{x\sim\mathcal{D}_{\mathrm{obs}}}[f(x)]\right| &= \left|\int f(x)\left(1 - \frac{p_{S^\star}(x)}{\mathbb{E}_{\mathcal{D}}\,p_{S^\star}(x)}\right)\mathcal{D}(x)\,\mathrm{d}x\right| \\
&\le \int |f(x)|\left|1 - \frac{p_{S^\star}(x)}{\mathbb{E}_{\mathcal{D}}\,p_{S^\star}(x)}\right|\mathcal{D}(x)\,\mathrm{d}x \\
&\le \frac{1}{\alpha}\mathbb{E}_{x\sim\mathcal{D}}|f(x)|
\end{aligned}$$

where we used the fact that, by Assumption 1, we have $1 - \frac{1}{\alpha} \leq 1 - \frac{p_{S^\star}(x)}{\mathbb{E}_{\mathcal{D}} \, p_{S^\star}(x)} \leq 1$, and therefore

$$\left| 1 - \frac{p_{S^\star}(x)}{\mathbb{E}_{\mathcal{D}} \, p_{S^\star}(x)} \right| \leq \max \left( \frac{1}{\alpha} - 1, 1 \right) \leq \frac{1}{\alpha}$$

$\square$

**Lemma F.3.** *Let $\mu := \mathbb{E}_{\mathcal{D}} \, x$ and $\widehat{\mu} := \mathbb{E}_{\mathcal{D}_{\mathrm{obs}}} \, x$. Suppose that Assumptions 1 and 4 hold. Then,*

$$\textit{for any unit vector } v, \qquad \langle v, \mu - \widehat{\mu} \rangle^2 \leq \frac{\Lambda^2}{\alpha^2} .$$

*Proof.* By definition of $\mathcal{D}_{\mathrm{obs}}$ in Equation (3), we have that

$$\langle v, \mu - \widehat{\mu} \rangle^2 = \left( \mathbb{E}_{\mathcal{D}} \langle v, x \rangle - \mathbb{E}_{\mathcal{D}_{\mathrm{obs}}} \langle v, x \rangle \right)^2 \leq \frac{1}{\alpha^2} \left( \mathbb{E}_{\mathcal{D}} |\langle v, x \rangle| \right)^2 \leq \frac{1}{\alpha^2} \cdot \mathbb{E}_{\mathcal{D}} \langle v, x \rangle^2 \leq \frac{\Lambda^2}{\alpha^2} .$$

where the second line is by Fact F.2, the third is by Jensen's inequality, and the last is by Assumption 4. $\square$

**Lemma F.4.** *Let $(x, y) \in \mathbb{R}^d \times \mathbb{R}$ be a* truncated *random vector such that the marginal of $x$ is $\mathcal{D}_{\mathrm{obs}}$ and the marginal of $y$ conditioned on $x$ is $\mathcal{N}(\langle w^\star, x \rangle, 1, S^\star)$. Suppose that Assumptions 1 and 4 hold. Then,*

$$\textit{for any unit vector } v, \qquad |\mathbb{E} \langle v, (y - \langle w^\star, x \rangle) \cdot x \rangle| \leq \frac{3\Lambda}{2\alpha} .$$

*Proof.* We have that

$$\begin{aligned}
|\mathbb{E} \langle v, (y - \langle w^\star, x \rangle) \cdot x \rangle| &\leq \mathbb{E}_{x \sim \mathcal{D}_{\mathrm{obs}}} \left[ \left| \mathbb{E}_{y \sim \mathcal{N}(\langle w^\star, x \rangle, 1, S^\star)} [y \mid x] - \langle w^\star, x \rangle \right| \cdot |\langle v, x \rangle| \right] \\
&\leq \frac{1}{\alpha} \mathbb{E}_{x \sim \mathcal{D}} \left[ p(x, w^\star; S^\star) \cdot \left( \sqrt{2 \log \frac{1}{p(x, w^\star; S^\star)}} + 1 \right) \cdot |\langle v, x \rangle| \right] \\
&\leq \frac{3}{2\alpha} \mathbb{E}_{x \sim \mathcal{D}} |\langle v, x \rangle| \\
&\leq \frac{3\Lambda}{2\alpha} .
\end{aligned}$$

where the first line is by Jensen's inequality, the second is by Fact F.1 and Lemma D.1, the third is because $\left( \sqrt{2 \log \frac{1}{t}} + 1 \right) t \leq \frac{3}{2}$ for all $t \in (0, 1)$, and the last is by Assumption 4. $\square$

### F.2. Consequences of Sub-Gaussianity of $\mathcal{D}$

In this section, we prove some concentration results that are required in many of our proofs. Specifically, we show that if $x \sim \mathcal{D}$ is a sub-Gaussian random vector (as in Assumption 2) then $x \sim \mathcal{D}_{\mathrm{obs}}$ is also a sub-Gaussian random vector, and furthermore, for $(x, y)$ drawn from our truncated regression model, the random vector $y \cdot x$ is sub-exponential. We start by reviewing the definition of the sub-Gaussian and sub-Exponential norm, as well as some standard facts about them.

**Definition 4** (sub-Gaussian and sub-exponential norms). *For any random variable $X$, its* sub-Gaussian *norm is defined as*

$$\|X\|_{\psi_2} := \inf \left\{ \sigma > 0 \colon \ \mathbb{E} \exp \left( \frac{X^2}{\sigma^2} \right) \leq 2 \right\}$$

*while its* sub-exponential *norm is defined as*

$$\|X\|_{\psi_1} := \inf \left\{ \sigma > 0 \colon \ \mathbb{E} \exp \left( \frac{|X|}{\sigma} \right) \leq 2 \right\} .$$

A fundamental relation between these two norms (see Lemma 2.8.6 of (Vershynin, 2018)) is that, for any random variables $X, Y$

$$\|XY\|_{\psi_1} \leq \|X\|_{\psi_2} \|Y\|_{\psi_2} . \tag{12}$$

It is also standard that the sub-Gaussian (or sub-exponential) norm of a centered random variable is at most a constant factor different from the same norm of the respective uncentered random variable (see Lemma 2.7.8 and Equation 2.26 of (Vershynin, 2018)). That is, there exist absolute constants $C_1, C_2$ such that, for any random variable $X$

$$\|X - \mathbb{E} X\|_{\psi_1} \leq C_1 \|X\|_{\psi_1} \quad \text{and} \quad \|X - \mathbb{E} X\|_{\psi_2} \leq C_2 \|X\|_{\psi_2} . \tag{13}$$

Having established the above, we now turn to showing that if $\mathcal{D}$ is a sub-Gaussian distribution, then so is $\mathcal{D}_{\text{obs}}$.

**Lemma F.5.** *Let $x \sim \mathcal{D}_{\text{obs}}$, and suppose Assumptions 1, 2 and 4 hold. Then, for any unit vector $v$,*

$$\|\langle v, x \rangle\|_{\psi_2} \leq O \left( \sigma \cdot \sqrt{\log \frac{1}{\alpha}} + \|\mu\|_2 \right) \qquad \text{and} \qquad \left\| \left\langle v, x - \mathbb{E}_{\mathcal{D}_{\text{obs}}} x \right\rangle \right\|_{\psi_2} \leq \widetilde{O} \left( \sigma + \frac{\Lambda}{\alpha} \right)$$

*where $\mu := \mathbb{E}_{\mathcal{D}} x$.*

*Proof.* Note that, by Fact F.1 and Assumption 2

$$\mathbb{E}_{\mathcal{D}_{\text{obs}}} \exp \left( \frac{\langle v, x - \mu \rangle^2}{\sigma^2} \right) \leq \frac{1}{\alpha} \mathbb{E}_{\mathcal{D}} \exp \left( \frac{\langle v, x - \mu \rangle^2}{\sigma^2} \right) \leq \frac{2}{\alpha} .$$

from which it follows that $\|\langle v, x - \mu \rangle\|_{\psi_2} \leq O \left( \sigma \cdot \sqrt{\log \frac{1}{\alpha}} \right)$. Therefore, the first claim follows by

$$\|\langle v, x \rangle\|_{\psi_2} \lesssim \|\langle v, x - \mu \rangle\|_{\psi_2} + |\langle v, \mu \rangle| \leq O \left( \sigma \cdot \sqrt{\log \frac{1}{\alpha}} \right) + \|\mu\|_2$$

where we used the triangle inequality and the fact that $\|1\|_{\psi_2} = 1/\sqrt{\ln 2}$. For the second claim, let $\widehat{\mu} := \mathbb{E}_{\mathcal{D}_{\text{obs}}} x$, and observe that

$$\|\langle v, x - \widehat{\mu} \rangle\|_{\psi_2} \lesssim \|\langle v, x - \mu \rangle\|_{\psi_2} + |\langle v, \mu - \widehat{\mu} \rangle| \leq O \left( \sigma \cdot \sqrt{\log \frac{1}{\alpha}} \right) + \frac{\Lambda}{\alpha}$$

where we used the triangle inequality and Lemma F.3. $\qquad \square$

We continue by considering the random vector $y \cdot x$. First, we establish that, in the simplest setting where we have no truncation on $y$, this vector concentrates.

**Lemma F.6.** *Let $(x, y) \in \mathbb{R}^d \times \mathbb{R}$ be an* untruncated *random vector such that the marginal of $x$ is $\mathcal{D}$ and the marginal of $y$ conditioned on $x$ is $\mathcal{N}(\langle w^\star, x \rangle, 1)$. Let $\mu := \mathbb{E}_{\mathcal{D}} x$, and suppose that Assumption 2 holds. Then there exists some absolute constant $C > 0$ such that*

$$\|\langle v, y \cdot x - \mathbb{E} y \cdot x \rangle\|_{\psi_1} \leq C \cdot (1 + \|w^\star\|_2) \cdot (\sigma + \|\mu\|_2)^2 .$$

*Proof.* Let $\xi$ be a standard normal random variable, independent of $x$, and let $v$ be an arbitrary unit vector. By Equation (13), for some absolute constant $C'$, we have that

$$
\begin{aligned}
\|\langle v, y \cdot x - \mathbb{E}\, y \cdot x \rangle\|_{\psi_1} &\le C' \, \|\langle v, y \cdot x \rangle\|_{\psi_1} \\
&= C' \, \|\langle w^\star, x \rangle \langle v, x \rangle + \xi \langle v, x \rangle\|_{\psi_1} \\
&\le C' \left( \|\langle w^\star, x \rangle \langle v, x \rangle\|_{\psi_1} + \|\xi \langle v, x \rangle\|_{\psi_1} \right)
\end{aligned}
$$

where we used the triangle inequality. We will bound each of those two terms separately. We have that

$$
\begin{aligned}
\|\langle w^\star, x \rangle \langle v, x \rangle\|_{\psi_1} &\le \|\langle w^\star, x \rangle\|_{\psi_2} \|\langle v, x \rangle\|_{\psi_2} \\
&\lesssim \left( \|\langle w^\star, x - \mu \rangle\|_{\psi_2} + |\langle w^\star, \mu \rangle| \right) \cdot \left( \|\langle v, x - \mu \rangle\|_{\psi_2} + |\langle v, \mu \rangle| \right) \\
&\le \|w^\star\|_2 \left( \sigma + \|\mu\|_2 \right)^2
\end{aligned}
$$

where the first line is by Equation (12), the second is by the triangle inequality, and the third is by the assumption on $x$ and the Cauchy–Schwarz inequality. Similarly, we have that

$$
\|\xi \langle v, x \rangle\|_{\psi_1} \le \|\xi\|_{\psi_2} \|\langle v, x \rangle\|_{\psi_2} \lesssim \left( \|\langle v, x - \mu \rangle\|_{\psi_2} + |\langle v, \mu \rangle| \right) \le \left( \sigma + \|\mu\|_2 \right)
$$

where we used the simple fact that $\|\xi\|_{\psi_2} = \sqrt{8/3}$ for $\xi \sim \mathcal{N}(0, 1)$.[12] The lemma follows by combining our above displays and using the fact that $\sigma \ge 1$. $\qquad\square$

We now generalize to the case where we actually have truncation on $y$, as is the case in our truncated regression model. First, we show an auxiliary lemma which establishes the sub-Gaussianity of a one-dimensional truncated Gaussian with arbitrary truncation set.

**Lemma F.7.** *Consider a one-dimensional truncated Gaussian $\mathcal{N}(\mu, 1, S)$, and let $a := \mathcal{N}(S; \mu, 1)$ be the mass that the untruncated Gaussian $\mathcal{N}(\mu, 1)$ places on $S$. Let $\xi \sim \mathcal{N}(\mu, 1, S)$. Then*

$$
\|\xi - \mathbb{E}\, \xi\|_{\psi_2} \le O\left( \sqrt{1 + \log \frac{1}{a}} \right) .
$$

*Proof.* We can assume without loss of generality that $\mu = 0$, as the result for general $\mu$ follows by shifting. We begin by upper-bounding the following expectation

$$
\mathbb{E} \exp\left( \frac{(\xi - \mathbb{E}\, \xi)^2}{4} \right) .
$$

For fixed $a = \mathcal{N}(S; \mu, 1)$, the above expectation is maximized by picking a set $S$ of mass $a$ that is as far away from $\mathbb{E}\, \xi$ as possible. Thus, the worst-case $S$ is $(-\infty, -c) \cup (c, \infty)$, which consists of both tails of $\mathcal{N}(0, 1)$, each having mass $a/2$. In other words, $Q(c) \ge a$, where $Q(c) := \Pr_{Z \sim \mathcal{N}(0,1)}(Z \ge c)$ is the Gaussian Q-function. It is standard that $Q(c) \le \exp\left( -\frac{c^2}{2} \right)$, which implies that

$$
c \le \sqrt{2 \log \frac{2}{a}} .
$$

For the above $S$, it is also clear that $\mathbb{E}\, \xi = 0$. Thus, the expectation we wish to upper bound becomes

$$
\mathbb{E} \exp\left( \frac{\xi^2}{4} \right) = \frac{\int_S \exp\left( \frac{z^2}{4} \right) \exp\left( -\frac{z^2}{2} \right) \mathrm{d}z}{\int_S \exp\left( -\frac{z^2}{2} \right) \mathrm{d}z} = \frac{\int_c^\infty \exp\left( -\frac{z^2}{4} \right) \mathrm{d}z}{\int_c^\infty \exp\left( -\frac{z^2}{2} \right) \mathrm{d}z} = \frac{\sqrt{2}\, Q(c/\sqrt{2})}{Q(c)}
$$

---

[12]To see this, note that $\xi^2$ is a chi-squared random variable whose MGF expression gives $\mathbb{E} \exp\left( \frac{\xi^2}{t^2} \right) = \left( 1 - \frac{2}{t^2} \right)^{-1/2}$ for $t \ge \sqrt{2}$, and the right-hand side of this expression becomes 2 when $t = \sqrt{8/3}$

where the second step follows because the integrands are even functions and $S$ is symmetric about the origin. We now claim that

$$\frac{\sqrt{2}Q(c/\sqrt{2})}{Q(c)} \leq 2(1+c^2)\exp\left(\frac{c2}{4}\right).$$

If $c < 1$, this bound can be checked analytically, while if $c \geq 1$ the bound follows by the standard fact that $\frac{x}{1+x^2}e^{-x^2/2} \leq \sqrt{2\pi} \cdot Q(x) \leq \frac{1}{x}e^{-x^2/2}$ for all $x \in \mathbb{R}$. Combining our last three displays with the fact that $1 + x^2 \leq 2e^{x^2/4}$, we obtain that

$$\mathbb{E}\exp\left(\frac{\xi^2}{4}\right) \leq 4\exp\left(\frac{c^2}{2}\right) \leq \frac{8}{a}.$$

Therefore, we have shown for any truncation set $S$ that

$$\mathbb{E}\exp\left(\frac{(\xi - \mathbb{E}\,\xi)^2}{4}\right) \leq \frac{8}{a}.$$

Picking $k := \Theta(1 + \log 1/a) > 1$ such that $(8/a)^{1/k} = 2$, and using Jensen's inequality, we finally obtain

$$\mathbb{E}\exp\left(\frac{(\xi - \mathbb{E}\,\xi)^2}{4k}\right) \leq \left(\mathbb{E}\exp\left(\frac{(\xi - \mathbb{E}\,\xi)^2}{4}\right)\right)^{1/k} \leq 2$$

and the lemma follows by recalling the definition of the sub-Gaussian norm. $\qquad\square$

We are now ready to show concentration of the vector $y \cdot x$ in our truncated regression model.

**Lemma F.8.** *Let* $(x, y) \in \mathbb{R}^d \times \mathbb{R}$ *be a* truncated *random vector such that the marginal of $x$ is $\mathcal{D}_{\text{obs}}$ and the marginal of $y$ conditioned on $x$ is $\mathcal{N}(\langle w^\star, x\rangle, 1, S^\star)$. Let $\mu := \mathbb{E}\,x$, and suppose that Assumptions 1 and 2 hold. Then there exists some absolute constant $C > 0$ such that*

$$\|\langle v, y \cdot x - \mathbb{E}\,y \cdot x\rangle\|_{\psi_1} \leq C \cdot (1 + \|w^\star\|_2) \cdot \left(\sigma^2 \log\frac{1}{\alpha} + \|\mu\|_2^2\right).$$

*Proof.* We proceed similarly to the proof of Lemma F.6. Let $\widehat{\mu} := \mathbb{E}_{\mathcal{D}_{\text{obs}}}\,x$. For any $x$, define the truncated Gaussian random variable $\xi_x \sim \mathcal{N}(0, 1, S^\star - \langle w^\star, x\rangle)$, where $S^\star - \langle w^\star, x\rangle = \{z \mid z + \langle w^\star, x\rangle \in S^\star\}$. Then we can write $y = \langle w^\star, x\rangle + \xi_x$. Hence, letting $v$ be an arbitrary unit vector, we have by Equation (13) that

$$
\begin{aligned}
\|\langle v, y \cdot x - \mathbb{E}\,y \cdot x\rangle\|_{\psi_1} &\leq C' \|\langle v, y \cdot x\rangle\|_{\psi_1} \\
&= C' \|\langle w^\star, x\rangle \langle v, x\rangle + \xi_x \langle v, x\rangle\|_{\psi_1} \\
&\leq C' \left(\|\langle w^\star, x\rangle \langle v, x\rangle\|_{\psi_1} + \|\xi_x \langle v, x\rangle\|_{\psi_1}\right)
\end{aligned}
\tag{14}
$$

where we used the triangle inequality. We will bound each of those terms separately. We have that

$$\|\langle w^\star, x\rangle \langle v, x\rangle\|_{\psi_1} \leq \|\langle w^\star, x\rangle\|_{\psi_2} \|\langle v, x\rangle\|_{\psi_2} \leq \|w^\star\|_2 \cdot O\left(\sigma^2 \log\frac{1}{\alpha} + \|\mu\|_2^2\right)$$

where the first step is by Equation (12) and the second is by Lemma F.5. Similarly, we have that

$$
\begin{aligned}
\|\xi_x \langle x, x\rangle\|_{\psi_1} &\leq \|\xi_x\|_{\psi_2} \|\langle v, x\rangle\|_{\psi_2} \\
&\leq \left(\|\xi_x - \mathbb{E}[\xi_x \mid x]\|_{\psi_2} + \|\mathbb{E}[\xi_x \mid x] - \langle w^\star, x\rangle\|_{\psi_2} + \|\langle w^\star, x\rangle\|_{\psi_2}\right) \cdot \|\langle v, x\rangle\|_{\psi_2}.
\end{aligned}
$$

To bound the two terms appearing in the above expression, observe that, by Lemma F.7

$$\mathbb{E}\left[\exp\left(\frac{(\xi_x - \mathbb{E}[\xi_x \mid x])^2}{4}\right)\right] = \mathop{\mathbb{E}}_{\mathcal{D}_{\text{obs}}}\left[\mathbb{E}\left[\exp\left(\frac{(\xi_x - \mathbb{E}[\xi_x \mid x])^2}{4}\right) \Big| x\right]\right] \leq \mathop{\mathbb{E}}_{\mathcal{D}_{\text{obs}}}\left[\frac{8}{p_{S^\star}(x)}\right] \leq \frac{8}{\alpha}$$

where the last step is by Equation (3) and Assumption 1, and we denote $p_S(x) := p(x, w^\star; S)$ for brevity. Hence, $\|\xi_x - \mathbb{E}\left[\xi_x \mid x\right]\|_{\psi_2} \leq O\left(\sqrt{1 + \log\frac{1}{\alpha}}\right)$. Furthermore, by Lemma D.1, if follows that

$$\underset{\mathcal{D}_{\text{obs}}}{\mathbb{E}}\left[\exp\left(\frac{(\mathbb{E}\left[\xi_x \mid x\right] - \langle w^\star, x\rangle)^2}{4}\right)\right] \leq O(1) \cdot \underset{\mathcal{D}_{\text{obs}}}{\mathbb{E}}\left[\frac{1}{p_{S^\star}(x)}\right] \leq O\left(\frac{1}{\alpha}\right)$$

so again $\|\mathbb{E}\left[\xi_x \mid x\right] - \langle w^\star, x\rangle\|_{\psi_2} \leq O\left(\sqrt{1 + \log\frac{1}{\alpha}}\right)$. Thus, applying Lemma F.5, we obtain that

$$\|\xi_x \langle x, x\rangle\|_{\psi_1} \leq O\left(1 + \|w^\star\|_2\right) \cdot O\left(\sigma^2 \log\frac{1}{\alpha} + \|\mu\|_2^2\right) .$$

We can now substitute our bounds into Equation (14) to obtain the lemma. $\qquad\square$

# G. Necessity of Assumption 3

In this section, we will give a simple example of a setting where Assumption 3 fails and recovering $w^\star$ is impossible, even while the rest of our assumptions are satisfied. Intuitively, if there exists a direction of $\mathbb{R}^d$ where we see almost no samples from the truncated regression model (*i.e.* if the eigenvalues of $\mathbb{E}_{x \sim \mathcal{D}_{\text{obs}}}\left[xx^\top\right]$ are not bounded away from zero), then we cannot hope to recover the component of $w^\star$ along that direction.

To state our example, we begin with some notation. For $i \in [d]$, let $x_i \in \mathbb{R}^d$ be the $i$-th standard unit vector of $\mathbb{R}^d$, so that $\{x_i\}_{i \in [d]}$ are the vertices of the simplex on $\mathbb{R}^d$. Let $\mathcal{D}$ be the uniform distribution over $\{x_i\}_{i \in [d]}$, and let $S^\star := [-1, 1]$ be our truncation set in one dimension. Since $\mathcal{D}$ has bounded support, it is immediate that Assumption 2 is satisfied, and clearly $S^\star$ is a union of a bounded number of intervals. For large $B \gg 1$, we consider two possible parameter vectors $w^{(1)} := B \cdot x_1$ and $w^{(2)} := -B \cdot x_1$. For $j = 1, 2$, let $\mathcal{P}^{(j)}$ denote the distribution over $(x, y) \in \mathbb{R}^d \times \mathbb{R}$ induced by our truncated regression model (Definition 1) with base distribution $\mathcal{D}$, survival set $S^\star$, and parameter vector $w^{(j)}$.

Our claim is that $\mathcal{P}^{(1)}$ and $\mathcal{P}^{(2)}$ have vanishingly small TV distance as $B$ grows, and therefore, even if we know a priori that the true parameter vector is either $w^{(1)}$ and $w^{(2)}$, we still cannot deduce which one it actually is using only sample access to the truncated regression model. To see this, we first compute the survival probability of each point $x_i$ in the support of $\mathcal{D}$. For $i = 1$, by symmetry of the Gaussian distribution and of $S^\star$, the survival probability is the same under both $w^{(1)}$ and $w^{(2)}$, and it is given by

$$\zeta := p_{S^\star}(x_1) = \mathcal{N}(B, 1; [-1, 1]) \leq \exp\left(-\frac{(B-1)^2}{2}\right)$$

where we used standard Gaussian concentration. Clearly, $\zeta \to 0$ as $B \to \infty$. For $j > 1$, the survival probability is again the same under both $w^{(1)}$ and $w^{(2)}$, and it is given by $\beta := p_{S^\star}(x_j) = \mathcal{N}(0, 1; [-1, 1]) \approx 0.68$. Hence, in both cases we have

$$\underset{\mathcal{D}}{\mathbb{E}}\left[p_{S^\star}(x)\right] = \frac{1}{d} \cdot \zeta + \frac{d-1}{d} \cdot \beta =: \alpha .$$

Clearly, $\alpha \geq \beta/2$ for $d \geq 2$ regardless of our choice of $B$, and so Assumption 1 is also satisfied in both cases. We now turn our attention to the observed distribution $\mathcal{D}_{\text{obs}}$ of the features under $w^{(1)}$ and $w^{(2)}$. By Equation (3), it is immediate that $\mathcal{D}_{\text{obs}}$ is the same under both $w^{(1)}$ and $w^{(2)}$, and its mass function is given by

$$\mathcal{D}_{\text{obs}}(x_1) = \frac{\zeta}{\alpha d} \quad \text{and} \quad \mathcal{D}_{\text{obs}}(x_j) = \frac{\beta}{\alpha d}, \ \forall j > 1 .$$

It is clear, then, that Assumption 3 is violated, since $\mathbb{E}_{x \sim \mathcal{D}_{\text{obs}}}\langle x, x_1\rangle^2 = \frac{\zeta}{\alpha d}$, and we can make this arbitrarily small by choosing $B$ sufficiently large. Now, to show our claim, for $j = 1, 2$, let $\mathcal{P}_{y|x_i}^{(j)}$ be the $y$-marginal of $\mathcal{P}^{(j)}$ conditioned on $x_i$,

so that $\mathcal{P}^{(j)}_{y|x_i} = \mathcal{N}(\langle w^{(j)}, x_i \rangle, 1, S^\star)$. For $i > 2$, it is clear that $\mathcal{P}^{(1)}_{y|x_i} = \mathcal{P}^{(2)}_{y|x_i}$ by definition of $w^{(1)}$ and $w^{(2)}$. Therefore

$$
\begin{aligned}
\mathrm{d}_{\mathsf{TV}}(\mathcal{P}^{(1)}, \mathcal{P}^{(2)}) &= \frac{1}{2} \sum_{i \in [d]} \int_{\mathbb{R}^d} \left| \mathcal{P}^{(1)}(x_i, y) - \mathcal{P}^{(2)}(x_i, y) \right| \mathrm{d}y \\
&= \frac{1}{2} \int_{\mathbb{R}^d} \sum_{i \in [d]} \mathcal{D}_{\mathrm{obs}}(x_i) \cdot \left| \mathcal{P}^{(1)}_{y|x_i}(y) - \mathcal{P}^{(2)}_{y|x_i}(y) \right| \mathrm{d}y \\
&= \frac{1}{2} \int_{\mathbb{R}^d} \mathcal{D}_{\mathrm{obs}}(x_1) \cdot \left| \mathcal{P}^{(1)}_{y|x_1}(y) - \mathcal{P}^{(2)}_{y|x_1}(y) \right| \mathrm{d}y \\
&\leq \mathcal{D}_{\mathrm{obs}}(x_1) = \frac{\zeta}{\alpha d}
\end{aligned}
$$

which, again, can be made vanishingly small by choosing $B$ sufficiently large.

