# OpenReview forum: "Linear Regression with Unknown Truncation Beyond Gaussian Features"
_ICML.cc/2026/Conference — ICML 2026 regular_

### Official Review · Reviewer_sesU · 2026-02-28

**Soundness:** 4
**Presentation:** 3
**Significance:** 2
**Originality:** 3
**Overall Recommendation:** 4
**Confidence:** 3

**Summary:**

This paper studies truncated linear regression in a challenging setting where the survival set, the region of outcomes that are observed, is unknown. Prior work gives efficient algorithms when this set is known or, in the unknown set case, under strong Gaussian assumptions on the covariates and with super-polynomial running time. The paper provides the first polynomial-time, polynomial-sample-complexity algorithm for recovering the regression vector when $S^*$ is a union of at most $k$ intervals, assuming only that the covariates are sub-Gaussian and satisfy mild identifiability and survival-probability conditions.

The method proceeds in two phases. First, the authors design a positive-only PAC learning algorithm that can efficiently learn unions of $k$ intervals when given samples from a distribution that is smooth with respect to the target distribution. They show how, in the truncated regression setting, such a smooth auxiliary distribution can be constructed using only sub-Gaussianity. This yields an efficient procedure to learn an approximation set $S$. Second, the paper defines a perturbed negative log-likelihood objective that remains well-behaved even when $S \neq S^*$, establishes strong convexity in a region around $w^*$, and shows how projected SGD with a biased but controlled gradient estimator recovers an $\varepsilon$-accurate estimate of $w^*$ in polynomial time.

Overall, the paper resolves an open problem by demonstrating that truncated regression with unknown truncation is computationally tractable under broad sub-Gaussian assumptions, and introduces a general positive-only interval-learning technique of independent interest.

**Compliance With Llm Reviewing Policy:**

Affirmed.

**Key Questions For Authors:**

1.  The smoothness parameter $q$ heavily influences the sample and runtime complexity in learning $S^*$. Is this dependence inherent, or it is mainly an artifact of the proof technique?

2. The approach fundamentally uses sub-Gaussianity to bound the tail behavior needed for constructing the smooth reference distribution. Is there a way to weaken this to finite second moments (e.g., through truncation, robust mean estimation, or median-of-means arguments)?

3. The PSGD step relies on gradient estimators with controlled bias and variance (Lemmas E.15 and E.16). Could you characterize more concretely how large the hidden constants are, and whether the resulting procedure has reasonable constants for implementation?

4. The final error bound on estimating $w^*$ depends on learning $S^*$ with very small symmetric-difference error. Is there a principled sense in which a coarser approximation of $S^*$ still yields consistent estimation of $w^*$?

**Limitations:**

Yes

**Strengths And Weaknesses:**

Soundness: The theoretical development appears technically solid and carefully structured. The main theorem, polynomial-time recovery of $w^*$ under unknown truncation and sub-Gaussian covariates, is supported by detailed proofs: construction of a smooth auxiliary distribution, a positive-only interval learner with provable guarantees, a warm-start argument for $w^*$, strong convexity of the perturbed NLL, and convergence of PSGD with biased gradients. The assumptions (sub-Gaussianity, survival probability, identifiability) are standard and justified. Potential weaknesses include the somewhat heavy dependence on smoothness parameters and the complexity of the PSGD bias analysis, which may leave room for tighter or clearer bounds.

Presentation: The exposition is thorough and rigorous, with long technical appendices and structured high-level overviews. The division into two phases (learning $S^*$ and learning $w^*$) is logical. However, the paper is dense and at times difficult to comprehend; core intuitions, especially for the smoothness lemma and perturbed NLL, could be made more accessible. More examples or visual illustrations (beyond the toy interval-learning plot) could improve readability. The absence of empirical demonstrations reduces practical intuition although is somewhat reasonable for a theory-focused work.

Significance: The result answers an open question: efficient truncated regression when the survival set is unknown and covariates are non-Gaussian. Since truncation appears across several fields, removing Gaussianity and super-polynomial complexity expands the practical relevance of the problem. The interval-learning subroutine also has independent value in positive-only learning. Significance is primarily theoretical, but the conceptual advancement is clear. The gap between algorithmic theory and practical utility remains large and is expected to persist in the coming years.

Originality: The combination of techniques, positive-only PAC learning under smoothness, construction of a smooth proxy distribution using only sub-Gaussian assumptions, and biased-gradient PSGD analysis for perturbed NLL, is new as a whole. The removal of the Gaussian feature assumption and the improvement from super-polynomial to fully polynomial time represent conceptual progress. While each ingredient draws on existing literature, the assembly into a unified, efficient algorithm is original and not a straightforward extension of previous work.

---

> ### Author Rebuttal · Authors · 2026-03-31
>
> Thank you for supporting our paper. We answer your specific questions below and hope you will strengthen your support.
>
> **The absence of empirical demonstrations reduces practical intuition although is somewhat reasonable for a theory-focused work.**
>
> We would be happy to include numerical experiments! We have run some preliminary experiments and are including the results in our response to reviewer KW5K above.
>
> **The smoothness parameter q heavily influences the sample and runtime complexity in learning $S^\star$. Is this dependence inherent, or it is mainly an artifact of the proof technique?**
>
> We believe that the exponential dependence on $q$ is inherent. It shows up because the smoothness condition only guarantees that $\mathcal{D}^\star(S) \leq O(\mathcal{D}(S)^{1/q})$, see Definition 2 (Lines 306–312). If, for example, $\mathcal{D}^\star(\mathbb{R} \setminus S^\star) \geq 1/2$, then we can only guarantee that $\mathcal{D}(\mathbb{R} \setminus S^\star) \geq \Omega(2^{-q})$, and so we would need to draw $\Omega(2^q)$ samples from the smooth distribution $\mathcal{D}$ before even observing a single sample outside of $S^\star$.
>
> **The approach fundamentally uses sub-Gaussianity to bound the tail behavior needed for constructing the smooth reference distribution. Is there a way to weaken this to finite second moments (e.g., through truncation, robust mean estimation, or median-of-means arguments)?**
>
> Currently, we cannot provably extend beyond sub-Gaussianity. The main difficulty in extending our analysis is to obtain the "smoothed" distribution $\tilde{\mathcal{D}}$ (Lemma 3.4, Lines 377–382). This is a very interesting direction for future work! From a practical perspective, if one can obtain $\tilde{\mathcal{D}}$, then it is possible to extend the rest of our analysis. We would be happy to add a discussion on it in the final version.
>
> **The PSGD step relies on gradient estimators with controlled bias and variance (Lemmas E.15 and E.16). Could you characterize more concretely how large the hidden constants are, and whether the resulting procedure has reasonable constants for implementation?**
>
> The constants in Lemmas E.15 and E.16 are not particularly large. We will add the details of the calculations in the final version. That said, we want to highlight that we have made no attempts to improve the constants and polynomial-dependencies in our results, as is standard in theoretical works.
>
> **The final error bound on estimating $w^\star$ depends on learning $S^\star$ with very small symmetric-difference error. Is there a principled sense in which a coarser approximation of $S^\star$ still yields consistent estimation of $w^\star$?**
>
> The MLE objective is sensitive to the choice of $S\approx S^\star$. For instance, the usual MLE objective becomes ill-defined for any $S \neq S^\star$. So, even to handle sets $S$ which have very small symmetric-difference with $S^\star$, we needed to make significant changes to the MLE objective and consider a "perturbed" objective (Lines 356–370). Understanding if a coarse estimate of $S^\star$ is sufficient for estimating $w^\star$ to a high accuracy is an interesting direction but we suspect it would require significant additional machinery.

---

> > ### Author Rebuttal · Reviewer_sesU · 2026-04-03
> >
> > The authors' responses are detailed and thorough. I have no further questions and will raise my rating from 4 to 5.

---

### Official Review · Reviewer_p8W6 · 2026-03-01

**Soundness:** 3
**Presentation:** 4
**Significance:** 4
**Originality:** 4
**Overall Recommendation:** 5
**Confidence:** 3

**Summary:**

With an unknown survival set $S^\*$ being a union of at most $k$ intervals, this paper gives the first algorithm for truncated linear regression that learns the unknown regression vector $w^*$ to $\epsilon$ accuracy in $\mathrm{poly}(dk/\epsilon)$ time under a set of assumptions that are the same as, or much weaker than, those in prior works. In particular, the authors weaken the Gaussian feature assumption to a sub-Gaussian one and improve the best running time $d^{\mathrm{poly}(k/\epsilon)}$ to $\mathrm{poly}(dk/\epsilon)$.

**Compliance With Llm Reviewing Policy:**

Affirmed.

**Final Justification:**

My concerns have been addressed and I keep my positive score.

**Key Questions For Authors:**

1. Could the existing analytic framework generalize to mis-specified model, i.e., we do not assume specific distribution for $\xi$? If not, what are the key technical challenges?
2. Could the authors comment some high-level idea to extend current techniques to non-linear settings? Perhaps also starting from some simple models such as single-index models.
3. Current algorithm heavily relies on PSGD in Phase II. I would appreciate it if the authors could comment on the feasibility of using other optimization algorithms. Does the current theoretical framework naturally extend to these variants?

**Limitations:**

yes

**Strengths And Weaknesses:**

**Strength:**
1. **Clarity:** The paper is well-written and easy to follow. The authors provide clear, intuitive explanations of their theoretical assumptions and offer a well-structured technical overview that effectively guides the reader.
2. **Significance:** The proposed algorithm achieves the best time complexity with weaker sub-Gaussian assumption.
3. **Technical novelty:** One of the key technical challenges is that relaxing the feature assumption to sub-Gaussian makes the estimation of the survival set $S^*$ highly non-trivial. The authors successfully tackle this challenge by adapting a positive-only PAC learning framework on a restricted hypothesis space of "at most $k$ intervals" and verifying a crucial smoothness condition under sub-Gaussianity.

**Weakness:**
1. **Setting:** The linear regression setting relies on a well-specified model assumption, i.e., $y=x^\top w^* + \xi$ with  $\xi \sim N(0,1)$. Further, the analytic framework is restricted to linear models.
2. **Lack of empirical evaluation:** The paper does not include empirical evaluation. Including numerical experiments to demonstrate the actual runtime scaling and the convergence behavior of the proposed algorithm would strengthen the manuscript.

---

> ### Author Rebuttal · Authors · 2026-03-31
>
> Thank you for your strong support. We are particularly happy that you like the results and theoretical overview. We address your specific comments and questions below.
>
> **Lack of empirical evaluation: The paper does not include empirical evaluation. Including numerical experiments to demonstrate the actual runtime scaling and the convergence behavior of the proposed algorithm would strengthen the manuscript.**
>
> We would be happy to include numerical experiments! We have run some preliminary experiments and are including the results in our response to reviewer KW5K above.
>
> **Could the existing analytic framework generalize to mis-specified model, i.e., we do not assume specific distribution for $\xi$? If not, what are the key technical challenges?**
>
> We believe the analytic framework can handle some non-Gaussian noise distributions $\xi$, using techniques from (Lee et al., 2023). However, this would require knowing the family in which the distributions belong. Developing algorithms which do not assume any distribution for $\xi$ appears much more challenging as, in this case, one cannot even write down the expression of negative log-likelihood. Indeed, this problem is open even for the much easier case where $S^\star$ is known.
>
> **Could the authors comment some high-level idea to extend current techniques to non-linear settings? Perhaps also starting from some simple models such as single-index models.**
>
> This is a great question. If the features are bounded, then our techniques can capture several nonlinear functions via kernel expansion. For example, we can capture polynomial regression by “expanding” the feature vector $x$ into the vector of monomials $(x_1,x_2,x_1x_2,\dots)$. (Here, we require features to be bounded to ensure that the features remain sub-Gaussian after the kernel expansion.)
>
> **Current algorithm heavily relies on PSGD in Phase II. I would appreciate it if the authors could comment on the feasibility of using other optimization algorithms. Does the current theoretical framework naturally extend to these variants?**
>
> We select PSGD for simplicity and it can be replaced by any (projected) stochastic optimization method which comes with provable guarantees for convex functions. Here, the only important requirement is that the method can work with stochastic gradients, since in our problem computing gradients exactly can be NP-hard (e.g., see Daskalakis et al. (2018)).

---

> > ### Author Rebuttal · Reviewer_p8W6 · 2026-04-02
> >
> > Thanks for the detailed rebuttal. I maintain my positive score.

---

### Official Review · Reviewer_KW5K · 2026-03-11

**Soundness:** 3
**Presentation:** 4
**Significance:** 4
**Originality:** 4
**Overall Recommendation:** 5
**Confidence:** 4

**Summary:**

This paper studies truncated linear regression where the survival set $S^\star$ is unknown and the feature distribution is only required to be sub-Gaussian. The main result (Theorem~3.1) gives the first $\mathrm{poly}(dk/\varepsilon)$-time algorithm for recovering $w^\star$ when $S^\star$ is a union of at most $k$ intervals. The algorithm operates in two phases: (I) learning $S^\star$ via a positive-only PAC learning algorithm for unions of intervals under a smoothness condition, and (II) estimating $w^\star$ via projected SGD on a perturbed negative log-likelihood.

The authors intend to analyze a fundamental issue in truncated regression: how to efficiently recover the regression parameter when the truncation set is unknown and must be learned from data. A broad topic addressed by this paper is statistical inference under selection bias with computational efficiency guarantees. Overall, the paper is of high theoretical value, with a well-defined problem, a systematic technical approach, and high quality.

**Compliance With Llm Reviewing Policy:**

Affirmed.

**Key Questions For Authors:**

1.  Could the authors display the explicit form of the exponent (including the exponential terms in $\alpha$ and $\rho$) in a remark after Theorem 3.1, so that readers can assess practical feasibility under heavy truncation or weak identifiability?

2. Although this is primarily a theoretical paper, would the authors consider including a small-scale synthetic simulation illustrating the algorithm's finite-sample behavior---for example, by comparing estimation error as a function of sample size against baselines such as OLS (ignoring truncation) and Oracle MLE (known set)?

**Limitations:**

No limitations are discussed.

**Strengths And Weaknesses:**

Strengths:

1.  The running time is improved from $d^{\mathrm{poly}(k/\varepsilon)}$ (Lee et al., 2024) to $\mathrm{poly}(dk/\varepsilon)$, and simultaneously the feature distribution assumption is relaxed from Gaussian to sub-Gaussian. This is a qualitative leap. Achieving both improvements at the same time is impressive.

2.  The paper resolves two core technical challenges: positive-only set learning and the analysis of the negative log-likelihood under non-Gaussian features. The smoothness lemma (Lemma~3.4), which shows that sub-Gaussianity of the covariates implies the smoothness condition needed for positive-only learning, is a key innovation.

3.  Despite the complex mathematical derivations involved, the authors' decomposition of the algorithm into two phases (set learning and parameter optimization) is very clear. The discussion of astronomical survey applications in the appendix also enhances readability.

Weaknesses:

1.  While Theorem 3.1 claims polynomial complexity in $d, k, 1/\varepsilon$, the exponent $\mathrm{poly}(\sigma, \beta, 1/\rho, 1/\alpha)$ deserves closer scrutiny. Specifically, Corollary E.9 reveals that the strong convexity constant scales as $\exp(-O(M^4\Lambda^4/(\rho^6\alpha^3)))$, where $M$ and $\Lambda$ are moment parameters that scale with $\sigma$ and $\|\mu\|_2$ (see the discussion after Assumption 4). This implies that the hidden constant in the polynomial complexity may be exponentially large in $1/\alpha$ and $1/\rho$, potentially limiting practical tractability under heavy truncation or weak identifiability. While the authors acknowledge this issue in Remark~E.20, the main theorem statement does not make this dependence transparent.

2.  The paper positions itself as providing a computationally efficient solution for practical applications (e.g., astronomy). While the theoretical results are strong, the absence of numerical experiments makes it difficult to gauge the practical impact of the large constants discussed in Weakness~1 and the complexity of the nested subroutines (e.g., rejection sampling). Since computational efficiency is one of the central claims of the paper, even a modest empirical illustration in moderate dimensions would help substantiate this claim and increase confidence in the algorithm's implementability.

---

> ### Author Rebuttal · Authors · 2026-03-31
>
> Thank you for your strong support. We are particularly happy that you liked the technical results and writing. We address your specific comments and questions below.
>
> **Could the authors display the explicit form of the exponent (including the exponential terms in $\alpha$ and $\rho$) in a remark after Theorem 3.1, so that readers can assess practical feasibility under heavy truncation or weak identifiability?**
>
> Certainly. Currently, the dependence of running time and sample complexity can be deduced from Theorems E.1 and E.2 to be $d^{\widetilde{O}((\sigma + \beta)^6 / (\rho^4 \alpha^2))}$.
>
> **Although this is primarily a theoretical paper, would the authors consider including a small-scale synthetic simulation illustrating the algorithm's finite-sample behavior -- for example, by comparing estimation error as a function of sample size against baselines such as OLS (ignoring truncation) and Oracle MLE (known set)?**
>
> Yes, we would be happy to include synthetic simulations! We have run preliminary simulations and are including the details here. Our simulation setup is the following: the feature distribution is a 10-dimensional mixture of five Gaussians, and the truncation set is a union of five intervals. Our baseline is Ordinary Least Squares (OLS), and we compare the performance of three algorithms: Projected Stochastic Gradient Descent (PSGD) on the MLE objective with misspecified truncation set $S$, PSGD on the MLE objective with the correct truncation set $S^\star$, and our algorithm. We note that PSGD with the correct $S^\star$ is an idealized algorithm that knows $S^\star$ a priori; we cannot actually implement it since $S^\star$ is unknown, and we only include it for comparison.
>
> Below are the errors (Euclidean distance from $w^\star$) of each algorithm at specified steps of PSGD for one run of the simulation.
>
> | Step                   | 500  | 1500 | 2500 | 3500 | 4500 |
> |-----------------------|------|------|------|------|------|
> | PSGD with misspecified $S^\star$ | $6.48$ | $6.60$ | $6.78$ | $6.93$ | $7.07$ |
> | PSGD with known $S^\star$         | $5.42$ | $2.23$ | $1.10$ | $0.61$ | $0.48$ |
> | Our algorithm (unknown $S^\star$)          | $5.46$ | $2.33$ | $1.23$ | $0.74$ | $0.56$ |
> | OLS                         |         |         |         |         | $9.81$ |
>
> Repeating the simulation 10 times total gives the following error means and standard deviations for the final iterates of each algorithm.
>
> | Algorithm | Error |
> |--------|-----------|
> | PSGD with misspecified $S^\star$ | $7.17 \pm 0.15$ |
> | PSGD with known $S^\star$  | $0.52 \pm 0.08$ |
> | Our algorithm (unknown $S^\star$)  | $0.65 \pm 0.11$ |
> | OLS | $9.85 \pm 0.25$ |
>
> We see that our algorithm performs comparably to PSGD with known truncation set, and converges to the true $w^\star$. We have uploaded our simulation code at the following anonymous repository: https://anonymous.4open.science/r/truncated-regression-AF02

---

> > ### Author Rebuttal · Reviewer_KW5K · 2026-04-01
> >
> > My questions are fully addressed. I will keep my score.

---

### Official Review · Reviewer_vdk4 · 2026-03-13

**Soundness:** 2
**Presentation:** 2
**Significance:** 3
**Originality:** 3
**Overall Recommendation:** 3
**Confidence:** 2

**Summary:**

This paper aims to provide a computational efficient algorithm for truncated liner regression, where only samples whose the response value falls inside a survival set are revealed. The proposed algorithm applies to the cases where the survival set is unknown and a union of a bounded number of intervals. Compared to previous methods that have polynomial running times but require known survival sets, a main contribution of the proposed algorithm is a computational efficient learning of the unknown survival set.

**Compliance With Llm Reviewing Policy:**

Affirmed.

**Final Justification:**

I will maintain my score as I feel that a significant rewriting is needed to reflect the actual contribution of this work through a more accurate positioning with respect to prior work and a clearer discussion on the underlining complexity of the targeted problem. Meanwhile I will also keep my confidence as low as the topic falls outside my area of expertise.

**Key Questions For Authors:**

- A central claim of this article is an improvement from exponential running time $d^{{\rm poly}{1/\epsilon}}$ to polynomial running time ${\rm poly}(d/\epsilon)$ for truncated linear regression with unknown survival set. The compared previous work by Lee et al. (2024) requires indeed exponential running time $d^{{\rm poly}{1/\epsilon}}$ but it addresses the problem of distribution parameter estimators where the unknown survival set is a $d$-dimensional subspace for the covariates $x\in\mathbb{R}^d$. Meanwhile the present work studies a one-dimensional survival set for the target variable $y\in\mathbb{R}$. Why is such a comparison reasonable?

- Since there exist already algorithms of polynomial running time ${\rm poly}(d/\epsilon)$ for truncated linear regression with known survival sets, the additional computational complexity is mainly due to the estimation of the unknown survival set $S^\star\subset\mathbb{R}$. As the unknown survival set $S^\star$ is one-dimensional, the computational complexity of its estimation should not be related to the dimension $d$ of the feature vector $x$. Could the authors explain why it might require an exponential running time $d^{{\rm poly}{1/\epsilon}}$ to estimate $S^\star\subset\mathbb{R}$?

- Does the proposed algorithm require a known distribution of the feature vector $x\in\mathbb{R}^d$?

- Is the equation in Lines 353-355 (right column) the same as the one in Lines 364-366 (right column)?

- Could the authors provide experimental results on the performance of the proposed algorithm?

**Limitations:**

Yes

**Strengths And Weaknesses:**

Strengths

- Significance&Originality: This paper addresses a relevant problem, namely truncated linear regression with unknown survival sets.

Weaknesses

- Soundness: I have some concerns regarding the claims on the theoretical contributions (Key Questions For Authors). No experiment is provided to test the proposed algorithm.

- Presentation: The article is dense with numerous theoretical results and little illustration, not very accessible for non-expert readers.

---

> ### Author Rebuttal · Authors · 2026-03-31
>
> Thank you for taking the time to review our manuscript. We answer your specific questions below and hope you will strengthen your support for the paper.
>
> **A central claim of this article is an improvement from exponential running time $d^{\mathrm{poly}(1/\varepsilon)}$ to polynomial running time $\mathrm{poly}(d/\varepsilon)$ for truncated linear regression with unknown survival set. The compared previous work by Lee et al. (2024) requires indeed exponential running time $d^{\mathrm{poly}(1/\varepsilon)}$ but it addresses the problem of distribution parameter estimators where the unknown survival set is a $d$-dimensional subspace for the covariates $x \in \mathbb{R}^d$. Meanwhile the present work studies a one-dimensional survival set for the target variable $y \in \mathbb{R}$ . Why is such a comparison reasonable?**
>
> While there are a few works studying the problem of truncated linear regression (including Lee et al. (2024)), none of these works can handle the setting with unknown survival set except Lee et al. (2024). This is why we compare our method to their method. That said, you are correct that their method can handle a more general problem where the survival set is $d$-dimensional. However, the crucial point is that their method does not become faster in the special case where the survival set is $1$-dimensional (since here they learn a $d$-dimensional extension of the one-dimensional object). Moreover, the focus of our work is the case where the feature distribution is sub-Gaussian (and non-parametric). In this case, Lee et al. (2024)’s method does not have provable guarantees (their results only apply to the specific case of Gaussian features).
>
> **Since there exist already algorithms of polynomial running time $\mathrm{poly}(d/\varepsilon)$ for truncated linear regression with known survival sets, the additional computational complexity is mainly due to the estimation of the unknown survival set $S^\star \subset \mathbb{R}$. As the unknown survival set $S^\star$ is one-dimensional, the computational complexity of its estimation should not be related to the dimension $d$ of the feature vector $x$.**
>
> Our learning algorithm is based on two procedures: the first learns the truncation set $S^\star$, and the second learns $w^\star$ given an approximation of $S^\star$. If we isolate the first procedure, and focus on learning $S^\star$ to $\varepsilon$ accuracy, then indeed our sample complexity is $\mathrm{poly}(1 / \varepsilon)$ which has no dependence on $d$ (Theorem E.1, Lines 1201–1216). The dependence on $d$ arises because our goal is to learn $w^\star$, which is a $d$-dimensional object. To estimate $w^\star$ to $\zeta$-acuraccy, we need to estimate the set $S^\star$ to accuracy $\mathrm{poly}(\zeta/d)$ (Theorem E.2, Lines 1234–1256).
>
> We also remark that it is insufficient to estimate $S^\star$ and substitute this estimate into existing algorithms for truncated linear regression. Indeed, these algorithms optimize the standard MLE objective, which we can only do if we know $S^\star$ exactly. Instead, we have to work with the *perturbed* MLE objective (Lines 356–370), and we require significant new analysis to establish that PSGD on this objective converges to $w^\star$.
>
>
> **Could the authors explain why it might require an $d^{\mathrm{poly}(1/\varepsilon)}$ running time to estimate $S \subset \mathbb{R}?$**
>
> As we mentioned, the only prior work handling unknown truncation is Lee et al. (2024). Applying their set estimation algorithm in our setting would yield a $d^{\mathrm{poly}(1/\varepsilon)}$ sample complexity (which seems unavoidable with their method). Moreover, their set estimation algorithm only comes with provable guarantees when the features are Gaussian, so we would not be able to apply it for non-parametric distributions of $x$ (which is the setting we actually consider).
>
> **Does the proposed algorithm require a known distribution of the feature vector $x\in \mathbb{R}^d$?**
>
> No, we do not require knowledge of $x$’s distribution; we discuss this in lines 249-250 (left column).
>
> **Is the equation in Lines 353-355 (right column) the same as the one in Lines 364-366 (right column)?**
>
> This is a typo: in the first expression (lines 353-355), both $S \cap S^\star$ and $S$ should be replaced by $S^\star$. We will fix it.
>
> **Could the authors provide experimental results on the performance of the proposed algorithm?**
>
> We would be happy to include numerical experiments! We have run some preliminary experiments and are including the results in our response to reviewer KW5K below.

---

> > ### Author Rebuttal · Reviewer_vdk4 · 2026-04-04
> >
> > I thank the authors for their reply. Please find below my follow-up comments.
> >
> > Q1: If the authors agree that the method of Lee et al. (2024) was designed to handle unknown $d$-dimensional survival sets, rather than the setup of one-dimensional unknown survival set studied in this paper, then this fundamental difference should be made explicit when claiming the main contribution with respect to related work. To me the main contribution should be more reasonably presented as proposing a method that achieves truncated linear regression under one-dimensional unknown survival set with the same computational efficiency as in truncated linear regression with known survival set.
> >
> > Q2: Thanks for the explanation. I think that these comments are crucial for the readers to grasp the actual computational complexity of the studied problem. Please highlight them in the main text.
> >
> > Q3:  In the case with known survival set $S^\star$, truncated linear regression optimizes the conditional negative log-likelihood (NLL)  $L_{{S^{\star}}}(w)$, which is determined by the joint distribution of $(x,y)\vert y\in S^\star$. This is why we need access to $S^\star$. Since the distribution $\mathcal{D}_{\rm obs}(x)$ of observed $x$ given in (1) depends on $\mathcal{D}(x)$, it would be clearer to explain in the main text  why $\mathcal{D}(x)$ is not required for the proposed method.

---

> > > ### Author Response · Authors · 2026-04-04
> > >
> > > Thank you for engaging with us. We respond to your followup comments and questions below.
> > >
> > > (1) We would be happy to highlight the framing as you suggest.
> > >
> > > (2) We will highlight these in the main body.
> > >
> > > (3) That is a good question. Our method optimizes the pseudo-negative log-likelihood (see first equation defining $\mathcal{L}\_S(\cdot)$ in Section 3.3) instead of the negative log-likelihood. Because of this, crucially, the "log-likelihood" term $\gamma\_{x,w,S}(y)$ does not depend on knowing the distribution of $x$. This is also the objective optimized in truncated linear regression algorithms for *known* survival sets, e.g., (Daskalakis et al., 2019), and, hence, even their method does not need to know the distribution of $x$. We would be happy to include an expanded discussion on this; currently, Lines 351-359 (right column) mention we use the pseudo-negative log-likelihood and Theorem 3.1 mentions we only need sample access to $\mathcal{D}_{\mathrm{obs}}$.

---

### Decision · Program_Chairs · 2026-04-30

**Decision:**

Accept (regular)

**Comment:**

This paper studies truncated linear regression with an unknown survival set, and proposed the first polynomial-time algorithm when the survival set is a union of a bounded number of intervals and the covariates are sub-Gaussian.

Key innovations of this paper include the positive only PAC learning for unions of intervals and the analysis of truncated likelihood optimization beyond Gaussian features. Reviewers characterize the work as resolving an open problem, with contributions that are not straightforward extensions of prior results. The rebuttal is thorough and convincing, addressing all technical questions and clarifying complexity dependencies, assumptions, and positioning relative to prior work. Reviewers also found the added preliminary simulations helpful illustrating finite sample behavior.

Overall, this is a strong theoretical work that advances the understanding of truncated linear regression under realistic assumptions and introduces techniques of independent interest.